# On Diffusion-based Multiplex Dynamic Attributed Network Generator

## Abstract

Multiplex dynamic attributed networks are essential for modeling complex systems, such as social platforms and telecommunication networks, where each layer represents distinct interaction types and attribute dynamics. However, existing generative models fall short in capturing their structural-semantic coupling, temporal evolution, and inter-layer dependencies, failing to reproduce network-level emergent behaviors like explosive synchronization and hysteresis. We introduce MulDyDiff, a diffusion-based generative framework that incorporates attribute-aware dynamic transition-based denoising, cross-layer correlation-aware denoising, and behavior-aware guidance. These components are unified through a novel Behavioral-guided Attributed Cross-layer Temporal (BACT) loss. Evaluations of three real-world datasets demonstrate that MulDyDiff consistently outperforms state-of-the-art dynamic graph generators, achieving 6%-9% improvement in terms of temporal metrics, offering a comprehensive solution for realistic multiplex dynamic attributed network synthesis.

## 1 Introduction

Modeling multiplex dynamic attributed networks (Liu et al., 2020) has gained increasing attention for applications ranging from influence analysis in social platforms (Li et al., 2021; Wu et al., 2022a) to telecommunication and transportation systems (Wan et al., 2020; Tudisco et al., 2018). Unlike single-layer views that collapse heterogeneous interactions, multiplex dynamic attributed networks preserve semantic distinctions across layers and time, revealing phenomena such as **explosive synchronization** (a sudden collective behavior after small perturbations (De Domenico, 2023), exemplified by the 2021 GameStop short squeeze (Bursztynsky, 2021)) and **hysteresis** (where systems resist reverting to prior states (Danziger et al., 2019), as seen in the persistence of remote work post-COVID-19 (Brynjolfsson et al., 2020)). These dynamics arise only in multiplex settings, since the evolution of one layer depends not only on itself but also on other layers across timestamps (see Figure 1 for an example; details in Appendix A).

Despite their importance, multiplex dynamic attributed networks are difficult to model due to data scarcity, privacy constraints (Li et al., 2023; He et al., 2025), and limited public benchmarks (Yang & Leskovec, 2012). Consequently, synthesizing realistic multiplex dynamics has become essential. However, existing generative models (Samanta et al., 2020; Chenthamarakshan et al., 2020; Martinkus et al., 2022; Huang et al., 2022; Vignac et al., 2023; Li et al., 2025) remain inadequate: they fail to jointly capture structural and attributive information, overlook temporal and cross-layer dependencies, and cannot reproduce emergent behaviors such as explosive synchronization and hysteresis. The key challenges are as follows. 1) *Node and edge attributes* are essential for capturing semantics (e.g., user interests) in profiling and classification (Chen et al., 2019; Jin et al., 2021), yet most models focus only on static structures (Jo et al., 2022; Tseng et al., 2023) or intra-layer structural evolution (Zhang et al., 2021a;b; Luo et al., 2021; Hosseini et al., 2025; Zheng et al., 2024), neglecting attribute modeling and even attribute dynamics where a node's state may depend on the structure and attributes of neighbors in the same layer or other layers. 2) *Dependencies across time and layers* are often ignored or oversimplified, even though real-world interactions frequently propagate across platforms and time (Starnini et al., 2017; Fan & Huang, 2020; Zhang et al., 2020b; Wu et al., 2022b; He et al., 2025); existing methods either focus on single-layer graphs (Fan & Huang, 2020) or treat them as static input (Zhang et al., 2020b), failing to jointly capture long-term structural evolution and attribute dynamics. 3) *Emergent behaviors* unique to multiplex dynamics, such as explosive synchronization

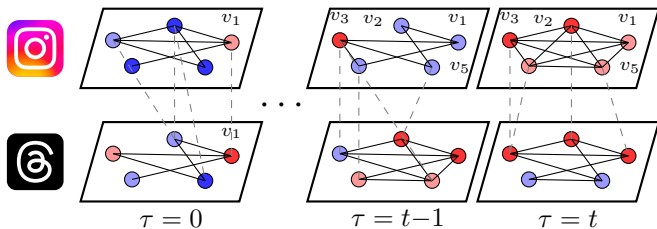

Figure 1: An illustrative example of a multiplex dynamic attributed network. Each layer represents a social media platform (e.g., Instagram, Threads). Nodes denote users with color-coded attributes, intra-layer edges (black solid lines) capture platform-specific interactions, and inter-layer edges (gray dashed lines) represent cross-platform links. The network evolves over time, reflecting both structural and attribute dynamics.

and hysteresis (De Domenico, 2023; Danziger et al., 2019), remain unmodeled in current generative frameworks, leaving a gap in reproducing realistic system-level phenomena.

To address these challenges, we propose *MulDyDiff (Multiplex Dynamic Diffusion Generator)*, a framework to synthesize multiplex dynamic attributed networks with realistic structural, temporal, and semantic characteristics, while preserving emergent behaviors.[1] MulDyDiff consists of: (i) *Attribute-aware dynamic transition-based denoising*, which couples structure and attributes over time; (ii) *Cross-layer correlation-aware denoising*, which reconstructs intra- and inter-layer links in order to capture the evolution between distinct layers; and (iii) *Behavior-aware guidance*, which aligns generated graphs with descriptors of explosive synchronization or hysteresis derived from the Kuramoto model (De Domenico, 2023; Danziger et al., 2019). These components are unified in the *Behavior-guided Attributed Cross-layer Temporal (BACT) loss*, ensuring semantic, temporal, and behavioral fidelity. Our contributions are:

- We propose the first generative framework, *MulDyDiff*, for synthesizing multiplex dynamic attributed networks while preserving network-level behaviors.

- We design a unified denoising framework with *attribute-aware*, *cross-layer correlation-aware*, and *behavior-aware* components, jointly optimized via the *BACT loss*.

- Extensive experiments show that MulDyDiff significantly outperforms dynamic graph generative models by 6%-9% improvement in the KS test of temporal metrics (Longa et al., 2024; Zeno et al., 2021).

## 2 RELATED WORKS

In this section, we compare our study with the related studies, with summarization tables in Appendix B.1 and more related works on static graph generation introduced in Appendix B.2.

**Dynamic Graph Generation.** The existing dynamic graph generators include statistical models and deep generative models. The statistical models mainly consider transitions of structural information between different timestamps (Liu & Sariyüce, 2023; Zeno et al., 2021), but do not consider multiplex structures and changes of node/edge attribute information. The deep generative models consist of auto-regressive approaches (Clarkson et al., 2022; Gupta et al., 2022; Fan & Huang, 2020), variational autoencoder-based approaches (Samanta et al., 2020; Zhang et al., 2021b), GAN-based approaches (He et al., 2025), and streaming-based models (Wang et al., 2022). DBGDGM (Campbell et al., 2024) works on multi-aspect dynamic brain graphs, considering the evolution of embeddings of nodes and clusters, as well as edge generation in each aspect independently at different timestamps, with a hierarchical deep generative model. However, these generators capture structural evolution without explicitly modeling attribute changes, leading to weakened long-term consistency and loss of historical information. They also overlook the joint modeling of temporal dynamics and intra-/inter-layer correlations. While DBGDGM captures embedding evolution, it lacks mechanisms for edge

---

[1]Diffusion models are well-suited for this task: their denoising process supports likelihood-based training to enhance attribute fidelity (Challenge 1), flexible conditioning for temporal and cross-layer dynamics (Challenge 2), and behavior-aware guidance for emergent phenomena (Challenge 3). Moreover, they naturally support permutation-invariant architectures, making them robust for graph generation.

dependencies across time and subjects, and its design is limited to brain graphs, without addressing emergent network-level behaviors in multiplex dynamic attribute networks.

**Diffusion Models.** Compared to statistical and deep generative models, diffusion models directly optimize likelihood and avoid common issues such as mode collapse in GANs or blurry outputs in VAEs, thus preserving structural fidelity. They also flexibly incorporate conditions to model desired structures and capture network-level emergent behaviors. Recently, diffusion models have also been applied to generate multimedia content (Adiya et al., 2024; Zhang et al., 2024; Bar-Tal et al., 2024; Guo et al., 2025), spatial-temporal data(Hu et al., 2024; Liu & Zhang, 2024), or graph data (Niu et al., 2020; Huang et al., 2022; Vignac et al., 2023; Chen et al., 2023; Xu et al., 2024; Bian et al., 2024; Li et al., 2025; Minello et al., 2025). In particular, DiGress (Vignac et al., 2023) synthesizes molecular graphs, exploiting regression guidance to lead the denoising process to generate graphs to meet the structural property. EDGE (Chen et al., 2023) is a discrete diffusion model that exploits graph sparsity to generate graphs while accounting for the change in node degree as a condition. (Xu et al., 2024) proposes a generative diffusion model based on a discrete-state continuous-time setting. Recently, (Bergmeister et al., 2024) develops a scalable graph generative model with progressive expansion techniques. However, the above studies mainly target denoising with the structural information of static graphs, and their denoising networks consider neither correlations between different layers nor emergent network dynamics. Furthermore, most existing diffusion models primarily focus on static graphs or multimedia content with moving objects, without modeling the evolution of structural and attributive information.

## 3 PROBLEM DEFINITION

In this section, we begin by introducing the definition of *multiplex dynamic attribute graphs* and formulating the problem of *multiplex dynamic attribute graph generation* accordingly. The table of notions mentioned in this section is presented in Appendix C.

**Definition 3.1** (Multiplex Dynamic Attributed Network). Given a multiplex dynamic attribute network sequence $\Gamma = \{\mathbb{G}_0, \mathbb{G}_1, \ldots, \mathbb{G}_T\}$ of $T$ $L$-layer graphs, each $L$-layer snapshot $\mathbb{G}_t = (G_t^{(I)}, G_t^{(B)})$ consists of $L$ *intra-layer graphs* $G_t^{(I)} = (\{(\mathbf{X}_{l,t}, \mathbf{E}_{l,t})\}_{l=1}^L)$, and *inter-layer bipartite graphs* $G_t^{(B)} = (\{\mathbf{X}_{l,t}, \mathbf{X}_{m,t}\}, \{\mathbf{B}_{(l,m),t}\}_{l \neq m})$, where $\mathbf{X}_{l,t} \in \mathbb{R}^{a \times N}$ is the node representation (in which there are $N$ nodes with $a$ attributes) of layer $l$ at timestamp $t$; $\mathbf{E}_{l,t} \in \mathbb{R}^{b \times N \times N}$ is the edge representation (in which there are $N \times N$ possible edges with $b$ attributes) of layer $l$ at timestamp $t$; $\mathbf{B}_{(l,m),t} \in \mathbb{R}^{2 \times N \times N}$ is the edge representation representing the existence of inter-layer connections between distinct layers $l$ and $m$.

Note that Definition 3.1 provides a general definition, where dynamic graphs ($L = 1, T \geq 2$) and multiplex graphs ($T = 1, L \geq 2$) are both special cases.

*Example* 3.2. Figure 1 illustrates a toy example of a two-layer dynamic graph capturing user interactions across Instagram and Threads from timestamps $\tau \in \{0, t - 1, t\}$, comprising intra-layer graphs (framed in parallelograms), $G_\tau^{(I)} = \{G_\tau^{Insta}, G_\tau^{Threads}\}$ (with $G_t^{Insta}$ and $G_t^{Threads}$ illustrated in the upper and lower parts, respectively), and inter-layer bipartite graphs, $G_\tau^{(B)}$, with bipartite edges represented by gray dashed lines. The intra-layer graphs represent intra-platform interactions, such as commenting on posts in Instagram or Threads, while the inter-layer bipartite graphs capture cross-layer relationships, such as shared accounts associated with the same user across different platforms or forwarding their posts on one platform to their friends on another platform.

**Definition 3.3** (Multiplex Dynamic Attributed Network Generation). Given an observed historical sequence $\Gamma_{past} = \{\mathbb{G}_0, \mathbb{G}_1, \ldots, \mathbb{G}_{t-1}\}$ of $L$-layer graphs in $t$ timestamps, the aim of this problem is to generate a $L$-layer future graph sequence $\Gamma_{future} = \{\mathbb{G}_t, \mathbb{G}_{t+1}, \ldots, \mathbb{G}_T\}$ with a parameterized model $p_\theta$ such that $p_\theta(\Gamma_{future}|\Gamma_{past})$ is approximated to the true conditional data distribution $p_{data}(\Gamma_{future}|\Gamma_{past})$ by minimizing the discrepancy (e.g., KL divergence) between the learned distribution $p_\theta$ and $p_{data}$, which is equivalent to minimizing the following negative log-likelihood.[2]

$$\min_\theta -\mathbb{E}_{\Gamma \sim p_{data}} \log p_\theta(\Gamma_{future}|\Gamma_{past}).$$

---

[2]The objective of graph generation and forecasting are intrinsically different. The former aims to reproducing data distribution; the latter aims to predict future values (more details are discussed in Appendix **??**).

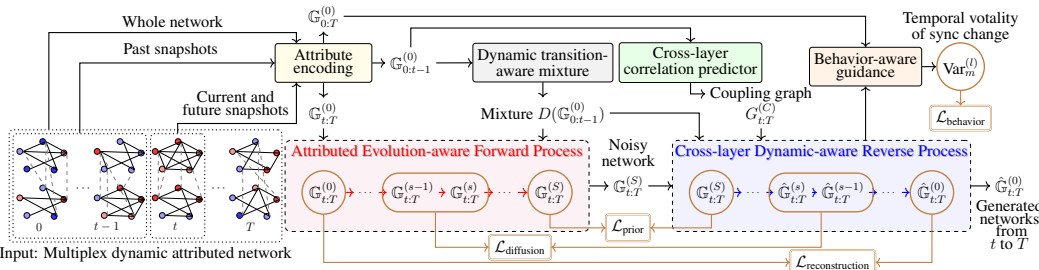

Figure 2: Workflow of MulDyDiff. The red arrows represent the forward diffusion, while the blue arrows denote the reverse denoising (including attribute-aware dynamic transition-based denoising and cross-layer correlation-aware denoising). The regions enclosed in brown rounded boxes are involved in the computation of the BACT loss.

To enable multiplex dynamic attributed network generation, a naïve approach is to independently generate each layer at each timestamp according to existing approaches (Hosseini et al., 2025; Zheng et al., 2024; Luo et al., 2021; Zhang et al., 2021b;a) and fuse them based on predefined cross-layer correlations. However, this approach fails to capture i) the coupling between structure and attributes, ii) the intertwined temporal and cross-layer dependencies, and iii) emergent behaviors such as explosive synchronization. Thus, we propose the framework MulDyDiff for multiplex dynamic graph synthesis based on diffusion models in Sec. 4.

## 4 MULTIPLEX DYNAMIC ATTRIBUTED NETWORK GENERATION

Built upon a discrete denoising diffusion probabilistic model (DDPM), MulDyDiff consists of (1) *attribute-aware dynamic transition-based denoising* , (2) *cross-layer correlation-aware denoising* to jointly capture structural-semantic coupling, temporal, and inter-layer dependencies, and (3) *behavior-aware guidance* to capture emergent behaviors. The workflow is shown in Figure 2, with more details, such as notations, in Appendix C, the background of DDPM in Appendix D, and detailed derivations (including proofs of theorems) in Appendices E and F. For brevity, we show only the node representations $\mathbf{X}_{l,t}$; edge-level formulations $\mathbf{E}_{l,t}$ and $\mathbf{B}_{(l,m),t}$ are defined analogously.

First, *attribute-aware dynamic transition-based denoising* models node/edge attributes and their temporal evolution by embedding both semantics and structure directly into the generative process. A temporal-transition mixture progressively blends past snapshots into each diffusion step, allowing attributes to evolve with neighboring contexts. In contrast, prior models (Clarkson et al., 2022; Gupta et al., 2022; Fan & Huang, 2020; Zhang et al., 2021a;b) either ignore attributes or only add them post hoc, making them unable to capture structure–semantics co-evolution in multiplex settings.

Second, *cross-layer correlation-aware denoising* embeds inter-layer interactions into denoising and leverages observed structural correlations for joint refinement across layers. This enables coherent temporal continuity and realistic intra- and inter-layer dynamics. Existing approaches (Zhang et al., 2021b; 2020a; Shiao et al., 2023; Hosseini et al., 2025; Zheng et al., 2024; Luo et al., 2021) instead treat history as static inputs and process layers in isolation, thus failing to capture coupled temporal and cross-layer evolution.

Third, we introduce *behavior-aware guidance*, incorporating global descriptors (derived from the Kuramoto model) into the denoising objective to encourage phenomena such as *explosive synchronization* and *hysteresis*. Unlike prior works (Campbell et al., 2024; Liu & Sariyüce, 2023; Clarkson et al., 2022; Gupta et al., 2022; Zeno et al., 2021; Zhang et al., 2021a;b) that optimize only for structural fidelity, our approach enforces the reproduction of higher-order dynamics.

Finally, we define the *Behavior-guided Attributed Cross-layer Temporal (BACT) loss*, which combines the three aforementioned notions. By jointly aligning structural, semantic, and behavioral properties, BACT ensures that the generated networks are both statistically faithful and behaviorally plausible.

Figure 2 provides an intuitive view of what the diffusion process captures. The forward process derived in Eqs. (4) to (7) in Sec. 4.2.1 (red arrows) gradually adds noise while continually mixing in clean historical snapshots through the dynamic transition-aware mixture. As a result, each timestamp reflects the accumulated temporal context rather than depending only on $t-1$, enabling the reverse process derived in Eq. (10) in Sec. 4.2.1 (blue arrows) to denoise with a history-aware prior and recover long-range temporal patterns that simple one-step models miss. In addition, the cross-

layer correlation predictor $(p_\theta(G_t^{(C)}|\mathbf{X}_{(1:L),(0:t-1)}^{(0)}))$ in Eq. (14) in Sec. 4.2.2) provides time-varying weights indicating how strongly different layers should influence one another during denoising, allowing the model to exploit implicit cross-layer co-evolution even when no explicit inter-layer edges exist. Moreover, the behavior guidance (Eq. (20) in Sec. 4.2.3) examines the graph-level behavior of the generated graph with Kuramoto model-based synchronization degree to regularize graph-level behavior of generated graphs to be similar to input graphs.

## 4.1 ATTRIBUTED EVOLUTION-AWARE FORWARD PROCESS

Existing approaches that rely solely on historical conditioning often fail to capture multiplex temporal structures (Cachay et al., 2023). We introduce an *attribute-aware dynamic transition mixture* by incorporating the historical snapshots into the diffusion process to jointly diffuse intra- and inter-layer sequences, preserving temporal continuity.

To encode temporal dependency for each layer $l$, we define a recursive mixture D, which encodes the dependency between a snapshot at timestamp $t$ and those during previous timestamps 0 to $t-1$:

$$\mathrm{D}(\mathbf{X}_{l,(0:t)}^{(0)}, \overline{\gamma}_s) = \begin{cases} \overline{\gamma}_s \mathbf{X}_{l,t}^{(0)} + (1 - \overline{\gamma}_s)\mathrm{D}(\mathbf{X}_{l,(0:t-1)}^{(0)}, \overline{\gamma}_s), t \geq 1, \\ \mathbf{X}_{l,0}^{(0)}, t = 0. \end{cases} \tag{1}$$

Starting from the standard categorical distribution-based forward process $\mathbf{Y}_{l,t}^{(s)} = \overline{\alpha}_s \mathbf{X}_{l,t}^{(0)} + (1 - \overline{\alpha}_s)\frac{\overline{\mathbf{1}}_a}{a}$, we define the $s$-th step in the dynamic transition-aware forward process of $\mathbf{X}_{l,t}$ by combining the diffused information at the previous timestamp $t-1$ with the standard categorical forward process at the current timestamp $t$:

$$\mathbf{X}_{l,t}^{(s)} = \overline{\gamma}_s(\overline{\alpha}_s \mathbf{X}_{l,t}^{(0)} + (1 - \overline{\alpha}_s)\frac{\overline{\mathbf{1}}_a}{a}) + (1 - \overline{\gamma}_s)\mathbf{X}_{l,t-1}^{(s)}, \forall s, t \geq 1, \tag{2}$$

with $\overline{\alpha}_s$ controlling noise strength and $\overline{\gamma}_s$ controlling the dependence on previous snapshots. Since $t = 0$ is the starting point, we initialize its forward diffusion to be the same as the traditional diffusion process in Eq. (3).

$$\mathbf{X}_{l,0}^{(s)} = \mathbf{Y}_{l,0}^{(s)} = \overline{\alpha}_s \mathbf{X}_{l,0}^{(0)} + (1 - \overline{\alpha}_s)\frac{\overline{\mathbf{1}}_a}{a}, \tag{3}$$

where $\overline{\mathbf{1}}_a \in \mathbb{R}^{a \times N}$ denotes an all-one matrix. The weight $\overline{\alpha}_s$ determines how much of the original structure is retained.

With Eqs. (1) and (2), the closed-form of the forward process is derived as follows (see Appendix E for details).

$$q(\mathbf{X}_{l,t}^{(s)}|\mathbf{X}_{l,(0:t)}^{(0)}) = \mathrm{Cat}(\overline{\alpha}_s \mathrm{D}(\mathbf{X}_{l,(0:t)}^{(0)}, \overline{\gamma}_s) + (1 - \overline{\alpha}_s)\frac{\overline{\mathbf{1}}_a}{a}), \tag{4}$$

which can be rewritten using the Markov transition matrix $\overline{\mathbf{Q}}_{\mathbf{X}_{l,(0:t-1)}}^{(s)} = \overline{\alpha}_s \overline{\gamma}_s \mathbf{I} + \overline{\alpha}_s(1 - \overline{\gamma}_s)\mathbf{1}_a \mathrm{D}(\mathbf{X}_{l,(0:t-1)}^{(0)}, \overline{\gamma}_s)^\top + (1 - \overline{\alpha}_s)\mathbf{1}_a \frac{\overline{\mathbf{1}}_a^\top}{a}$ (see Appendix E for details):

$$q(\mathbf{X}_{l,t}^{(s)}|\mathbf{X}_{l,(0:t)}^{(0)}) = Cat(\mathbf{X}_{l,t}^{(s)}; \overline{\mathbf{Q}}_{\mathbf{X}_{l,(0:t-1)}}^{(s)}{}^\top \mathbf{X}_{l,t}^{(0)}) = \mathbf{X}_{l,t}^{(s)}{}^\top \overline{\mathbf{Q}}_{\mathbf{X}_{l,(0:t-1)}}^{(s)}{}^\top \mathbf{X}_{l,t}^{(0)}. \tag{5}$$

By subtracting $\alpha_s \gamma_s \mathbf{X}_{l,t}^{(s-1)}$ from $\mathbf{X}_{l,t}^{(s)}$ (using Eq. (4) to cancel out $\mathbf{X}_{l,t}^{(0)}$), we derive the single-step time-aware stepwise transition process as follows (see Appendix E for details).

$$q(\mathbf{X}_{l,t}^{(s)}|\mathbf{X}_{l,t}^{(s-1)}, \mathbf{X}_{l,(0:t)}^{(0)}) = \mathrm{Cat}(\alpha_s \gamma_s \mathbf{X}_{l,t}^{(s-1)} + \overline{\alpha}_s[(1 - \overline{\gamma}_s)\mathrm{D}(\mathbf{X}_{l,(0:t-1)}^{(0)}, \overline{\gamma}_s)$$
$$- (\gamma_s - \overline{\gamma}_s)\mathrm{D}(\mathbf{X}_{l,(0:t-1)}^{(0)}, \overline{\gamma}_{s-1})] + [1 - \alpha_s \gamma_s - \overline{\alpha}_s(1 - \gamma_s)]\frac{\overline{\mathbf{1}}_a}{a}), \tag{6}$$

which can be rewritten using $\mathbf{Q}_{\mathbf{X}_{l,(0:t-1)}}^{(s)} = \overline{\alpha}_s \overline{\gamma}_s \mathbf{I} + \overline{\alpha}_s(1 - \overline{\gamma}_s)\mathbf{1}_a \mathrm{D}(\mathbf{X}_{l,(0:t-1)}^{(0)}, \overline{\gamma}_s)^\top + (1 - \overline{\alpha}_s)\mathbf{1}_a \frac{\overline{\mathbf{1}}_a^\top}{a}$:

$$q(\mathbf{X}_{l,t}^{(s)}|\mathbf{X}_{l,t}^{(s-1)}, \mathbf{X}_{l,(0:t)}^{(0)}) = Cat(\mathbf{X}_{l,t}^{(s)}; \mathbf{Q}_{\mathbf{X}_{l,(0:t-1)}}^{(s)}{}^\top \mathbf{X}_{l,t}^{(s-1)}) = \mathbf{X}_{l,t}^{(s)}{}^\top \mathbf{Q}_{\mathbf{X}_{l,(0:t-1)}}^{(s)}{}^\top \mathbf{X}_{l,t}^{(s-1)}, \tag{7}$$

and the following theorem shows that Eq. (4) is the marginal distribution of Eq. (6) (see Appendix E for details).

**Theorem 4.1.** *Eq. (4) gives the marginal distribution of Eq. (6), i.e.,*

$$q(\mathbf{X}_{l,t}^{(s)}|\mathbf{X}_{l,(0:t)}^{(0)}) = \sum_{\mathbf{X}_{l,t}^{(s-1)}} q(\mathbf{X}_{l,t}^{(s)}|\mathbf{X}_{l,t}^{(s-1)}, \mathbf{X}_{l,(0:t)}^{(0)})q(\mathbf{X}_{l,t}^{(s-1)}|\mathbf{X}_{l,(0:t)}^{(0)}), \forall s = 1, \ldots, S.$$

*Proof.* See Appendix E for details. □

From Eq. (4), we derive the prior loss as follows.

$$\mathcal{L}_{\text{prior}} = D_{KL}[q(\mathbb{G}_0^{(S)}|\mathbb{G}_0^{(0)})\|p_\theta(\mathbb{G}_0^{(S)})] + \sum_{t=1}^{T} D_{KL}[q(\mathbb{G}_t^{(S)}|\mathbb{G}_{0:t}^{(0)})\|p_\theta(\mathbb{G}_t^{(S)})], \tag{8}$$

where $D_{KL}$ represents the Kullback–Leibler (KL) divergence between the prior $p_\theta(\mathbb{G}_t^{(S)})$ and the diffusion process $q(\mathbb{G}_t^{(S)}|\mathbb{G}_{0:t}^{(0)})$ for $t \geq 0$.

**Remark.** Unlike prior temporal graph generators (Campbell et al., 2024; Liu & Sariyüce, 2023; Wang et al., 2022; Gupta et al., 2022; Zeno et al., 2021), our forward process explicitly encodes temporal dependencies between the current and previous timestamps, enabling history-guided denoising and naturally supporting forecasting via conditional generation. The differences between MulDyDiff and prior temporal graph generators are presented in Appendix B.3.

### 4.2 CROSS-LAYER DYNAMIC-AWARE REVERSE PROCESS

In the *cross-layer dynamic-aware reverse process*, we extend the forward process in Sec. 4.1 to an *attribute-aware transition-based denoising process*, further incorporating cross-layer correlations into a *correlation-aware denoising process* to model critical cross-layer transitions. Finally, we introduce *behavior-aware guidance* to steer generation such that explosive synchronization of node attributes emerges at specific timestamps.

#### 4.2.1 ATTRIBUTE-AWARE DYNAMIC TRANSITION-BASED DENOISING

To reverse the diffusion, we use Bayes' theorem to compute the posterior distribution over the previous noisy state given the current noisy state and the clean history. From Eqs. (5) and (7), we derive the posterior of the forward process $q$ as stated in the following theorem:

**Theorem 4.2.**

$$q(\mathbf{X}_{l,t}^{(s-1)}|\mathbf{X}_{l,t}^{(s)}, \mathbf{X}_{l,(0:t)}^{(0)}) = \mathbf{X}_{l,t}^{(s-1)^\top} \frac{\mathbf{Q}_{\mathbf{X}_{l,(0:t-1)}}^{(s)} \mathbf{X}_{l,t}^{(s)} \odot \overline{\mathbf{Q}}_{\mathbf{X}_{l,(0:t-1)}}^{(s-1)^\top} \mathbf{X}_{l,t}^{(0)}}{\mathbf{X}_{l,t}^{(s)^\top} \overline{\mathbf{Q}}_{\mathbf{X}_{l,(0:t-1)}}^{(s)^\top} \mathbf{X}_{l,t}^{(0)}}. \tag{9}$$

*Proof.* See Appendix E for details. □

With Eq. (9), we approximate the denoising process by conditioning on past snapshots $\mathbb{G}_{0:t-1}$ to generate the current snapshot $\hat{\mathbb{G}}_t$ via a dynamic transition-aware denoising network (Eq. (10)). Specifically, at each timestamp $t$, we denoise from a noisy graph $\mathbb{G}_t^{(S)}$ to a denoised graph $\hat{\mathbb{G}}_t$ while incorporating historical context $\mathbb{G}_{0:t-1}$.

The reverse denoising process for node attributes is formulated as follows:

$$p_\theta(\mathbf{X}_{l,t}^{(s-1)}|\mathbf{X}_{l,t}^{(s)}, \mathbf{X}_{l,(0:t-1)}^{(0)}) = \sum_{\mathbf{X}_{l,t}^{(0)}} q(\mathbf{X}_{l,t}^{(s-1)}|\mathbf{X}_{l,t}^{(s)}, \mathbf{X}_{l,(0:t-1)}^{(0)}, \mathbf{X}_{l,t}^{(0)})\hat{p}_{l,t}^{(X)}(\mathbf{X}_{l,t}^{(0)}|\mathbf{X}_{l,t}^{(s)}, \mathbf{X}_{l,(0:t-1)}^{(0)}),$$

$$\tag{10}$$

where the first term in the summation is the posterior of the forward process, and the second term in the summation is the probability distribution learned via a dynamic transition-aware denoising network.

#### 4.2.2 CROSS-LAYER CORRELATION-AWARE DENOISING

To facilitate denoising with cross-layer states for accurate multiplex dynamic attributed network generation, we further extend the denoising process in Sec. 4.2.1 by incorporating cross-layer correlations. To accurately capture cross-layer correlations, we first define the *cross-layer coupling graph*, which serves as a structural guide, identifying the relevant layers to incorporate during denoising.

**Definition 4.3** (Cross-layer Coupling Graph). A cross-layer coupling graph $G_t^{(C)} = (V_t^{(C)}, E_t^{(C)})$ is defined at timestamp $t$ to represent the structural dependencies between layers in a multiplex dynamic attributed network. The node set $V_t^{(C)} = \{1, \ldots, L\}$ corresponds to the $L$ layers in the graph. The edge set $E_t^{(C)}$ encodes the existence of inter-layer connections, i.e., $E_t^{(C)} = \{(l, m) \mid l, m \in V_t^{(C)}, \mathbf{B}_{(l,m),t} \neq \mathbf{0}\}$, where $\mathbf{B}_{(l,m),t}$ represents the edges between the nodes in the $l$-th and

$m$-th layers. An edge $(l, m) \in E_t^{(C)}$ indicates at least one cross-layer connection between the nodes in the $l$-th and $m$-th layers.

We first assume that the forward noising processes in layers $l = 1, \ldots, L$ are independent. Then the forward process of node representations in layers $l = 1, \ldots, L$ of a multi-layer temporal graph sequence is extended from Eqs. (5) and (7) as follows:

$$q(\mathbf{X}_{(1:L),t}^{(s)}|\mathbf{X}_{(1:L),(0:t)}^{(0)}) = \prod_{l=1}^{L} q(\mathbf{X}_{l,t}^{(s)}|\mathbf{X}_{l,(0:t)}^{(0)});$$

$$q(\mathbf{X}_{(1:L),t}^{(s-1)}|\mathbf{X}_{(1:L),t}^{(s)}, \mathbf{X}_{(1:L),(0:t)}^{(0)}) = \prod_{l=1}^{L} q(\mathbf{X}_{l,t}^{(s)}|\mathbf{X}_{l,t}^{(s-1)}, \mathbf{X}_{l,(0:t)}^{(0)}). \tag{11}$$

Thus, the posterior of the forward process of a multi-layer temporal graph sequence is extended from Eq. (9) as follows (see Appendix F for details):

$$q(\mathbf{X}_{(1:L),t}^{(s-1)}|\mathbf{X}_{(1:L),t}^{(s)}, \mathbf{X}_{(1:L),(0:t)}^{(0)}) = \prod_{l=1}^{L} q(\mathbf{X}_{l,t}^{(s-1)}|\mathbf{X}_{l,t}^{(s)}, \mathbf{X}_{l,(0:t)}^{(0)}). \tag{12}$$

Formally, layers in a multi-layer temporal graph sequence exhibit interdependencies and implicit co-evolution, which can be modeled by the denoising distribution expressed as the product of the distributions of all layers conditioned on $\mathbf{X}_{(1:L),(0:t-1)}^{(0)}$ as follows (see Appendix F for details):

$$p_\theta(\mathbf{X}_{(1:L),t}^{(0)}|\mathbf{X}_{(1:L),t}^{(s)}, \mathbf{X}_{(1:L),(0:t-1)}^{(0)}) = \prod_{l=1}^{L} p_\theta(\mathbf{X}_{l,t}^{(0)}|\mathbf{X}_{l,t}^{(s)}, \mathbf{X}_{(1:L),(0:t-1)}^{(0)}), \tag{13}$$

where $p_\theta(\mathbf{X}_{l,t}^{(0)}|\mathbf{X}_{l,t}^{(s)}, \mathbf{X}_{(1:L),(0:t-1)}^{(0)})$ is learned by a denoising network with a cross-layer attention mechanism, with weights determined by the cross-layer coupling graph $G_t^{(C)}$ predicted by a learnable prior over cross-layer dependencies $p_\theta(G_t^{(C)}|\mathbf{X}_{(1:L),(0:t-1)}^{(0)})$ (see Appendix F for details).

$$p_\theta(\mathbf{X}_{l,t}^{(0)}|\mathbf{X}_{l,t}^{(s)}, \mathbf{X}_{(1:L),(0:t-1)}^{(0)}) = \sum_{G_t^{(C)}} p_\theta(\mathbf{X}_{l,t}^{(0)}|\mathbf{X}_{l,t}^{(s)}, \mathbf{X}_{(1:L),(0:t-1)}^{(0)}, G_t^{(C)}) p_\theta(G_t^{(C)}|\mathbf{X}_{(1:L),(0:t-1)}^{(0)}). \tag{14}$$

The reverse process of a multi-layer graph sequence conditioning on the clean snapshots $\mathbf{X}_{(1:L),(0:t-1)}^{(0)}$ is the product of the reverse processes of all layers $l$ as follows:

$$p_\theta(\mathbf{X}_{(1:L),t}^{(s-1)}|\mathbf{X}_{(1:L),t}^{(s)}, \mathbf{X}_{(1:L),(0:t-1)}^{(0)}) = \prod_{l=1}^{L} p_\theta(\mathbf{X}_{l,t}^{(s-1)}|\mathbf{X}_{l,t}^{(s)}, \mathbf{X}_{(1:L),(0:t-1)}^{(0)}), \tag{15}$$

where the reverse process of each layer $l$ can be derived by approximation using the posterior (Eq. (12)) and denoiser (Eq. (14)) of each layer $l$ (see Appendix F for details),

$$p_\theta(\mathbf{X}_{l,t}^{(s-1)}|\mathbf{X}_{l,t}^{(s)}, \mathbf{X}_{(1:L),(0:t-1)}^{(0)}) = \sum_{\mathbf{X}_{l,t}^{(0)}} q(\mathbf{X}_{l,t}^{(s-1)}|\mathbf{X}_{l,t}^{(s)}, \mathbf{X}_{l,(0:t)}^{(0)}) p_\theta(\mathbf{X}_{l,t}^{(0)}|\mathbf{X}_{l,t}^{(s)}, \mathbf{X}_{(1:L),(0:t-1)}^{(0)}). \tag{16}$$

The architectures of the denoising network with a cross-layer correlation predictor are presented in Appendix G.

From the above denoising process, we derive the reconstruction loss and diffusion loss as follows.

$$\mathcal{L}_{\text{reconstruction}} = -\log p_\theta(\mathbb{G}_0^{(0)}|\mathbb{G}_0^{(1)}) - \sum_{t=1}^{T} -\log p_\theta(\mathbb{G}_t^{(0)}|\mathbb{G}_t^{(1)}, \mathbb{G}_{0:t-1}^{(0)}), \tag{17}$$

$$\mathcal{L}_{\text{diffusion}} = \sum_{s=2}^{S-1} \left[ D_{KL}[q(\mathbb{G}_0^{(s-1)}|\mathbb{G}_0^{(s)}, \mathbb{G}_0^{(0)})\|p_\theta(\mathbb{G}_0^{(s-1)}|\mathbb{G}_0^{(s)})] \right.$$

$$\left. + \sum_{t=1}^{T} D_{KL}[q(\mathbb{G}_t^{(s-1)}|\mathbb{G}_t^{(s)}, \mathbb{G}_{0:t}^{(0)})\|p_\theta(\mathbb{G}_t^{(s-1)}|\mathbb{G}_t^{(s)}, \mathbb{G}_{0:t-1}^{(0)})] \right], \tag{18}$$

which calculates the KL divergence between the true posterior $q(\mathbb{G}_t^{(s-1)}|\mathbb{G}_t^{(s)}, \mathbb{G}_{0:t}^{(0)})$ in Eq. (18) and the reverse denoising process $p_\theta(\mathbb{G}_t^{(s-1)}|\mathbb{G}_t^{(s)}, \mathbb{G}_{0:t-1}^{(0)})$ in Eq. (18). The former is the product of the dynamic transition-aware posterior of nodes (derived in Eq. (12); similarly for intra-/inter-layer

edges); the latter is the product of the denoising processes of nodes and intra-/inter-layer edges (derived in Eq. (15)).

### 4.2.3 BEHAVIOR-AWARE GUIDANCE

To softly steer the generative process toward realistic global dynamics, we introduce *behavior-aware guidance* based on external descriptors. Specifically, for explosive synchronization, we compute layer-wise synchronization via the Kuramoto order parameter (De Domenico, 2023; Danziger et al., 2019) and track its temporal volatility to detect abrupt alignment shifts, forming a descriptor that guides generation toward emergent behaviors. The motivation for behavior-aware guidance (with an illustrative example), as well as the description of hysteresis, is presented in Appendix H.

**Temporal Vitality of synchronization.** To characterize emergent dynamics, we use the Kuramoto-based synchronization measure $R_m^{(l)}(t)$, which quantifies the degree of phase coherence (detailed in Appendix H) to compute the variance of synchronization change as a descriptor, quantifying the volatility of temporal alignment across nodes. For the $m$-th attribute in layer $l$, we define the first-order difference by $\Delta R_m^{(l)}(t) = R_m^{(l)}(t+1) - R_m^{(l)}(t)$ for $t = 1, \ldots, T-1$ and calculate its variance as

$$\text{Var}_m^{(l)} = \frac{1}{T-1} \sum_{t=1}^{T-1} \left( \Delta R_m^{(l)}(t) - \overline{\Delta R}_m^{(l)} \right)^2, \tag{19}$$

where $\overline{\Delta R}_m^{(l)} = \frac{1}{T-1} \sum_{t=1}^{T-1} \Delta R_m^{(l)}(t)$. Larger variance values indicate abrupt synchronization changes, which are key markers of explosive dynamics.

To encourage the emergence of realistic dynamic phenomena, we incorporate a behavioral loss based on the Kuramoto-based descriptor $\text{Var}_m^{(l)}$. To ensure training stability and gradient flow, we adopt a smooth surrogate using softmax aggregation:

$$\mathcal{L}_{\text{behavior}} = -\log \left( \sum_{m=1}^{M} \sum_{l=1}^{L} \exp(\lambda \cdot \text{Var}_m^{(l)}) \right), \tag{20}$$

where $\lambda > 0$ controls the sharpness of aggregation.[3]

**Remark.** 1) Eq. (10) denoises $\mathbf{X}_{l,t}^{(s)}$ based on past snapshots using $\hat{p}_{l,t}^{(X)}$ learned from the denoising network, overcoming the limits of temporal graph generative models (Starnini et al., 2017; Fan & Huang, 2020; Zhang et al., 2020b; Wu et al., 2022b; He et al., 2025). 2) The cross-layer correlation-aware network learns the conditional distribution of clean intra- and inter-layer graphs from the given past snapshots with the assistance of $G_t^{(C)}$, addressing the limitations of prior multiplex and diffusion-based generators (Zhang et al., 2020a; Shiao et al., 2023; Niu et al., 2020; Huang et al., 2022; Vignac et al., 2023; Chen et al., 2023; Xu et al., 2024). 3) The variance $\text{Var}_m^{(l)}$ serves as a proxy for detecting sudden shifts or persistent irregularities in synchronization, which are indicative of higher-order network behaviors. By simply adding Eq. (20) as the behavior loss, this steers the generative process toward reproducing the global behaviors observed in multiplex systems.

Building on the preceding designs, we propose the *Behavior-guided Attributed Cross-layer Temporal (BACT) loss*, which jointly accounts for attribute consistency in reconstruction loss (Eq.( 17)), cross-layer correlation-aware temporal dependencies in prior and diffusion loss (Eq. (8) and Eq. (18)), and emergent behavioral signals (Eq. (20)) in multiplex dynamic attributed networks.

$$\mathcal{L}_{BACT} = \mathcal{L}_{\text{reconstruction}} + \mathcal{L}_{\text{prior}} + \mathcal{L}_{\text{diffusion}} + \mathcal{L}_{\text{behavior}}.$$

## 5 EXPERIMENTS

**Datasets.** The experiments are conducted on three real-world multiplex temporal networks: 1) Wiki-vote (Leskovec et al., 2010), 2) Twitter (De Domenico et al., 2013), and 3) Superuser (Paranjape et al., 2017). The statistics and descriptions of the datasets are presented in Appendix J.

**Baselines.** We compare the proposed models with the following baseline temporal graph generators: (1) AGE Fan & Huang (2020): an attention-based graph evolution model that considers the transformation between graphs in different states; (2) DAMNETS Clarkson et al. (2022): a deep generative

---

[3]We can add losses of all behaviors if the guidance of multiple behaviors is needed.

Table 1: Comparative study results on KS metrics.

| Data/Model | | node behavior (↓) | RW (↓) | degree centrality (↓) | betweenness centrality (↓) |
|---|---|---|---|---|---|
| Wiki-vote | AGE | 0.8052 | 0.0766 | 0.9991 | 0.8592 |
| | DAMNETS | 0.6853 | 0.1805 | 0.7152 | 0.7318 |
| | TagGen | 0.9500 | 0.1400 | 0.8750 | 0.8500 |
| | DYMOND | 0.8256 | 0.2111 | 0.8398 | 0.8941 |
| | MoDiff | 1.0000 | 0.4000 | 0.9474 | 0.8500 |
| | MulDyDiff | 0.5430 | 0.2219 | 0.8281 | 0.6563 |
| Twitter | AGE | 0.7122 | 0.2086 | 0.9257 | 0.8943 |
| | DAMNETS | 0.6325 | 0.0851 | 0.8133 | 0.8002 |
| | TagGen | 0.9000 | 0.2600 | 0.6750 | 0.9000 |
| | DYMOND | 0.6791 | 0.0602 | 0.6631 | 0.7484 |
| | MoDiff | 1.0000 | 0.3000 | 1.0000 | 0.7288 |
| | MulDyDiff | 0.5957 | 0.1172 | 0.6489 | 0.6895 |
| Superuser | AGE | 0.8988 | 0.2286 | 0.9870 | 0.6513 |
| | DAMNETS | 0.6000 | 0.1544 | 0.8156 | 0.7325 |
| | TagGen | 0.7000 | 0.1800 | 0.5250 | 0.5500 |
| | DYMOND | 0.6937 | 0.0470 | 0.6326 | 0.6815 |
| | MoDiff | 1.0000 | 0.0500 | 1.0000 | 0.1069 |
| | MulDyDiff | 0.5484 | 0.1684 | 0.6411 | 0.6119 |

model that generates temporal graph sequences in an autoregressive manner with a GAT-based encoder-decoder architecture; (3) TagGen (Zhou et al., 2020): a generative model based on temporal random walks; (4) DYMOND (Zeno et al., 2021): a generative model that captures dynamic changes with temporal motif activities; (5) MoDiff (Xu & Ma, 2025): a diffusion model that considers the spectral properties of motifs. The comparison between the time complexities of MulDyDiff and the baselines is presented in Appendix I.

**Metrics.** The performance metrics include: 1) Kolmogorov-Smirnov (KS) distance (Zeno et al., 2021; Longa et al., 2024), which evaluates temporal fidelity by comparing the distributions of structural metrics (node behavior, random walk (RW), degree centrality, and betweenness centrality) between real and generated graphs at each timestamp using the KS statistic; 2) explosive synchronization degree $R(t)$; 3) Maximum Mean Discrepancy (MMD) of degree distributions and spectral values, etc. Clarkson et al. (2022); Martinkus et al. (2022); Chen et al. (2023); Vignac et al. (2023); and 4) training and sampling time. Due to space constraints, we present the results of the KS distance and explosive synchronization degree in this section, with more details of the experimental setup and more results are reported in Appendices J and K, respectively.[4]

## 5.1 KS EVALUATION

Table 1 presents the evaluation results in KS metrics of MulDyDiff compared with baselines on Wiki-vote, Twitter, and Superuser, as KS metrics are more suitable than MMD metrics for (multi-layer) temporal graphs (Longa et al., 2024; Zeno et al., 2021) (with a detailed explanation in Appendix J and MMD results in Appendix K). On the Wiki-vote dataset, MulDyDiff outperforms the baselines in almost all metrics listed in the table, as it captures structural and attributive evolution simultaneously. Some baselines perform slightly better in the KS of random walk on the Wiki-vote dataset. Nevertheless, MulDyDiff overall outperforms these baselines since they only perform well in one or two KS metrics. This is insufficient to demonstrate the effectiveness of the baselines in multi-layer temporal graph generation, as the effectiveness needs to be assessed comprehensively by various metrics. On the Twitter and Superuser datasets, MulDyDiff outperforms almost all other methods in terms of the KS of node behaviors, with a 6%-9% improvement (compared with the second-best competitor, DAMNETS) because the metric can effectively examine whether a generative model captures both intra- and inter-layer relationships during generation. MulDyDiff also shows stable performance on other metrics. In contrast, TagGen performs better in the KS of degree centrality on the Superuser dataset, but it performs worse in terms of the KS of node behavior and random walk. DAMNETS performs second-best in terms of the KS of node behavior and random walk on the Twitter dataset, but it performs worse regarding other metrics. Although DYMOND takes motif sampling into consideration, it achieves performance comparable to that of MulDyDiff on the Twitter dataset. However, since it is unable to deal with the multi-layer structure, it performs 9% worse than MulDyDiff in terms of the KS of node behavior.

---

[4]The source code is published in the anonymous repository: `https://anonymous.4open.science/r/MulDyDiff-8815`

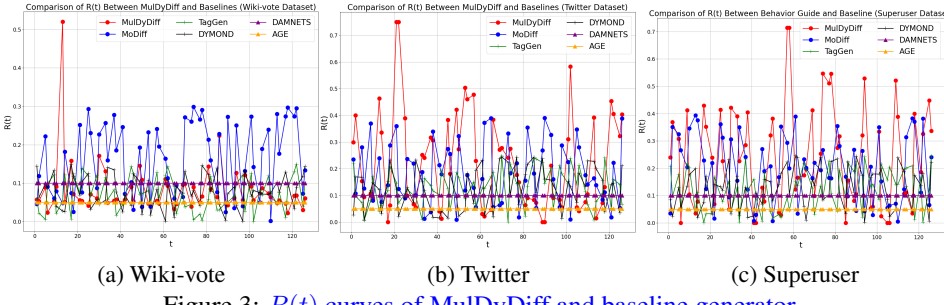

(a) Wiki-vote        (b) Twitter        (c) Superuser

Figure 3: $R(t)$ curves of MulDyDiff and baseline generator.

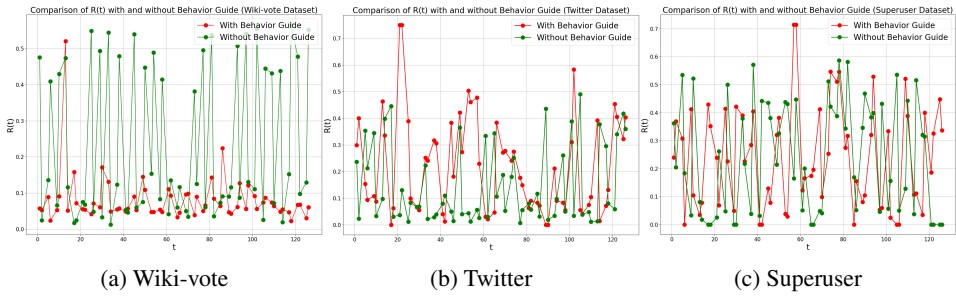

(a) Wiki-vote        (b) Twitter        (c) Superuser

Figure 4: $R(t)$ curves with and without $\mathcal{L}_{behavior}$.

## 5.2 BEHAVIOR GUIDANCE EVALUATION

**Comparisons with baselines.** We conduct the evaluations using the synchronization degree $R(t)$ to assess explosive synchronization in graph sequences generated by MulDyDiff and the baselines in Figs. 3a to 3c. Across the three datasets, MulDyDiff (red) exhibits clear and sharp peaks corresponding to explosive increases of $R(t)$, indicating that MulDyDiff can generate node attributes that faithfully reflect emergent synchronization when trained with $\mathcal{L}_{behavior}$. In contrast, methods with node attributes that do not change over time, such as AGE (orange) and DAMNETS (purple), produce flat curves with constant values of $R(t)$, showing that static attributes cannot trigger emergent synchronization. Methods with time-varying node attributes, such as MoDiff (blue), yield strongly oscillatory $R(t)$ curves but without clear explosive peaks, suggesting that simply perturbing node attributes over time is insufficient to capture emergent behaviors. Finally, TagGen (green) and DYMOND (black), which do not model node attributes and instead assign them randomly in post-processing, only display random fluctuations in $R(t)$ without any pronounced bursts.

**Ablation study with and without $\mathcal{L}_{behavior}$.** Figs. 4a to 4c present the results of $R(t)$ curves on the Wiki-vote, Twitter, and Superuser datasets with $\mathcal{L}_{behavior}$ (red) and without $\mathcal{L}_{behavior}$ (green). On the Wiki-vote dataset, MulDyDiff is able to capture the abrupt increase in $R(t)$ at $t = 13$ of the attribute "receive" in the layer "support" on Wiki-vote dataset. In contrast, the curve of $R(t)$ obtained without $\mathcal{L}_{behavior}$ indicates unstable and over-reactive updates of $R(t)$. This manifests the contribution of $\mathcal{L}_{behavior}$ to capturing emergent behaviors. On the Superuser and Twitter datasets, the red curves exhibit sharper peaks coinciding with the timestamps where emergent behaviors occur. In contrast, the green curves remain relatively bounded and fail to reflect these sudden changes.

## 6 CONCLUSION

To address the structural-semantic complexity, temporal dynamics, inter-layer dependencies, and emergent behavioral phenomena inherent in real-world systems, this paper presents MulDyDiff, the first diffusion-based framework for synthesizing multiplex dynamic attributed networks. MulDyDiff introduces a unified denoising architecture that consists of attribute-aware dynamic transition-based denoising, cross-layer correlation-aware denoising, and behavior-aware guidance. These components capture not only local structural and attribute fidelity but also network-level phenomena such as explosive synchronization and hysteresis, jointly optimized through the proposed BACT loss. Experimental results demonstrate that MulDyDiff consistently surpasses state-of-the-art dynamic graph generators, achieving a 6%-9% improvement over the second-best competitor in terms of dynamic evaluation metrics.

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

## A    EXPLANATION OF THE ILLUSTRATIVE EXAMPLE

Figure 1 illustrates an example of a multiplex dynamic attributed network, where the two layers represent Instagram and Threads, respectively. The network exhibits both *structural and attribute dynamics*, with different node colors indicating user engagement levels (blue for low, red for high). At $\tau = 0$, $v_1$ maintains moderate engagement on Instagram despite being surrounded by inactive peers on the same platform. This deviation from traditional peer influence is attributed to *cross-layer influence* from Threads, where both $v_1$ and its neighbors are more active, offering additional social reinforcement. At $\tau = t$, a new link forms between $v_2$ and $v_3$ on Instagram, driven by shared followees $v_1$ and $v_5$ at $\tau = t - 1$, exemplifying *temporal dependency*. Furthermore, at $\tau = t - 1$, most users on Instagram exhibit low engagement. However, due to frequent interactions with highly engaged users on Threads, these cross-layer influences collectively trigger a sudden shift (similar to the behavioral transition from passive observation to aggressive buying exhibited by Robinhood users during the GameStop short squeeze). At $\tau = t$, Instagram users abruptly increase their engagement, demonstrating a *network-level phenomenon of explosive synchronization* unique to multiplex networks.

## B    MORE RELATED WORK COMPARISONS

### B.1    RELATED WORK COMPARISON TABLE

In this section, we compare our study with the related studies to indicate their differences, which are also summarized in Tables 2 and 3. The former summarizes their application scenarios; the latter compares whether they consider the factors such as structural/attribute changes, temporal dependency and cross-layer dependency or not (O: yes; X: no).

### B.2    STATIC GRAPH GENERATION

Static graph generation methods are mainly statistical or deep generative. Statistical models rely on network statistics (Schweimer et al., 2022), correlations (Erling et al., 2015), community structures (Luo et al., 2020), or degree distributions (Wang et al., 2021), and are often extended for tasks such as frequent pattern mining (Shuai et al., 2013), cross-platform similarity (Shuai et al., 2018), privacy preservation (Ying & Wu, 2009), and scalability (Edunov et al., 2018). Deep models

Table 2: Related work comparison table (part 1).

| Models | Scenario | Architecture |
|---|---|---|
| Ours | Multi-layer temporal graph generation (w/ cross-layer links) | Diffusion and GraphTransformer |
| DAMNETS | Temporal graph generation | GAT |
| TIGGER | Temporal graph generation | GCN and GraphSAGE |
| SGNN-GR (Wang et al., 2022) | Temporal graph generation | GAN and GraphSAGE |
| DYMOND | Temporal graph generation | Algorithm-based |
| AGE (Fan & Huang, 2020) | Temporal graph generation | Self-attention |
| NetTrans (Zhang et al., 2020a) | Network alignment | GCN and encoder-decoder |
| TenGAN (Shiao et al., 2023) | Multi-layer graph generation | GAN and GCN |
| DBGDGM (Campbell et al., 2024) | Multi-layer temporal graph generation (w/o cross-layer links) | Diffusion and GNN |

Table 3: Related work comparison table (part 2).

| Models | Structural change | Attribute change | Temporal dependency | Cross-layer dependency |
|---|---|---|---|---|
| Ours | O | O | O | O |
| DAMNETS | O | X | O | X |
| TIGGER | O | X | O | X |
| SGNN-GR (Wang et al., 2022) | O | X | O | X |
| DYMOND | O | X | O | X |
| AGE (Fan & Huang, 2020) | O | X | O | X |
| NetTrans (Zhang et al., 2020a) | By generalization | O | O | O |
| TenGAN (Shiao et al., 2023) | X | X | X | O |
| DBGDGM (Campbell et al., 2024) | O | X | O | O |

adopt auto-regression (Liao et al., 2019; Shi et al., 2020), variational autoencoders (Guo et al., 2021; Samanta et al., 2020), or GANs (Martinkus et al., 2022), e.g., SPECTRE (Martinkus et al., 2022), which conditions on Laplacian eigenvectors. A few works also study static multiplex graphs (Zhang et al., 2020a; Shiao et al., 2023). However, both statistical and deep models mainly focus on structural information of static single-layer graphs or generate layers independently, neglecting cross-layer correlations and dynamic behaviors such as structural evolution and emergent patterns.

### B.3 MORE COMPARISONS WITH PRIOR TEMPORAL GRAPH GENERATORS

The gap between our work and prior temporal graph generators from the aspects of (1) temporal dependency, (2) multi-layer structure, (3) inductivity, (4) graph-level behavior guidance, (5) global and local evolution. The differences are summarized in Table 4.

1. Temporal dependency: While several existing models indeed condition on only a single snapshot (the immediately preceding snapshot $G_{t-1}$ Campbell et al. (2024); Wang et al. (2022) or $G_{t-\Delta t}$ Gupta et al. (2022)) or merely local motif transition statistics Liu & Sariyüce (2023); Zeno et al. (2021), our contribution is not the mere use of temporal conditioning but the design of an explicitly history-mixing forward diffusion that recursively aggregates all past snapshots $G_{0:t-1}$ when defining the distribution at time $t$. This formulation provides long-range temporal coupling and continuity that extend beyond the typical one-step conditioning scheme. Specifically, Sec. 4.1 introduces an attribute-aware dynamic transition mixture D (Eq. (1)). For $t \geq 1$, the forward process combines the current snapshot with the recursively accumulated context from all previous timestamps, so that the marginal $q(\mathbf{X}_{l,t}^{(s)} \mid \mathbf{X}_{l,0:t}^{(0)})$ (Eq. (4)) depends on the entire trajectory up to $t$. This enables the generation of future

snapshots to exploit long-range temporal signals. In contrast, many existing temporal graph generators specify their forward or conditional distributions using only $G_{t-1}$, and therefore do not couple multiple past snapshots within the forward process.

2. Multi-layer structure: Among the compared methods, most of the compared studies only consider single-layer structure; only DBGDGM in Campbell et al. (2024) handles multiple aspects. In contrast, MulDyDiff models cross-layer correlation during generation, supported by a learned cross-layer coupling graph. MulDyDiff not only captures temporal evolution in a single layer but also co-evolution of each layer influenced by other layers.

3. Inductivity: Most previous works are transductive since they do not consider unseen nodes. MulDyDiff (ours) and TIGGER-I in Gupta et al. (2022) are inductive since the former adopts a permutation-invariant temporal graph transformer architecture, which does not rely on node ID information; the latter builds a multi-mode decoder to learn distributions of node embeddings; and the others are not inductive since they tend to use fixed nodes and cannot generalize to unseen nodes.

4. Graph-level behavior guidance: None of the prior temporal or multiplex generators model system-level phenomena such as explosive synchronization or hysteresis. In contrast, MulDyDiff is the first to introduce behavior-aware guidance to reproduce these global dynamics, enabling to regularize graph-level behavior of generated graphs to be similar to input graphs.

5. Global and local evolution: In contrast to previous works focusing on generating the current snapshot conditioning on only a single snapshot (or merely local motif transition statistics) with only local evolution taken into account, our model enables future snapshot generation considering both local and global evolution from given historical snapshots.

Table 4: Comparison table between MulDyDiff and prior temporal graph generators.

| Models | Temporal dependency | Multi-layer structure | Inductive | Graph-level behavior guidance | Global evolution | Local evolution |
|---|---|---|---|---|---|---|
| MulDyDiff (ours) | $p(G_t \mid G_{0:t-1})$ | O (with cross-layer dependency) | O | O | O | O |
| DBGDGM Campbell et al. (2024) | $p(G_t \mid G_{t-1})$ | O (without cross-layer dependency) | X | X | X | O |
| MTM Liu & Sariyüce (2023) | local motif transition statistics | X | X | X | X | O |
| SGNN Wang et al. (2022) | $p(G_t \mid G_{t-1})$ | X | X | X | X | O |
| TIGGER Gupta et al. (2022) | $p(G_t \mid G_{t-\Delta t})$ | X | O | X | X | O |
| DYMOND Zeno et al. (2021) | local motif transition statistics | X | X | X | X | O |

## C  NOTATION TABLE

The notations in this paper are listed in Table 5.

## D  PRELIMINARY: DISCRETE DENOISING DIFFUSION PROBABILISTIC MODEL

We introduce the background of denoising diffusion probabilistic models (DDPM). Typically, a DDPM consists of two components: the forward noising process and the reverse denoising process. For the diffusion step $s \geq 1$, the forward noising process of a DDPM for a graph $G^{(0)}$ is defined by $q(G^{(s)}|G^{(s-1)})$ and $q(G^{(S)}|G^{(0)}) = \prod_{s=1}^{S} q(G^{(s)}|G^{(s-1)})$, where $S$ is the maximum diffusion step.

Table 5: Notation table

| Notation | Description |
|---|---|
| $s = 0, 1, \ldots, S$ | diffusion steps |
| $t = 0, 1, \ldots, T$ | timestamps |
| $l = 1, \ldots, L$ | layers |
| $\Gamma$ | Multi-layer graph sequence $\{\mathbb{G}_0, \ldots, \mathbb{G}_T\}$ |
| $\mathbb{G}_t$ | $L$-layer graph at timestamp $t$ |
| $G_t^{(I)}$ | Intra-layer graph $(\{(\mathbf{X}_{l,t}, \mathbf{E}_{l,t})\}_{l=1}^L)$ |
| $\mathbb{G}_{0:t}$ | $L$-layer graph sequence from timestamp 0 to $t$ |
| $\mathbf{X}_{l,t}^{(s)} \in \mathbb{R}^{a \times N}, \mathbf{E}_{l,t} \in \mathbb{R}^{b \times N \times N}$ | Diffused node/edge representation of $N$ nodes at step $s$ with $a$ node types and $b$ edge types of layer $l$ in $\mathbb{G}_t$ |
| $G_{l,(0:T)}$ | intra-layer graph sequence $\{G_{l,t} = (\mathbf{X}_{l,t}, \mathbf{E}_{l,t})\}_{t=0}^T$ for layer $l$ |
| $G_t^{(B)}$ | inter-layer bipartite graph $(\{\mathbf{X}_{l,t}, \mathbf{X}_{t,m}\}, \{\mathbf{B}_{(l,m),t}\}_{l \neq m})$ |
| $\mathbf{B}_{(l,m),t}$ | inter-layer edge representation between layers $l$ and $m$ in $\mathbb{G}_t$ |
| $G_{(l,m),(0:T)}$ | intra-layer graph sequence $(\{\mathbf{X}_{l,t}, \mathbf{X}_{m,t}\}_{t=0}^T, \{\mathbf{B}_{(l,m),t}\}_{t=0}^T)$ for layers $l$ and $m$ |
| $\overline{\mathbf{Q}}_{\mathbf{X}_{l,(0:t-1)}}^{(s)}$ | multi-step Markov transition matrix that transits $\mathbf{X}_{l,t}^{(0)}$ to $\mathbf{X}_{l,t}^{(s)}$ |
| $\mathbf{Q}_{\mathbf{X}_{l,(0:t-1)}}^{(s)}$ | single-step Markov transition matrix that transits $\mathbf{X}_{l,t}^{(s-1)}$ to $\mathbf{X}_{l,t}^{(s)}$ |
| $G_t^{(C)}$ | cross-layer coupling graph $(V_t^{(C)}, E_t^{(C)})$ |
| $V_t^{(C)}$ | nodes representing layer IDs $\{1, \ldots, L\}$ |
| $E_t^{(C)}$ | edges representing link existence between distinct layers $\{(l,m) \mid l, m \in V_t^{(C)}, \mathbf{B}_{(l,m),t} \neq \mathbf{0}\}$ |
| $\alpha_s$ | a parameter that controls noise strength, defining how fast information is washed out by noise along the diffusion axis $s$; $\overline{\alpha}_s = \prod_{i=1}^s \alpha_i$ |
| $\gamma_s$ | a parameter that controls the dependence on previous snapshots, specifying how strong the temporal smoothing is along the time axis $t$; $\overline{\gamma}_s = \prod_{i=1}^s \gamma_i$ |
| $\mathrm{D}(\cdot, \overline{\gamma}_s)$ | temporal transition-aware mixture with hyper-parameter $\overline{\gamma}_s$ |
| $q$ | diffusion process |
| $p_\theta$ | reverse denoising process |
| $\phi_\theta$ | dynamic transition denoising network |
| $\hat{p}_{l,t}^{(X)}, \hat{p}_{l,t}^{(E)}, \hat{p}_{(l,m),t}^{(B)}$ | denoising distributions learned from dynamic transition denoising network $\phi_\theta$ |
| $\phi_\theta^{(C)}$ | cross-layer correlation-aware dynamic transition denoising network |
| $\hat{p}_{C,(l,t)}^{(X)}, \hat{p}_{C,(l,t)}^{(E)}, \hat{p}_{C,(l,t)}^{(B)}$ | denoising distributions learned from cross-layer correlation-aware dynamic transition denoising network $\phi_\theta^{(C)}$ |
| $p_t^{(C)}$ | distribution learned to predict cross-layer correlations at $t$ in $G_t^{(C)}$ according to $\mathbb{G}_{0:t}$ |
| $p_t^{(I)}$ | distribution learned to predict intra-layer structure at $t$ according to $\mathbb{G}_{0:t}$ |

Given $G^{(0)} = (\mathbf{X}^{(0)}, \mathbf{E}^{(0)})$, the standard categorical forward process of a node attribute representation is:

$$\mathbf{X}^{(s)} = \overline{\alpha}_s \mathbf{X}^{(0)} + (1 - \overline{\alpha}_s)\frac{\overline{\mathbf{1}}_a}{a}, \quad \mathbf{E}^{(s)} = \overline{\alpha}_s \mathbf{E}^{(0)} + (1 - \overline{\alpha}_s)\frac{\overline{\mathbf{1}}_a}{a}, \tag{21}$$

with $\overline{\alpha}_s$ controlling noise strength.

For the reverse denoising process, given $G^{(s)}$, a denoising neural network $\phi_\theta$ (parameterized by $\theta$) is designed to predict the denoised graph $G^{(s-1)}$, deriving the reverse denoising process $p_\theta$ as follows:

$$p_\theta(G^{(s-1)}|G^{(s)}) = q(G^{(s-1)}|G^{(s)}, G^{(0)})p_\theta(G^{(0)}|G^{(s)});$$

$$q(G^{(s-1)}|G^{(s)}) \propto q(G^{(s)}|G^{(s-1)}, G^{(0)})q(G^{(s-1)}|G^{(0)})$$

$$= q(G^{(s)}|G^{(s-1)})q(G^{(s-1)}|G^{(0)}),$$

where $q(G^{(s-1)}|G^{(s)})$ can be approximated by the noising process.

# E   DETAILED DERIVATIONS AND PROOFS OF ATTRIBUTE-AWARE DYNAMIC TRANSLATION-BASED DENOISING

## E.1   INTUITION OF CAPTURING LONG-RANGE TEMPORAL COUPLING AND CONTINUITY.

Unlike standard temporal models that rely on one-step Markov dependencies (conditioning only on $G_{t-1}$), our forward process explicitly incorporates the entire history $G_{0:t-1}$ when defining the distribution at time $t$. This is achieved through the attribute-aware dynamic transition mixture D in Eq. (1):

$$\mathrm{D}(\mathbf{X}_{l,(0:t)}^{(0)}, \bar{\gamma}_s) = \bar{\gamma}_s \mathbf{X}_{l,t}^{(0)} + (1 - \bar{\gamma}_s)\mathrm{D}(\mathbf{X}_{l,(0:t-1)}^{(0)}, \bar{\gamma}_s),$$

which recursively accumulates clean snapshots from all previous timestamps. As a result, the forward prior for $\mathbf{X}_{l,t}^{(s)}$ is not a local variation of $\mathbf{X}_{l,t-1}^{(0)}$ but rather a history-mixed representation that embeds long-range temporal signals. This design has two key benefits: i) it enables the reverse denoising network to perform history-guided denoising, capturing persistent temporal structures and long-range dependencies that one-step models miss; and ii) since the mixture is formed from clean states $\mathbf{X}^{(0)}$, it mitigates the error propagation issue of autoregressive temporal generators that repeatedly condition on noisy predictions.

## E.2   DERIVATION LOGIC OF CAPTURING LONG-RANGE TEMPORAL COUPLING AND CONTINUITY.

We aim to capture long-range temporal coupling and continuity in temporal graph sequences, which is achieved by deriving the temporal-aware diffusion model with the following logic flow:

(1) We first define a temporal aggregation function (Eq. (1)) that summarizes all past clean snapshots using an exponentially weighted mixture. This establishes how temporal information from earlier timestamps is incorporated into the model.

(2) We then inject noise in a temporally consistent way (Eqs. (2) and (3)) by blending local diffusion at time $t$ with the diffused representation at time $t - 1$. This step defines how attribute noise interacts with temporal smoothness.

(3) We show that this forward process admits a closed-form expression (Eq. (4)), which explicitly reveals the influence of the entire history.

(4) We reinterpret the closed form of the forward process as a Markov transition (Eq. (5)), clarifying how each diffusion step decomposes into self-preservation, history-driven drift, and uniform noise injection.

(5) We decompose the multi-step transition into single-step transitions (Eqs. (6) and (7)) to make posterior inference tractable.

(6) We derive the exact posterior for reverse diffusion (Eq. (9)), enabling us to compute the probability of the previous noisy state conditioned on the current one.

(7) Finally, we approximate the reverse process with a learned denoiser (Eq. (10)), which maps noisy states back to clean states in a history-aware manner.

Together, these steps establish a temporally coherent forward diffusion process, whose reverse process can reconstruct each snapshot using the entire clean history, enabling the model to capture long-range temporal dependencies rather than purely local transitions.

### E.3 DETAILED DERIVATION OF THE TEMPORAL DIFFUSION PROCESS

#### E.3.1 RECURSIVE EXPANSION ALONG THE TEMPORAL DIMENSION

To incorporate information from all historical snapshots up to time $t$, we define a recursive temporal aggregation function. The idea is to let the most recent snapshot contribute most strongly, while earlier snapshots contribute with exponentially decaying weights. Given historical snapshots $\mathbf{X}_{l,(0:t-1)}^{(0)}$ and the current snapshot $\mathbf{X}_{l,t}^{(0)}$, we define the dynamic transition in Eq. (1) by combining the current snapshot at timestamp $t$ and the dynamic transition over past snapshots from timestamp $0$ to $t-1$ as follows:

$$\mathrm{D}(\mathbf{X}_{l,(0:t)}^{(0)}, \overline{\gamma}_s) = \overline{\gamma}_s \mathbf{X}_{l,t}^{(0)} + (1 - \overline{\gamma}_s)\mathrm{D}(\mathbf{X}_{l,(0:t-1)}^{(0)}, \overline{\gamma}_s),$$

with the initial condition $\mathrm{D}(\mathbf{X}_{l,0}^{(0)}, \overline{\gamma}_s) = \mathbf{X}_{l,0}^{(0)}$. This recursion D means that each historical snapshot influences the aggregated representation, but with strength controlled by $\overline{\gamma}_s$. A larger $\overline{\gamma}_s$ prioritizes the current snapshot, while a smaller one increases the influence of earlier snapshots.

#### E.3.2 EXTENSION TO TEMPORAL DIFFUSION PROCESS

In the traditional diffusion process, the diffused snapshot $\mathbf{Y}_{l,t}^{(s)}$ in the $s$-th step of the snapshot $\mathbf{X}_{l,t}^{(0)}$ is $\mathbf{Y}_{l,t}^{(s)} = \overline{\alpha}_s \mathbf{X}_{l,t}^{(0)} + (1 - \overline{\alpha}_s)\frac{\overline{\mathbf{1}}_a}{a}$ at timestamp $t$. To generate temporally consistent noisy representations $\mathbf{X}_{l,t}^{(s)}$, we blend the locally diffused state $\mathbf{Y}_{l,t}^{(s)}$ at time $t$ with the diffused state $\mathbf{X}_{l,t-1}^{(s)}$ from the previous timestamp $t-1$, which ensures smoothness across time. Thus, we define the dynamic transition-aware forward process in Eq. (2):

$$\mathbf{X}_{l,t}^{(s)} = \overline{\gamma}_s \mathbf{Y}_{l,t}^{(s)} + (1 - \overline{\gamma}_s)\mathbf{X}_{l,t-1}^{(s)}$$
$$= \overline{\gamma}_s(\overline{\alpha}_s \mathbf{X}_{l,t}^{(0)} + (1 - \overline{\alpha}_s)\frac{\overline{\mathbf{1}}_a}{a}) + (1 - \overline{\gamma}_s)\mathbf{X}_{l,t-1}^{(s)}, \forall s, t \geq 1,$$

where the first term in Eq. (2) adds noise to the current snapshot, while the second term propagates temporal influence forward from $t-1$. The parameter $\overline{\gamma}_s$ adjusts the balance: larger values emphasize the current snapshot; smaller values enforce stronger temporal continuity. Since $t = 0$ is the starting point, we initialize its forward diffusion to be the same as the traditional diffusion process in Eq. (3).

$$\mathbf{X}_{l,0}^{(s)} = \mathbf{Y}_{l,0}^{(s)} = \overline{\alpha}_s \mathbf{X}_{l,0}^{(0)} + (1 - \overline{\alpha}_s)\frac{\overline{\mathbf{1}}_a}{a},$$

where $\overline{\mathbf{1}}_a \in \mathbb{R}^{a \times N}$ denotes an all-one matrix. The weight $\overline{\alpha}_s$ determines how much of the original structure is retained.

#### E.3.3 CLOSED-FORM EXPRESSION OF THE FORWARD PROCESS

By expanding the recursive temporal-aware forward equations (Eq. (2) with induction on $t$), we obtain a direct relationship between the $s$-step noisy snapshot and all historical clean snapshots, explicitly incorporating the entire history $\mathbf{X}_{l,(0:t)}^{(0)}$ in Eq. (4) as follows:

$$\mathbf{X}_{l,t}^{(s)} = \overline{\alpha}_s \mathrm{D}(\mathbf{X}_{l,(0:t)}^{(0)}, \overline{\gamma}_s) + (1 - \overline{\alpha}_s)\frac{\overline{\mathbf{1}}_a}{a}.$$

Thus, the noisy snapshot is a mixture of a history-aggregated signal and a uniform noise baseline. As $\overline{\alpha}_s$ decreases, the influence of the uniform noise grows, gradually removing temporal structure.

#### E.3.4 MULTI-STEP MARKOV TRANSITION

We express Eq. (4) as a Markov transition that changes the state of $\mathbf{X}_{l,t}^{(s)}$, through Eq. (1)

$$\mathbf{X}_{l,t}^{(s)} = \overline{\alpha}_s\overline{\gamma}_s\mathbf{X}_{l,t}^{(0)} + \overline{\alpha}_s(1 - \overline{\gamma}_s)\mathrm{D}(\mathbf{X}_{l,(0:t-1)}^{(0)}, \overline{\gamma}_s) + (1 - \overline{\alpha}_s)\frac{\overline{\mathbf{1}}_a}{a} \quad \text{(By Eq. (1))},$$

$$= (\overline{\alpha}_s\overline{\gamma}_s\mathbf{I} + \overline{\alpha}_s(1 - \overline{\gamma}_s)\mathrm{D}(\mathbf{X}_{l,(0:t-1)}^{(0)}, \overline{\gamma}_s)\overline{\mathbf{1}}_a^\top + (1 - \overline{\alpha}_s)\overline{\mathbf{1}}_a\frac{\overline{\mathbf{1}}_a^\top}{a})\mathbf{X}_{l,t}^{(0)}$$

$$(\because \overline{\mathbf{1}}_a\overline{\mathbf{1}}_a^\top\mathbf{X}_{l,t}^{(0)} = \overline{\mathbf{1}}_a, \mathrm{D}(\mathbf{X}_{l,(0:t-1)}^{(0)}, \overline{\gamma}_s)\overline{\mathbf{1}}_a^\top\mathbf{X}_{l,t}^{(0)} = \mathrm{D}(\mathbf{X}_{l,(0:t-1)}^{(0)}, \overline{\gamma}_s))$$

$$= (\overline{\alpha}_s\overline{\gamma}_s\mathbf{I} + \overline{\alpha}_s(1 - \overline{\gamma}_s)\overline{\mathbf{1}}_a\mathrm{D}(\mathbf{X}_{l,(0:t-1)}^{(0)}, \overline{\gamma}_s)^\top + (1 - \overline{\alpha}_s)\overline{\mathbf{1}}_a\frac{\overline{\mathbf{1}}_a^\top}{a})^\top\mathbf{X}_{l,t}^{(0)} \quad \text{(Transpose)}$$

$$= \overline{\mathbf{Q}}_{\mathbf{X}_{l,(0:t-1)}}^{(s)}{}^\top\mathbf{X}_{l,t}^{(0)},$$

where $\overline{\mathbf{Q}}^{(s)}_{\mathbf{X}_{l,(0:t-1)}} = \overline{\alpha}_s\overline{\gamma}_s\mathbf{I}+\overline{\alpha}_s(1-\overline{\gamma}_s)\mathbf{1}_a\mathrm{D}(\mathbf{X}^{(0)}_{l,(0:t-1)},\overline{\gamma}_s)^\top+(1-\overline{\alpha}_s)\mathbf{1}_a\frac{\overline{\mathbf{1}}_a^\top}{a}$ is a Markov transition matrix decomposing the diffusion into three intuitive effects: (1) self-preservation (i.e., staying in the same state), (2) drifting toward the aggregated history $D(\cdot)$, and (3) injecting uniform noise.

Thus, we rewrite Eq. (4) using the Markov transition matrix $\overline{\mathbf{Q}}^{(s)}_{\mathbf{X}_{l,(0:t-1)}}$ as in Eq. (5):

$$q(\mathbf{X}^{(s)}_{l,t}|\mathbf{X}^{(0)}_{l,(0:t)}) = Cat(\mathbf{X}^{(s)}_{l,t}; \overline{\mathbf{Q}}^{(s)}_{\mathbf{X}_{l,(0:t-1)}}{}^\top \mathbf{X}^{(0)}_{l,t}) = \mathbf{X}^{(s)}_{l,t}{}^\top \overline{\mathbf{Q}}^{(s)}_{\mathbf{X}_{l,(0:t-1)}}{}^\top \mathbf{X}^{(0)}_{l,t}.$$

### E.3.5 SINGLE-STEP MARKOV TRANSITION

To derive the posterior and the reverse process, we rewrite the multi-step update as a single-step Markov transition. By subtracting $\alpha_s\gamma_s\mathbf{X}^{(s-1)}_{l,t}$ from $\mathbf{X}^{(s)}_{l,t}$ (using Eq. (4) to cancel out $\mathbf{X}^{(0)}_{l,t}$), we obtain the single-step forward process in Eq. (6):

$$\mathbf{X}^{(s)}_{l,t} = \alpha_s\gamma_s\mathbf{X}^{(s-1)}_{l,t} + \overline{\alpha}_s[(1-\overline{\gamma}_s)\mathrm{D}(\mathbf{X}^{(0)}_{l,(0:t-1)},\overline{\gamma}_s)$$
$$- (\gamma_s - \overline{\gamma}_s)\mathrm{D}(\mathbf{X}^{(0)}_{l,(0:t-1)},\overline{\gamma}_{s-1})] + [1 - \alpha_s\gamma_s - \overline{\alpha}_s(1-\gamma_s)]\frac{\overline{\mathbf{1}}_a}{a},$$

in which the three parts correspond to: (1) keeping part of the previous noisy state via $\alpha_s\gamma_s\mathbf{X}^{(s-1)}_{l,t}$, (2) adjusting toward the history-consistent direction implied by the multi-step dynamics (the two $\mathrm{D}(\cdot)$ terms ensure that the single-step behavior matches the $s$-step closed form), and (3) injecting uniform noise to maintain stochasticity and preserve a valid categorical distribution. This decomposition reveals how the model preserves previous noise, incorporates temporal structure, and adds randomness.

Similar to Eq. (5), we have Eq. (7) as follows:

$$q(\mathbf{X}^{(s)}_{l,t}|\mathbf{X}^{(s-1)}_{l,t},\mathbf{X}^{(0)}_{l,(0:t)}) = Cat(\mathbf{X}^{(s)}_{l,t}; \mathbf{Q}^{(s)}_{\mathbf{X}_{l,(0:t-1)}}{}^\top \mathbf{X}^{(s-1)}_{l,t}) = \mathbf{X}^{(s)}_{l,t}{}^\top \mathbf{Q}^{(s)}_{\mathbf{X}_{l,(0:t-1)}}{}^\top \mathbf{X}^{(s-1)}_{l,t},$$

where $\mathbf{Q}^{(s)}_{\mathbf{X}_{l,(0:t-1)}} = \overline{\alpha}_s\overline{\gamma}_s\mathbf{I}+\overline{\alpha}_s(1-\overline{\gamma}_s)\mathbf{1}_a\mathrm{D}(\mathbf{X}^{(0)}_{l,(0:t-1)},\overline{\gamma}_s)^\top+(1-\overline{\alpha}_s)\mathbf{1}_a\frac{\overline{\mathbf{1}}_a^\top}{a}$ is a Markov transition matrix encoding: (1) self-preservation, (2) a shift toward the aggregated history $D(\cdot)$, and (3) movement toward the uniform noise baseline.

It is worth noting that $\mathbf{Q}^{(s)}_{\mathbf{X}_{l,(0:t-1)}}$ is a transition matrix satisfying the property of a Markov chain, i.e., $\overline{\mathbf{Q}}^{(s-1)}_{\mathbf{X}_{l,(0:t-1)}}\mathbf{Q}^{(s)}_{\mathbf{X}_{l,(0:t-1)}} = \overline{\mathbf{Q}}^{(s)}_{\mathbf{X}_{l,(0:t-1)}}$. Thus, the distribution $q(\mathbf{X}^{(s)}_{l,t}|\mathbf{X}^{(0)}_{l,(0:t)})$ can be marginalized by $q(\mathbf{X}^{(s)}_{l,t}|\mathbf{X}^{(s-1)}_{l,t}\mathbf{X}^{(0)}_{l,(0:t)})$ and $q(\mathbf{X}^{(s-1)}_{l,t}|\mathbf{X}^{(0)}_{l,(0:t)})$ as follows:

$$q(\mathbf{X}^{(s)}_{l,t}|\mathbf{X}^{(0)}_{l,(0:t)})$$
$$= \sum_{\mathbf{X}^{(s-1)}_{l,t}} q(\mathbf{X}^{(s)}_{l,t},\mathbf{X}^{(s-1)}_{l,t}|\mathbf{X}^{(0)}_{l,(0:t)}) \quad \text{(Marginalization)}$$
$$= \sum_{\mathbf{X}^{(s-1)}_{l,t}} q(\mathbf{X}^{(s)}_{l,t}|\mathbf{X}^{(s-1)}_{l,t},\mathbf{X}^{(0)}_{l,(0:t)})q(\mathbf{X}^{(s-1)}_{l,t}|\mathbf{X}^{(0)}_{l,(0:t)}) \quad \text{(By Bayesian formula)}$$
$$= \sum_{\mathbf{X}^{(s-1)}_{l,t}} (\mathbf{X}^{(s)}_{l,t}{}^\top \mathbf{Q}^{(s)}_{\mathbf{X}_{l,(0:t-1)}}{}^\top \mathbf{X}^{(s-1)}_{l,t})(\mathbf{X}^{(s-1)}_{l,t}{}^\top \overline{\mathbf{Q}}^{(s-1)}_{\mathbf{X}_{l,(0:t-1)}}{}^\top \mathbf{X}^{(0)}_{l,t}) \quad \text{(By Eqs. (5) and (7))}$$
$$= \mathbf{X}^{(s)}_{l,t}{}^\top \mathbf{Q}^{(s)}_{\mathbf{X}_{l,(0:t-1)}}{}^\top \sum_{\mathbf{X}^{(s-1)}_{l,t}} (\mathbf{X}^{(s-1)}_{l,t}\mathbf{X}^{(s-1)}_{l,t}{}^\top)\overline{\mathbf{Q}}^{(s-1)}_{\mathbf{X}_{l,(0:t-1)}}{}^\top \mathbf{X}^{(0)}_{l,t}$$
$$= \mathbf{X}^{(s)}_{l,t}{}^\top \mathbf{Q}^{(s)}_{\mathbf{X}_{l,(0:t-1)}}{}^\top \overline{\mathbf{Q}}^{(s-1)}_{\mathbf{X}_{l,(0:t-1)}}{}^\top \mathbf{X}^{(0)}_{l,t} \quad (\because \sum_{\mathbf{X}^{(s-1)}_{l,t}} (\mathbf{X}^{(s-1)}_{l,t}\mathbf{X}^{(s-1)}_{l,t}{}^\top) = \mathbf{I})$$
$$= \mathbf{X}^{(s)}_{l,t}{}^\top \overline{\mathbf{Q}}^{(s)}_{\mathbf{X}_{l,(0:t-1)}}{}^\top \mathbf{X}^{(0)}_{l,t} \quad (\because \overline{\mathbf{Q}}^{(s-1)}_{\mathbf{X}_{l,(0:t-1)}}\mathbf{Q}^{(s)}_{\mathbf{X}_{l,(0:t-1)}} = \overline{\mathbf{Q}}^{(s)}_{\mathbf{X}_{l,(0:t-1)}}),$$

which proves Theorem 4.1.

### E.3.6 POSTERIOR DISTRIBUTION

To reverse the diffusion, we use Bayes' theorem to compute the posterior distribution over the previous noisy state given the current noisy state and the clean history. From Eqs. (5) and (7), we

derive the posterior of the forward process $q$ in Eq. (9) as follows:

$$q(\mathbf{X}_{l,t}^{(s-1)}|\mathbf{X}_{l,t}^{(s)}, \mathbf{X}_{l,(0:t)}^{(0)}) = \frac{q(\mathbf{X}_{l,t}^{(s)}|\mathbf{X}_{l,t}^{(s-1)}, \mathbf{X}_{l,(0:t)}^{(0)})q(\mathbf{X}_{l,t}^{(s-1)}|\mathbf{X}_{l,(0:t)}^{(0)})}{q(\mathbf{X}_{l,t}^{(s)}|\mathbf{X}_{l,(0:t)}^{(0)})} \quad \text{(By Bayesian formula)}$$

$$= \frac{(\mathbf{X}_{l,t}^{(s)\top}\mathbf{Q}_{\mathbf{X}_{l,(0:t-1)}}^{(s)\top}\mathbf{X}_{l,t}^{(s-1)})(\mathbf{X}_{l,t}^{(s-1)\top}\overline{\mathbf{Q}}_{\mathbf{X}_{l,(0:t-1)}}^{(s-1)\top}\mathbf{X}_{l,t}^{(0)})}{(\mathbf{X}_{l,t}^{(s)\top}\overline{\mathbf{Q}}_{\mathbf{X}_{l,(0:t-1)}}^{(s)\top}\mathbf{X}_{l,t}^{(0)})} \quad \text{(By Eqs. (5) and (7))}$$

$$= \frac{(\mathbf{X}_{l,t}^{(s-1)\top}\mathbf{Q}_{\mathbf{X}_{l,(0:t-1)}}^{(s)}\mathbf{X}_{l,t}^{(s)})(\mathbf{X}_{l,t}^{(s-1)\top}\overline{\mathbf{Q}}_{\mathbf{X}_{l,(0:t-1)}}^{(s-1)\top}\mathbf{X}_{l,t}^{(0)})}{(\mathbf{X}_{l,t}^{(s)\top}\overline{\mathbf{Q}}_{\mathbf{X}_{l,(0:t-1)}}^{(s)\top}\mathbf{X}_{l,t}^{(0)})}$$

(Transpose the first term in the numerator)

$$= \mathbf{X}_{l,t}^{(s-1)\top}\frac{\mathbf{Q}_{\mathbf{X}_{l,(0:t-1)}}^{(s)}\mathbf{X}_{l,t}^{(s)}\odot\overline{\mathbf{Q}}_{\mathbf{X}_{l,(0:t-1)}}^{(s-1)\top}\mathbf{X}_{l,t}^{(0)}}{\mathbf{X}_{l,t}^{(s)\top}\overline{\mathbf{Q}}_{\mathbf{X}_{l,(0:t-1)}}^{(s)}\mathbf{X}_{l,t}^{(0)}},$$

which proves Theorem 4.2 (the closed form of the posterior). The numerator multiplies: (a) how likely each prior state leads to $\mathbf{X}^{(s)}$, and (b) how likely it is under the multi-step history-informed prior. The denominator normalizes these weights into a valid categorical distribution.

### E.3.7 REVERSE PROCESS

Since the true clean snapshot is unknown, we approximate the reverse transition by combining the exact posterior with a learned clean-state predictor. The reverse denoising process is approximated by the posterior (Eq. (9)) in Eq. (10) through marginalization as follows:

$$p_\theta(\mathbf{X}_{l,t}^{(s-1)}|\mathbf{X}_{l,t}^{(s)}, \mathbf{X}_{l,(0:t-1)}^{(0)})$$

$$= \sum_{\mathbf{X}_{l,t}^{(0)}} p_\theta(\mathbf{X}_{l,t}^{(s-1)}, \mathbf{X}_{l,t}^{(0)}|\mathbf{X}_{l,t}^{(s)}, \mathbf{X}_{l,(0:t-1)}^{(0)}) \quad \text{(Marginalization)}$$

$$= \sum_{\mathbf{X}_{l,t}^{(0)}} p_\theta(\mathbf{X}_{l,t}^{(s-1)}|\mathbf{X}_{l,t}^{(s)}, \mathbf{X}_{l,(0:t-1)}^{(0)}, \mathbf{X}_{l,t}^{(0)})p_\theta(\mathbf{X}_{l,t}^{(0)}|\mathbf{X}_{l,t}^{(s)}, \mathbf{X}_{l,(0:t-1)}^{(0)}) \quad \text{(By Bayesian formula)}$$

$$\approx \sum_{\mathbf{X}_{l,t}^{(0)}} q(\mathbf{X}_{l,t}^{(s-1)}|\mathbf{X}_{l,t}^{(s)}, \mathbf{X}_{l,(0:t-1)}^{(0)}, \mathbf{X}_{l,t}^{(0)})\hat{p}_{l,t}^{(X)}(\mathbf{X}_{l,t}^{(0)}|\mathbf{X}_{l,t}^{(s)}, \mathbf{X}_{l,(0:t-1)}^{(0)}),$$

where $\hat{p}_{l,t}^{(X)}$ is learned by a denoising network (parameterized with $\theta$) that denoises $\mathbf{X}_{l,t}^{(s)}$ to the clean representation $\mathbf{X}_{l,t}^{(0)}$ conditioned on the given historical information $\mathbf{X}_{l,(0:t-1)}^{(0)}$, which captures long-range temporal coupling and coherence. The denoiser predicts plausible clean states, and the model averages the exact backward transitions over those predictions, yielding an effective reverse diffusion step.

## F DETAILED DERIVATIONS AND PROOFS OF CROSS-LAYER CORRELATION-AWARE DENOISING

### F.1 INTUITION OF CAPTURING IMPLICIT CO-EVOLUTION VIA ATTENTION

Our model captures both explicit structural coupling and implicit co-evolution. We do not rely solely on static, observed inter-layer edges; instead, we leverage attention to capture latent correlations where layers evolve in synchrony without direct connections. This is achieved in the cross-layer correlation-aware denoising module (Eqs. (15) and (16)), specifically through the predicted coupling graph $\hat{G}_t^{(C)}$, which dynamically estimates cross-layer correlation strengths at each timestamp. The denoising network then uses these correlations through a cross-attention mechanism, assigning weights to other layers based on their state similarity and temporal co-evolution—even when no explicit inter-layer edge $B_{(l,m)}$ exists (detailed in Appendix G). For example, if nodes in layer $A$ and nodes in layer $B$ repeatedly undergo similar attribute transitions or community-level changes at the same times (a "shared temporal shock"), the attention mechanism can learn this pattern and allow layer $A$ to guide the reconstruction of layer $B$, and vise versa, despite the absence of direct inter-layer links between specific node pairs.

## F.2 DERIVATION LOGIC OF CAPTURING IMPLICIT CO-EVOLUTION VIA ATTENTION

We aim to capture explicit structural coupling and implicit co-evolution in multi-layer temporal graph sequences, which is achieved by multiplying the temporal-aware forward and reverse processes of all layers with the following logic flow:

(1) The multi-step (single-step) forward process of a multi-layer temporal graph sequence is extended to Eq. (11) from Eq. (5) (Eq. (7)) by multiplying the multi-step (single-step) forward processes of all layers, which are assumed to be independent of one another.

(2) The posterior of the forward process of a multi-layer temporal graph sequence is extended to Eq. (12) from Eq. (9) by multiplying the posteriors of all layers with the Bayesian formula using Eq. (11).

(3) The denoising distribution is derived in Eq. (13). It is the product of the denoising distributions of all layers, as shown in Eq. (14). These layer-wise distributions are learned separately and are conditionally independent given the clean snapshots from timestamps 0 to $t-1$. Each layer's distribution is learned through a cross-layer attention mechanism. The attention weights are learned from the cross-layer coupling graph predicted by a cross-layer predictor.

(4) The reverse process of a multi-layer temporal graph sequence is extended to Eq. (15) by multiplying the reverse processes of all layers (Eq. (16)), which are conditionally independent given the clean snapshots $\mathbf{X}^{(0)}_{(1:L),(0:t-1)}$ during the preceding timestamps 0 to $t-1$.

## F.3 DERIVATION DETAILS OF CAPTURING IMPLICIT CO-EVOLUTION VIA ATTENTION

### F.3.1 FORWARD DIFFUSION PROCESS

We assume that the noise-injection process at diffusion step $s$ is independent across layers. Intuitively, each layer is corrupted by noise separately, while the reverse denoising process later leverages cross-layer attention to reintroduce interdependencies. The multi-step and single-step forward processes of a multi-layer temporal graph sequence are extended from Eqs. (5) and (7) to Eq. (11) as follows:

$$q(\mathbf{X}^{(s)}_{(1:L),t}|\mathbf{X}^{(0)}_{(1:L),(0:t)}) = \prod_{l=1}^{L} q(\mathbf{X}^{(s)}_{l,t}|\mathbf{X}^{(0)}_{l,(0:t)});$$

$$q(\mathbf{X}^{(s-1)}_{(1:L),t}|\mathbf{X}^{(s)}_{(1:L),t},\mathbf{X}^{(0)}_{(1:L),(0:t)}) = \prod_{l=1}^{L} q(\mathbf{X}^{(s)}_{l,t}|\mathbf{X}^{(s-1)}_{l,t},\mathbf{X}^{(0)}_{l,(0:t)}).$$

### F.3.2 POSTERIOR

Due to the independence of the forward processes of all layers, the posterior of the forward process of a multi-layer temporal graph sequence is extended from Eq. (9) to Eq. (12) as follows:

$$
\begin{aligned}
q(\mathbf{X}^{(s-1)}_{(1:L),t}|\mathbf{X}^{(s)}_{(1:L),t},\mathbf{X}^{(0)}_{(1:L),(0:t)}) &= \frac{q(\mathbf{X}^{(s)}_{(1:L),t}|\mathbf{X}^{(s-1)}_{(1:L),t},\mathbf{X}^{(0)}_{(1:L),(0:t)})q(\mathbf{X}^{(s-1)}_{(1:L),t}|\mathbf{X}^{(0)}_{(1:L),(0:t)})}{q(\mathbf{X}^{(s)}_{(1:L),t}|\mathbf{X}^{(0)}_{(1:L),(0:t)})} && \text{(By Bayesian formula)} \\
&= \frac{\prod_{l=1}^{L} q(\mathbf{X}^{(s)}_{l,t}|\mathbf{X}^{(s-1)}_{l,t},\mathbf{X}^{(0)}_{l,(0:t)}) \prod_{l=1}^{L} q(\mathbf{X}^{(s-1)}_{l,t}|\mathbf{X}^{(0)}_{l,(0:t)})}{\prod_{l=1}^{L} q(\mathbf{X}^{(s)}_{l,t}|\mathbf{X}^{(0)}_{l,(0:t)})} && \text{(By Eq. (11))} \\
&= \prod_{l=1}^{L} \frac{q(\mathbf{X}^{(s)}_{l,t}|\mathbf{X}^{(s-1)}_{l,t},\mathbf{X}^{(0)}_{l,(0:t)})q(\mathbf{X}^{(s-1)}_{l,t}|\mathbf{X}^{(0)}_{l,(0:t)})}{q(\mathbf{X}^{(s)}_{l,t}|\mathbf{X}^{(0)}_{l,(0:t)})} \\
&= \prod_{l=1}^{L} q(\mathbf{X}^{(s-1)}_{l,t}|\mathbf{X}^{(s)}_{l,t},\mathbf{X}^{(0)}_{l,(0:t)}).
\end{aligned}
$$

### F.3.3 DENOISING DISTRIBUTION

Formally, layers exhibit interdependencies and implicit co-evolution in a multi-layer temporal graph sequence. To model such co-evolution, since layers are conditionally independent given the clean snapshots $\mathbf{X}^{(0)}_{(1:L),(0:t-1)}$, the denoising distribution is the product of the distributions of all layers

conditioned on $\mathbf{X}_{(1:L),(0:t-1)}^{(0)}$ in Eq. (13) as follows:

$$p_\theta(\mathbf{X}_{(1:L),t}^{(0)}|\mathbf{X}_{(1:L),t}^{(s)}, \mathbf{X}_{(1:L),(0:t-1)}^{(0)}) = \prod_{l=1}^{L} p_\theta(\mathbf{X}_{l,t}^{(0)}|\mathbf{X}_{l,t}^{(s)}, \mathbf{X}_{(1:L),(0:t-1)}^{(0)}),$$

where $p_\theta(\mathbf{X}_{l,t}^{(0)}|\mathbf{X}_{l,t}^{(s)}, \mathbf{X}_{(1:L),(0:t-1)}^{(0)})$ is learned by a denoising network with a cross-layer attention mechanism, with weights determined by the predicted cross-layer coupling graph $G_t^{(C)}$, enabling it to capture implicit co-evolution through cross-layer attention in Eq. (14) as follows.

$$p_\theta(\mathbf{X}_{l,t}^{(0)}|\mathbf{X}_{l,t}^{(s)}, \mathbf{X}_{(1:L),(0:t-1)}^{(0)}) = \sum_{G_t^{(C)}} p_\theta(\mathbf{X}_{l,t}^{(0)}|\mathbf{X}_{l,t}^{(s)}, \mathbf{X}_{(1:L),(0:t-1)}^{(0)}, G_t^{(C)}) p_\theta(G_t^{(C)}|\mathbf{X}_{(1:L),(0:t-1)}^{(0)}),$$

where $p_\theta(G_t^{(C)}|\mathbf{X}_{(1:L),(0:t-1)}^{(0)})$ serves as a learnable prior over cross-layer dependencies. It maps the clean historical snapshots into a latent adjacency-like structure that reflects the current degree of coupling across layers at timestamp $t$. This latent graph then parameterizes the cross-layer attention mechanism in $p_\theta(\mathbf{X}_{l,t}^{(0)}|\mathbf{X}_{l,t}^{(s)}, \mathbf{X}_{(1:L),(0:t-1)}^{(0)}, G_t^{(C)})$, meaning that the denoiser's parameters are modulated by the inferred strength of cross-layer correlations. As a result, cross-layer dependencies are injected before factorization, ensuring that the final product form in Eq. (13) still embeds inter-layer influence. This mechanism allows the model to capture both edge-level cross layer correlations (when explicit inter-layer edges exist) and higher-order co-evolution patterns (when groups of nodes across layers move together in time, even without explicit inter-layer links).

### F.3.4 REVERSE DENOISING PROCESS

The reverse process of a multi-layer graph sequence conditioning on the clean snapshots $\mathbf{X}_{(1:L),(0:t-1)}^{(0)}$ is the product of the reverse processes of all layers in Eq. (15) as follows:

$$p_\theta(\mathbf{X}_{(1:L),t}^{(s-1)}|\mathbf{X}_{(1:L),t}^{(s)}, \mathbf{X}_{(1:L),(0:t-1)}^{(0)}) = \prod_{l=1}^{L} p_\theta(\mathbf{X}_{l,t}^{(s-1)}|\mathbf{X}_{l,t}^{(s)}, \mathbf{X}_{(1:L),(0:t-1)}^{(0)}),$$

where the reverse process of each layer $l$ can be derived by approximation using the posterior (Eq. (12)) and denoiser (Eq. (14)) of each layer $l$ as in Eq. (16),

$$p_\theta(\mathbf{X}_{l,t}^{(s-1)}|\mathbf{X}_{l,t}^{(s)}, \mathbf{X}_{(1:L),(0:t-1)}^{(0)}) = \sum_{\mathbf{X}_{l,t}^{(0)}} q(\mathbf{X}_{l,t}^{(s-1)}|\mathbf{X}_{l,t}^{(s)}, \mathbf{X}_{l,(0:t)}^{(0)}) p_\theta(\mathbf{X}_{l,t}^{(0)}|\mathbf{X}_{l,t}^{(s)}, \mathbf{X}_{(1:L),(0:t-1)}^{(0)}).$$

The reverse process factorizes across layers because each layer produces its own categorical transition from step $s$ to $s-1$. However, the transition probabilities themselves are not independent: they depend on the shared cross-layer attention weights computed from the predicted coupling graph $G_t^{(C)}$. Thus, each layer's reverse update incorporates information from all other layers before generating its own transition. This design cleanly separates (1) how information flows across layers (attention) and (2) how categorical diffusion is applied per layer (reverse transition), making the derivation tractable while still capturing rich cross layer co-evolution.

## G DENOISING NETWORK ARCHITECTURE

By exploiting the cross-layer coupling graph $G_t^{(C)}$, we simultaneously introduce the notion of **intra-layer** denoising and **inter-layer** denoising. The former reconstructs the intra-layer structure by learning its distribution via a denoising network with $G_t^{(C)}$ learned from a cross-layer correlation predictor as a condition (detailed later). The latter models the relationships between distinct layers by considering the correlations between different layers in $G_t^{(C)}$. Specifically, to generate $\hat{G}$, it is necessary to consider their intra-layer graph $(G_l, G_m)$ and cross-layer coupling graph $(G_t^{(C)})$ as conditions in the reverse denoising process so that $\hat{G}_{(l,m),t}$ can be generated by considering $G_{l,t}, G_{m,t}$ and the correlation between them through the cross-layer correlation predictor.

As illustrated in Figure 5, we extract the encoded embeddings from $\mathbb{G}_{0:t-1}$ to capture the dynamics from $\mathbb{G}_{0:t-1}$ via the encoder (detailed in Figure 6a) equipped with a self-attention mechanism (detailed in Figure 7a). Then we follow (Vignac et al., 2023) to build the decoder (detailed in Figure 6b) with a cross-attention mechanism (detailed in Figure 7b) by extracting structural and

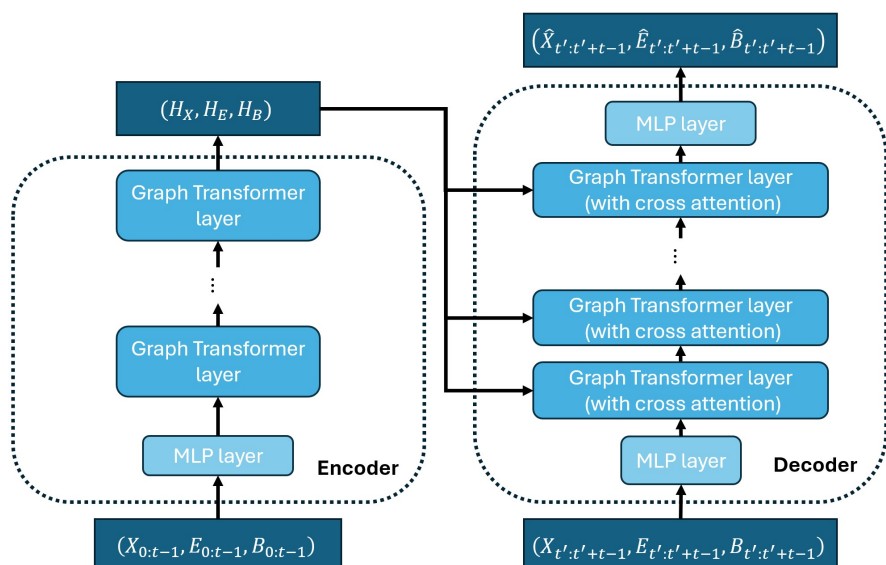

Figure 5: Denoising network architecture.

spectral features of noisy graphs $\mathbb{G}_{t:T}^{(s)} = (\{\mathbf{X}_{l,(t:T)}^{(s)}, \mathbf{E}_{l,(t:T)}^{(s)}\}_{l=1}^{L}, \{\mathbf{B}_{(l,m),(t:T)}^{(s)}\}_{l \neq m})$ at diffusion step $s$ during timestamps $t$ to $T$, which are then processed by an MLP layer and graph transformer layers with cross attention (detailed in Figure 7b). Then the denoised graph can be derived by exploiting the following prediction results from the noisy graphs $\mathbb{G}_{t:T}^{(s)}$ and embeddings $\mathbf{X}_{l,(t:T)}^{(0)}$, $\mathbf{E}_{l,(t:T)}^{(0)}$, and $\mathbf{B}_{(l,m),(t:T)}^{(0)}$, of snapshots from timestamp $t$ to $T$ by the denoising network into the result derived from Theorem 4.2

$$\hat{\mathbf{X}}_{l,t}^{(0)} = \phi_\theta(\mathbf{X}_{l,t}^{(s)}, \mathbf{X}_{l,(0:t-1)}^{(0)}, s), \hat{\mathbf{E}}_{l,t}^{(0)} = \phi_\theta(\mathbf{E}_{l,t}^{(s)}, \mathbf{E}_{l,(0:t-1)}^{(0)}, s), \hat{\mathbf{B}}_{(l,m),t}^{(0)} = \phi_\theta(\mathbf{B}_{(l,m),t}^{(s)}, \mathbf{B}_{(l,m),(0:t-1)}^{(0)}, s),$$

which are plugged into Eq. (10) to denoise noisy graphs.

**Intra-layer denoising.** By extending Eq. (10), the intra-layer denoising process for each layer $l$ can be derived from the following marginalization.

$$p_\theta(G_{l,t}^{(s-1)}|G_{l,t}^{(s)}, \mathbb{G}_{0:t-1}^{(0)}) = \sum_{G_{l,t}^{(0)}} q(G_{l,t}^{(s-1)}|G_{l,t}^{(s)}, G_{l,(0:t-1)}^{(0)}) p_I(G_{l,t}^{(0)}|G_{l,t}^{(s)}, \mathbb{G}_{0:t-1}^{(0)}), \quad (22)$$

where $p_I$ can be learned by training a denoising network to predict $G_{l,t}^{(0)}$ from $G_{l,t}^{(s)}, \mathbb{G}_{0:t-1}^{(0)}$ with the assistance of a cross-layer coupling graph $G_t^{(C)}$ by marginalization over the edges $(l, m)$ incident to node $l$ in the cross-layer coupling graph (i.e., layers connecting to $l$)

$$p_I(G_{l,t}^{(0)}|G_{l,t}^{(s)}, \mathbb{G}_{0:t-1}^{(0)}) = \sum_{(l,m) \in E(G_t^{(C)})} \hat{p}_I(G_{l,t}^{(0)}|G_{l,t}^{(s)}, \mathbb{G}_{0:t-1}^{(0)}, G_t^{(C)}) \hat{p}_C(G_t^{(C)}|G_{l,t}^{(s)}, \mathbb{G}_{0:t-1}^{(0)})$$

with $\hat{p}_I$ learned by training a denoising network to predict $G_{l,t}^{(0)}$ from $G_{l,t}^{(s)}, \mathbb{G}_{0:t-1}^{(0)}$ and $G_t^{(C)}$, and $\hat{p}_C$ learned by training a neural network-based cross-layer correlation predictor to predict the link in $G_t^{(C)}$ from $G_{l,t}^{(s)}, \mathbb{G}_{0:t-1}^{(0)}$, addressing the need for cross-layer correlation that is not supported in existing static and dynamic graph generators.

**Inter-layer denoising.** The reverse denoising process of inter-layer edges $\mathbf{B}_{(l,m),t}$ can be derived through marginalization (extended from Eq. (10)):

$$p(\mathbf{B}_{(l,m),t}^{(s-1)}|\mathbf{B}_{(l,m),t}^{(s)}, \mathbb{G}_{(0:t-1)}^{(0)}) = \sum_{\mathbf{B}_{(l,m),t}^{(0)}} q(\mathbf{B}_{(l,m),t}^{(s-1)}|\mathbf{B}_{(l,m),t}^{(s)}, \mathbf{B}_{(l,m),(0:t-1)}^{(0)}) p_B(\mathbf{B}_{(l,m),t}^{(0)}|\mathbf{B}_{(l,m),t}^{(s)}, \mathbb{G}_{(0:t-1)}^{(0)}),$$

$$(23)$$

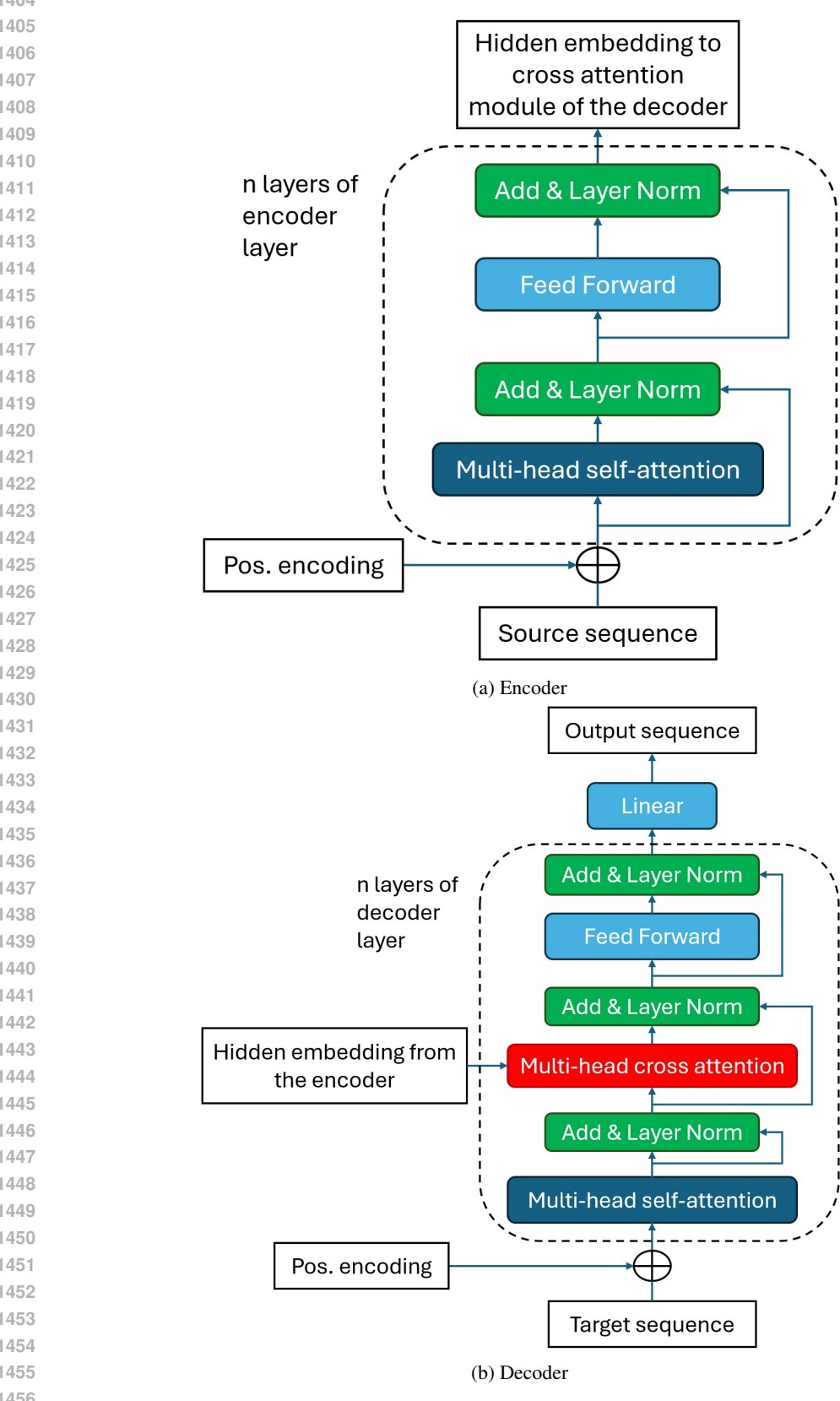

(a) Encoder

(b) Decoder

Figure 6: Detailed architectures of the encoder and decoder.

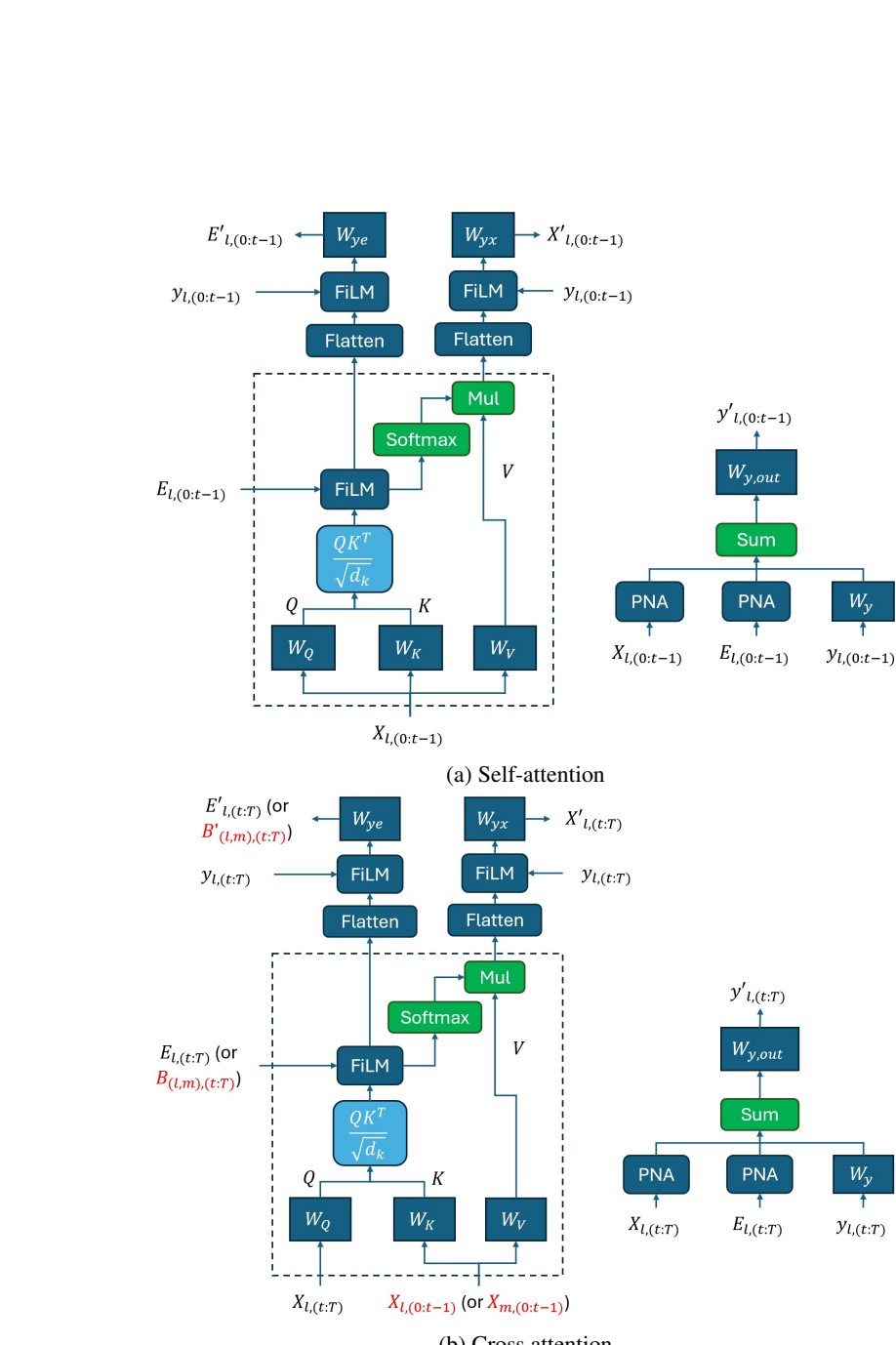

(a) Self-attention

(b) Cross attention

Figure 7: Detailed architectures of each module.

where $p_B$ can be learned by training a denoising network, considering $\mathbf{B}^{(s)}_{(l,m),t}$, $\mathbb{G}^{(0)}_{(0:t-1)}$, and the cross-layer coupling graph $G^{(C)}_t$ as conditions to predict $\mathbf{B}^{(0)}_{(l,m),t}$ by the following marginalization:

$$p_B(\mathbf{B}^{(0)}_{(l,m),t}|\mathbf{B}^{(s)}_{(l,m),t}, \mathbb{G}^{(0)}_{(0:t-1)})$$

$$= \sum_{(G^{(0)}_{l,t}, G^{(0)}_{m,t})} \left[ \hat{p}_B(\mathbf{B}^{(0)}_{(l,m),t}|\mathbf{B}^{(s)}_{(l,m),t}, \mathbf{B}^{(0)}_{(l,m),(0:t-1)}, (G^{(0)}_{l,t}, G^{(0)}_{m,t})) \cdot \hat{p}_I((G^{(0)}_{l,t}, G^{(0)}_{m,t})|\mathbf{B}^{(s)}_{(l,m),t}, \mathbb{G}^{(0)}_{(0:t-1)}) \right],$$

$$\hat{p}_I((G^{(0)}_{l,t}, G^{(0)}_{m,t})|\mathbf{B}^{(s)}_{(l,m),t}, \mathbb{G}^{(0)}_{(0:t-1)})$$

$$= \tilde{p}_I((G^{(0)}_{l,t}, G^{(0)}_{m,t})|\mathbf{B}^{(s)}_{(l,m),t}, \mathbb{G}^{(0)}_{(0:t-1)}, (l,m) \in E(G^{(C)}_t)) \cdot \tilde{p}_C((l,m) \in E(G^{(C)}_t)|\mathbf{B}^{(s)}_{(l,m),t}, \mathbb{G}^{(0)}_{(0:t-1)})$$

$$+ \tilde{p}_I((G^{(0)}_{l,t}, G^{(0)}_{m,t})|\mathbf{B}^{(s)}_{(l,m),t}, \mathbb{G}^{(0)}_{(0:t-1)}, (l,m) \notin E(G^{(C)}_t)) \cdot \tilde{p}_C((l,m) \notin E(G^{(C)}_t)|\mathbf{B}^{(s)}_{(l,m),t}, \mathbb{G}^{(0)}_{(0:t-1)}),$$

with $\tilde{p}_I$ learned by a denoising network to predict the structure of layers $l$ and $m$ from $\mathbf{B}^{(s)}_{(l,m),t}, \mathbb{G}^{(0)}_{(0:t-1)}$ and $\tilde{p}_C$ learned by the cross-layer predictor to predict whether there is any correlation between layers $l$ and $m$ at timestamp $t$ from $\mathbf{B}^{(s)}_{(l,m),t}, \mathbb{G}^{(0)}_{(0:t-1)}$.

As illustrated in Figure 5, the cross-layer correlation-aware denoising network $\phi^{(C)}_\theta$ is constructed by dividing a noisy multiplex graph $\mathbb{G}^{(s)}_t$ into intra-layer ($\{\mathbf{X}^{(s)}_{l,t}, \mathbf{E}^{(s)}_{l,t}\}^L_{l=1}$) and inter-layer ($\{\mathbf{B}^{(s)}_{(l,m),t}\}_{l \neq m}$) parts. The encoder part remains the same with using only temporal information. As for the decoder, we process the intra-layer part with the cross-attention layer using the output of the encoder from each layer $l$ in previous snapshots $G_{l,(0:t-1)}$ as the input of keys and values, and the target sequence $G_{l,(t:T)}$ for the corresponding layer $l$ as the query. The inter-layer part is processed by using the output of the encoder from different layers $m \neq l$ in previous snapshots $G_{m,(0:t-1)}$ as the input of keys and values, and the target sequence $G_{l,(t:T)}$ for the corresponding layer $l$ as the query, to predict the inter-layer links $\mathbf{B}_{(l,m),(t:T)}$ from the intra-layer graph and cross-layer coupling graph.

# H  DETAILS OF BEHAVIOR-AWARE GUIDANCE

## H.1  KURAMOTO-BASED SYNCHRONIZATION MEASURE

To quantify the synchronization level of a node attribute associated with a user $i$ at a specific time and layer, we adopt the *Kuramoto order parameter* (De Domenico, 2023; Danziger et al., 2019) as a continuous-phase descriptor, measured by the degree of phase coherence among nodes with the $m$-th attribute in layer $l$ at time $t$:

$$R^{(l)}_m(t) = \left| \frac{1}{N^{(l)}_t} \sum^{N^{(l)}_t}_{i=1} e^{j \cdot \theta^{(l)}_{i,m}(t)} \right|, \tag{24}$$

where $j$ is the imaginary unit and $N^{(l)}_t$ is the number of nodes in layer $l$ at time $t$.

Since this measure operates in the angular domain, we first normalize and project the $m$-th raw attribute value $a_m = \mathbf{x}^{(0)}_{l,t}[i,m], \forall m = 1,\ldots,M$ of node $i$ in layer $l$ at timestamp $t$ onto the unit circle:

$$\theta^{(l)}_{i,m}(t) = \pi \cdot \frac{(\mathbf{x}^{(0)}_{l,t}[i,m] - \min_j \mathbf{x}^{(0)}_{l,t}[j,m])}{(\max_j \mathbf{x}^{(0)}_{l,t}[j,m] - \min_j \mathbf{x}^{(0)}_{l,t}[j,m])}. \tag{25}$$

To clarify the motivation for behavior-aware guidance, we provide an illustrative example in Figure 8 inspired by the dynamics of real social platforms such as Instagram or X. These platforms support multiple interaction types—most notably repost/share and reply/comment—and each interaction has a directional nature. When user $A$ reposts user $B$'s post, $A$'s active repost count increases, and $B$'s passive repost count increases. The same active/passive semantics apply to replies. Such data naturally form a temporal multiplex network, where

- each layer corresponds to an interaction type (e.g., repost vs. reply).

- each attribute corresponds to the direction of participation (active vs. passive activity).

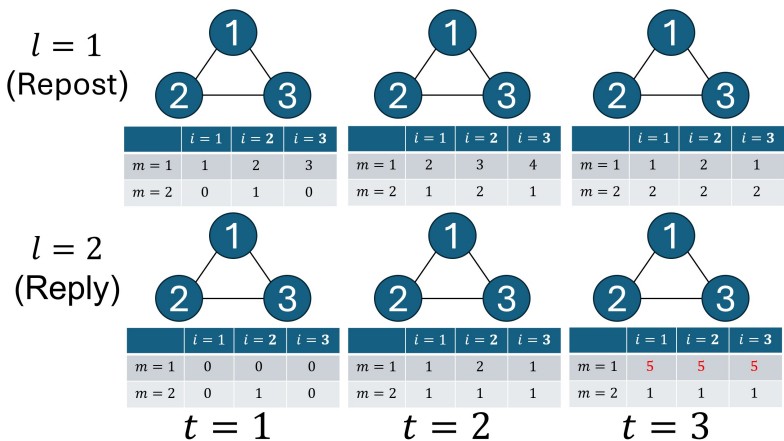

Figure 8: Illustrative example of behavior-aware guidance.

This representation reflects the fact that user behavior is usually not uniform across interactions: frequent reposters may rarely reply, and vise versa.

Consider a common scenario on Instagram. A brand plans a giveaway and instructs its followers: Before the giveaway opens ($t = 1, 2$), followers are encouraged to repost the promotional poster to inform their friends. This produces moderately elevated repost activity across the network—users amplify the message but not explosively. At the giveaway time ($t = 3$): The brand announces, "Leave a comment on our post now for a chance to win!" Suddenly, a large number of users reply at exactly the same moment, producing a sharp, collective surge—an explosive synchronization event—in the reply layer. Importantly, the reply activity at earlier times does not show strong pre-burst signs, even though the repost layer has already become active due to brand promotion. This pattern—one interaction channel warming up while another exhibits a sudden synchronized burst—is extremely common across real social platforms.

To make this concrete, we consider three nodes $i = 1, 2, 3$ connecting with one another, two layers: repost ($l = 1$) and reply ($l = 2$), and two directional activity attributes: active ($m = 1$) and passive ($m = 2$). We construct a minimal temporal sequence of interaction records $\mathbf{x}_{l,t}[i, m]$ (e.g., $\mathbf{x}_{1,2}[3, 1] = 5$ is the frequency of active ($m = 1$) repost ($l = 1$) at $t = 2$) that is consistent with the above scenario:

$$\mathbf{x}_{1,1}[i, 1] = [1, 2, 3]; \mathbf{x}_{1,2}[i, 1] = [2, 3, 4]; \mathbf{x}_{1,3}[i, 1] = [1, 2, 1],$$

representing moderately elevated active reposts during brand promotion.

Passive reposts are also moderately elevated:

$$\mathbf{x}_{1,1}[i, 2] = [0, 1, 0]; \mathbf{x}_{1,2}[i, 2] = [1, 2, 1]; \mathbf{x}_{1,3}[i, 2] = [2, 2, 2].$$

For the reply layer, the pre-event active reply activity is low:

$$\mathbf{x}_{2,1}[i, 1] = [0, 0, 0]; \mathbf{x}_{2,2}[i, 1] = [1, 2, 1],$$

but when the giveaway opens at $t = 3$, users synchronously comment: $\mathbf{x}_{2,3}[i, 1] = [5, 5, 5]$.

Passive replies are similarly stable:

$$\mathbf{x}_{2,1}[i, 2] = [0, 1, 0]; \mathbf{x}_{2,2}[i, 2] = [1, 1, 1]; \mathbf{x}_{2,3}[i, 2] = [1, 1, 1].$$

Thus, the repost layer shows early promotional activity at $t = 2$, while the reply layer exhibits a sudden burst at $t = 3$, mimicking the real behavior observed on social platforms during time-sensitive events. This example highlights that collective temporal behavior, such as event-driven synchronous reply bursts, cannot be captured by structure-only generators. Traditional models replicate edges, degrees, and multiplex topology but fail to reproduce: 1) burst intensity and 2) cross-interaction causal influence (repost $\rightarrow$ reply burst).

Thus, behavior-aware guidance is essential for generating temporal multiplex networks that faithfully preserve how users behave over time—not just their connections with one another.

## H.2   HYSTERESIS

In dynamic social networks, hysteresis refers to the phenomenon where a system's state evolution is path-dependent (De Domenico, 2023; Danziger et al., 2019), which implies that the current state of a network does not immediately return to its original state even after external driving factors (e.g., attention and engagement stimuli) are removed or after a phenomenon of state change (such as explosive synchronization) ends. Such a phenomenon particularly plays a crucial role in the dynamics of temporal attributes, in which the lagging inertia of user behaviors or sentiments is revealed after explosive changes. For instance, even when the initial stimuli subside, there still exists a sustained high activity following a viral trend.

To capture such a behavioral effect in generated networks, we design a hysteresis-aware regularization based on global synchronization descriptors derived from the Kuramoto-based synchronization measure. Specifically, the Kuramoto-based synchronization measure quantifies synchronization among node attributes (e.g., engagement levels) as a function of $\theta_{i,m}^{(l)}$ in Eq. (25). During hysteresis, the Kuramoto order parameter in Eq. (24) exhibits a bistable loop—the trajectory during increasing $\theta_{i,m}^{(l)}(t)$ (forward) diverges from the trajectory during decreasing $\theta_{i,m}^{(l)}(t)$ (backward). To this end, we formulate the hysteresis-aware regularization term as follows:

$$\mathcal{L}_{\text{hyst}} = -|R_{m,f} - R_{m,b}|, \tag{26}$$

where $R_{m,f}(t)$ and $R_{m,b}(t)$ exhibit the trajectories of increasing and decreasing values of $\theta_{i,m}^{(l)}(t)$, respectively:

$$R_{m,f}(t) = \sum\nolimits_{t:\Delta R_m^{(l)}(t)>0} R_m^{(l)}(t); R_{m,b}(t) = \sum\nolimits_{t:\Delta R_m^{(l)}(t)<0} R_m^{(l)}(t).$$

This loss encourages the generated graph sequences to exhibit non-reversible dynamics consistent with real-world social hysteresis.

## H.3   ILLUSTRATIVE EXAMPLE SHOWING BEHAVIOR-AWARE GUIDANCE

Motivated by the above observation from Figure 8, we follow the Kuramoto model to define the phase angle of the $m$-th attribute of each node $i$ in layer $l$ at timestamp $t$ by using the attribute value $\mathbf{x}_{l,t}^{(0)}[i,m]$ of the $m$-th attribute (e.g., the frequency of delivering reposts) of each node $i$ in layer $l$ at timestamp $t$ in Eq. (25).

We map these activity levels of active ($m = 1$) reply ($l = 2$) to Kuramoto phases as follows:

$$\theta_{1,1}^{(2)}(1) = \theta_{2,1}^{(2)}(1) = \theta_{3,1}^{(2)}(1) = 0;$$

$$\theta_{1,1}^{(2)}(2) = \pi \cdot \tfrac{1-1}{2-1} = 0; \theta_{2,1}^{(2)}(2) = \pi \cdot \tfrac{2-1}{2-1} = \pi; \theta_{3,1}^{(2)}(2) = \pi \cdot \tfrac{1-1}{2-1} = 0;$$

$$\theta_{1,1}^{(2)}(3) = \theta_{2,1}^{(2)}(3) = \theta_{3,1}^{(2)}(3) = 0,$$

which yields dispersed phases at $t = 1$, perfectly aligned phases at the burst time $t = 2$, and dispersed phases again at $t = 3$.

Then we denote the synchronization degree $R_m^{(l)}(t)$ of the $m$-th attribute in interaction layer $l$ at timestamp $t$ by following Kuramoto model $R_m^{(l)}(t) = \left| \frac{1}{N_t^{(l)}} \sum_{i=1}^{N_t^{(l)}} e^{j \cdot \theta_{i,m}^{(l)}(t)} \right|$ (Eq. (24)), which quantifies collective phase coherence: values near 1 indicate that many nodes occupy nearly the same behavioral stage, whereas values near 0 reflect dispersed or uncoordinated behavior. To observe the explosive synchronization, we calculate the corresponding variance with Eq. (19) in Sec. 4.2.3 accordingly to obtain

$$R_1^{(2)}(1) = |\tfrac{1}{3}(1+1+1)| = 1, R_1^{(2)}(2) = |\tfrac{1}{3}(1-1+1)| = 1/3, R_1^{(2)}(3) = |\tfrac{1}{3}(1+1+1)| = 1,$$

whose first-order differences $\Delta R_1^{(2)}(1) = 1/3 - 1 = -2/3, \Delta R_1^{(2)}(2) = 1 - 1/3 = 2/3$, and $\overline{\Delta R_1^{(2)}} = 0$ produce a large variance $\text{Var}_1^{(2)} = \frac{1}{3-1}[(-2/3)^2 + (2/3)^2] = 4/9$ by Eq. (19) in Sec. 4.2.3, capturing a strong explosive-synchronization phenomenon of active reply.

The calculation results of the variances in all layers and attributes are listed in Table 6. From the

Table 6: Variance calculation in the illustrative example.

| Layer $l$ | Attribute $m$ | $R_m^{(l)}(t)$ $(t = 1, 2, 3)$ | $\Delta R_m^{(l)}(t)$ $(t = 1, 2)$ | $Var_m^{(l)}$ |
|---|---|---|---|---|
| 1 | 1 | $\{1/3, 1/3, 1/3\}$ | $\{0, 0\}$ | 0 |
| 1 | 2 | $\{1/3, 1/3, 1\}$ | $\{0, +2/3\}$ | 1/9 |
| 2 | 1 | $\{1, 1/3, 1\}$ | $\{-2/3, +2/3\}$ | 4/9 |
| 2 | 2 | $\{1/3, 1, 1\}$ | $\{+2/3, 0\}$ | 1/9 |

above results, since the variance $Var_1^{(2)}$ in active reply behaviors is the largest among all behavior types, we observe that the explosive synchronization phenomenon of active reply behaviors is strong.

To observe the hysteresis phenomenon, we follow the above example and observe that the rising and falling trajectories of active ($m = 1$) reply ($l = 2$) differ ($R_{1,f} = R_1^{(2)}(2) = 1/3, R_{1,b} = R_1^{(2)}(3) = 1 \Rightarrow |R_{m,f} - R_{b,f}| = 1 - 1/3 = 2/3$), indicating a clear hysteresis gap. The values of $|R_{m,f} - R_{b,f}|$ in all layers and attributes are calculated in the following Table 7.

Table 7: Hysteresis calculation in the illustrative example.

| Layer $l$ | Attribute $m$ | $R_m^{(l)}(t)$ $(t = 1, 2, 3)$ | $\Delta R_m^{(l)}(t)$ $(t = 1, 2)$ | $|R_{m,f} - R_{m,b}|$ |
|---|---|---|---|---|
| 1 | 1 | $\{1/3, 1/3, 1/3\}$ | $\{0, 0\}$ | 0 |
| 1 | 2 | $\{1/3, 1/3, 1\}$ | $\{0, +2/3\}$ | 1 |
| 2 | 1 | $\{1, 1/3, 1\}$ | $\{-2/3, +2/3\}$ | 2/3 |
| 2 | 2 | $\{1/3, 1, 1\}$ | $\{+2/3, 0\}$ | 1 |

## H.4  MORE DISCUSSIONS

**Rationale for using a temporal multiplex representation.** We clarify that multiplexity is not required, but is a natural and effective modeling choice when the underlying system involves interacting contexts. While explosive synchronization and hysteresis can arise on non-multiplex graphs, our motivation for adopting temporal multiplex graphs follows (De Domenico, 2023; Danziger et al., 2019), which emphasize that many real systems exhibiting such behaviors unfold across multiple interacting contexts (e.g., different social platforms). In such settings, single-layer representations often collapse cross-context dependencies that critically shape the resulting dynamics, making it difficult for a generative model to preserve behavior. Temporal multiplex graphs, by contrast, offer an explicit structure for representing interdependence, competition, and asymmetry across layers, which better preserves the mechanisms driving explosive synchronization and hysteresis.

**Clarification that no alignment is required.** We clarify that we use the Kuramoto order parameter purely as a phase-coherence descriptor without simulating continuous-time Kuramoto dynamics; therefore, no alignment between oscillator time and graph timestamps is required. To make behavior guidance computationally feasible inside the diffusion steps, we compute the Kuramoto order parameter snapshot-wise by projecting node attributes to phases, rather than simulating continuous-time Kuramoto dynamics. This avoids the substantial computational and implementation complexity that full Kuramoto ODE simulation would require, and keeps the guidance practical for long temporal sequences. In addition, our formulation avoids any assumptions about timescales. Since the order parameter is computed independently for each snapshot, there is no continuous oscillator time variable that must be aligned with the discrete timestamps of the temporal graph.

**Spectral properties for behavior guidance.** We also discuss the usage of spectral properties for behavior guidance, following the multiplex-spectral framework (Berner et al., 2021; Liu et al., 2024). The idea is fully compatible with our formulation, and the spectral quantities such as grounded supra-Laplacian eigenvalues (Liu et al., 2024) in and spectral heterogeneity in (Berner et al., 2021) can naturally serve as behavior descriptors within our behavior-aware guidance mechanism.

- Pinning-control spectral theory in (Liu et al., 2024) provides grounded supra-Laplacian eigenvalues such as $\lambda_1(\tilde{L})$ which quantify the structural tendency of a multiplex network to sustain or resist coherent states. A derived index such as $1/\lambda_1(\tilde{L})$ can be directly inserted into the behavior-aware loss as a structural guidance signal.

- Multiplex decomposition and generalized master stability analysis in (Berner et al., 2021) offer mode-wise stability parameters $\psi_i$ derived from the supra-Laplacian spectrum. Their

Table 8: Complexity comparison between MulDyDiff and baselines.

| Model/Algorithm | Parameter count |
|---|---|
| MulDyDiff (Ours) | $O(Sl(nd_x + n^2 d_e))$ |
| MoDiff (Xu & Ma, 2025) | $O(kn^2 + Sln^2 d)$ |
| DAMNETS (Clarkson et al., 2022) | $O(Tnd)$ |
| DYMOND (Zeno et al., 2021) | $O(n^3 T)$ |
| TagGen (Zhou et al., 2020) | $O(n^2 T^2)$ |
| AGE (Fan & Huang, 2020) | $O(n^2 dl)$ |

Table 9: Multi-layer temporal social network dataset statistics.

| Dataset | # layers | # nodes | # edges |
|---|---|---|---|
| Wiki-vote | 4 | 7.1K | 103K |
| Twitter | 3 | 456K | 14M |
| Superuser | 3 | 194K | 1.44M |

distribution (e.g., $\mathrm{Var}_i(\psi_i)$) characterizes spectral heterogeneity across modes and can likewise be integrated as a structural behavior descriptor.

These spectral quantities serve as structural indicators that enrich the behavioral information available to the diffusion process. Our formulation of behavior guidance does not require any modifications to accommodate these spectral terms, and their inclusion naturally broadens the range of behaviors that can be captured.

## I COMPLEXITY ANALYSIS

The parameter counts of MulDyDiff and the baselines are presented in Table 8. MulDyDiff requires $O(Sl(nd_x + n^2 d_e))$ to process graph snapshots (in parallel with the graph transformer architecture), where $S$ denotes the number of diffusion steps; $l$ denotes the number of (graph) transformer layers; $n$ denotes the number of nodes; $d_x$ denotes the number of node attributes; $d_e$ denotes the number of edge attributes. MoDiff (Xu & Ma, 2025) requires $O(n^2)$ for Hermitian encoding, $O(kn^2)$ to perform spectral decomposition with $k$ eigenvalues selected, and $O(Sln^2 d)$ for a $l$-layer GCN model denoising a $d$-dimensional features of $n$ nodes with $S$ diffusion steps. DAMNETS (Clarkson et al., 2022) requires $O(n)$ to process an adjacency row for each of $T$ snapshots, and each node has a $d$-dimensional embedding. DYMOND (Zeno et al., 2021) requires $O(n^3 T)$ to scan 3-node motifs in each of $T$ snapshots. TagGen (Zhou et al., 2020) requires $O(n^2 T^2)$ to process random-walk sequences of length $T$ with bi-level self-attention. AGE (Fan & Huang, 2020) requires $O(n^2 dl)$ for $l$ self-attention layers in the encoder. Although DAMNETS has lower complexity than MulDyDiff, DAMNETS is insufficient to capture cross-layer dependency and graph-level behaviors.

The complexity of processing a graph snapshot is $O(Sl(nd_x + n^2 d_e))$, where $S$ denotes the number of diffusion steps; $l$ denotes the number of (graph) transformer layers; $n$ denotes the number of nodes; $d_x$ denotes the number of node attributes; $d_e$ denotes the number of edge attributes. To reduce the overhead for scaling to very large network datasets, we process the layers in parallel with cross-layer attention mechanism according to the weights determined by cross-layer predictor. In order to achieve efficiency and avoid error propagation, we aggregate the past snapshots with dynamic transition function to generate future snapshots according to clean previous snapshots instead of noisy ones with a graph transformer architecture in parallel.

Furthermore, the cross-layer attention mechanism in MulDyDiff can adopt sparse attention in convolutional transformer layers and message-passing with a random attention mechanism in (Qin et al., 2025) to restrict attention to existing edges, reducing the time complexity of processing edges from $O(ln^2 d_e)$ to $O(lmd_e)$, where $l$ denotes the number of (graph) transformer layers; $n$ denotes the number of nodes; $m$ denotes the number of edges; $d_e$ denotes the number of edge attributes.

## J EXPERIMENT SETTINGS

**Details of the datasets.** The experiments are conducted on three real-world multiplex temporal networks: 1) The Wiki-vote dataset (Leskovec et al., 2010) has 4 layers containing nominations and voting with 3 types of opinions (support/neutral/oppose). 2) The Twitter dataset (De Domenico et al.,

Table 10: Hyperparameter settings.

| Hyperparameter | Value |
|---|---|
| Graph transformer layers | 6 |
| Batch size | 64 |
| Epochs | 50 |
| Learning rate | 0.2 |
| Weight decay | $10^{-12}$ |
| Diffusion steps ($S$) | 1000 |
| $\overline{\alpha}_s$ | $\cos^2\left[0.5\pi\left(\frac{s/S+p}{1+p}\right)\right], p = 0.008$ |
| $\overline{\gamma}_s$ | $\eta\overline{\alpha}_s + (1-\eta), \eta = 0.5$ |

2013) has 3 layers, with each layer representing relationships of retweets, mentions, and sending messages to others. 3) The Superuser dataset (Paranjape et al., 2017) also has 3 layers, with each layer consisting of answering questions, commenting on questions, and commenting on answers. The statistics of the datasets are presented in Table 9.

**Input data preparation.** For each dataset, we sample 20% of the input dataset as the testing set, and then we sample 80% of the remaining data as the training set and 20% of the remaining data as the validation set. The batch size is 16 graphs per batch. Since the memory of a GPU card has a capacity of around 1000 nodes, we sample 100 subgraphs with an average of 50 nodes for each layer in each of the datasets.

**Computing resources.** The experiments are conducted on a server equipped with 2 Intel(R) Xeon(R) Gold 6154 @ 3.00 GHz CPUs, 8 NVIDIA Tesla V100 GPUs, and 720 GB RAM.

**Metrics.** The usage of KS metrics is explained as follows.

KS metrics: We choose KS metrics since it better captures local and global property changes of graph sequences than MMD. According to Zeno et al. (2021), the importance of KS metrics for significant graph property change detection lies in 1) assessing the discrepancy of the distributions between input and generated graph sequences, 2) capturing individual behavior of each node and joint behavior of all nodes in a graph snapshots, and 3) benefits in capturing variability or dispersion with KS tests on inter-quartile range, in which the latter two cannot be achieved by MMD.

Per-layer degree/betweenness centrality distributions: This metric measures the distributional fidelity of degree and betweenness centrality via the KS distance. Matching these distributions verifies whether the generator preserves the heterogeneity of roles within each layer (i.e., the relative proportions of hubs, bridges, and peripheral nodes), which strongly influences temporal dynamics.

Cross-layer node-behavior distribution: It is defined as the number of unique neighbors a node has across all layers. This metric assesses whether the model accurately reproduces the heterogeneity of cross-layer engagement, thereby complementing the per-layer centrality metrics.

Cross-layer random-walk reachability distribution: This metric evaluates cross-layer, multi-hop accessibility by the number of distinct nodes reachable within a fixed-length random walk. It tests whether the generator preserves higher-order structural connectivity beyond direct neighbors.

**Parameter settings.** The hyperparameter settings used to derive the main results are listed in Table 10. We train the denoising network, which consists of 6 graph transformer layers, for 50 epochs with the Adam optimizer, in which the learning rate is set to 0.2, and the weight decay is set to $10^{-12}$. The number of steps in the proposed models is set to 1000.

# K  ADDITIONAL EXPERIMENT RESULTS

## K.1  MMD RESULTS

Table 11 presents the evaluation results of MulDyDiff compared with the baselines on Wiki-vote, Twitter and Superuser. On the Wiki-vote dataset, MulDyDiff outperforms most baselines including MoDiff in almost all metrics since MulDyDiff jointly considers structural and attribute changes. In contrast, since MoDiff primarily focuses on analyzing motif structure changes with spectral values, it does not perform well in most cases. However, there is no model with consistent outperformance in

Table 11: Comparative study results on MMD metrics.

| Data | Model | degree ($\downarrow$) | spectral ($\downarrow$) |
|---|---|---|---|
| Wiki-vote | AGE | 0.5145 | 0.3395 |
| | DAMNET | 0.3904 | 0.6162 |
| | TagGen | 0.6365 | 0.4231 |
| | DYMOND | 0.5384 | 0.3069 |
| | MoDiff | 0.9919 | 0.4127 |
| | MulDyDiff | 0.3676 | 0.3214 |
| Twitter | AGE | 0.2000 | 0.1607 |
| | DAMNET | 0.3123 | 0.1895 |
| | TagGen | 0.5624 | 0.2702 |
| | DYMOND | 0.3132 | 0.1536 |
| | MoDiff | 1.0128 | 0.6621 |
| | MulDyDiff | 0.4062 | 0.3402 |
| Superuser | AGE | 0.3670 | 0.1345 |
| | DAMNET | 0.2919 | 0.1729 |
| | TagGen | 0.4420 | 0.1965 |
| | DYMOND | 0.2461 | 0.1040 |
| | MoDiff | 0.9396 | 0.2255 |
| | MulDyDiff | 0.3389 | 0.3023 |

Table 12: Ablation study results on Wiki-vote.

| Component | KS-node behavior ($\downarrow$) | KS-RW ($\downarrow$) |
|---|---|---|
| Plain | 1.0000 | 1.0000 |
| Temporal | 0.6953 | 0.2563 |
| Cross-layer | 0.8038 | 0.6000 |
| Both | 0.5430 | 0.2219 |

the evaluated MMD metrics on all datasets, demonstrating that only using MMD is insufficient to assess the effectiveness of multiplex dynamic graph generation.

### K.2 ABLATION STUDY

To examine the model's capability in handling multiplex dynamic networks, we compare the full version of MulDyDiff with three variants: 1) *Plain*, which corresponds to static single-layer generation, omitting both temporal and cross-layer denoising; 2) *Temporal*, supporting dynamic single-layer networks; 3) *Cross-layer*, capturing static multiplex structures.

Table 12 demonstrates the results evaluated on Wiki-vote. MulDyDiff generally outperforms the variants that consider only cross-layer correlations or temporal dynamics in terms of KS metrics because it effectively addresses the joint considerations in both dimensions. Furthermore, on the Wiki-vote dataset, temporal dynamics perform better than the plain ones, and considering both mechanisms is superior to denoising with solely cross-layer correlation because attribute-aware and dynamic transition-aware denoising effectively capture the evolution in structural and attribute information. Without considering layer correlations and temporal dynamics, the plain variant performs the worst in terms of almost all metrics.

### K.3 SENSITIVITY TESTS

We conduct the sensitivity tests on Wiki-vote to evaluate the performance under various numbers model parameters, which are determined by the number of graph transformer layers, in Tables 13 and 14. From the results, we observe that the performance obtained with 2 to 6 graph transformer layers is comparable, but an excessive number of graph transformer layers may deteriorate the performance due to overfitting (Zhao et al., 2023). The advantages of MulDyDiff lie on the capability of capturing cross-layer dependencies cross-layer coupling with graph-level behaviors with behavioral-aware guidance instead of massive parameters.

We conduct sensitivity tests on Wiki-vote dataset with various diffusion steps and present the execution time of various diffusion steps in Table 15. From the above results, we observe that the training time does not vary significantly since we adopt multi-step Markov transition matrices with a sampled number $s$ of diffusion steps in each iteration of the training phase. Since the denoising process iterates in all steps, the sampling time slightly increases as the number of diffusion steps increases.

Table 13: Sensitivity test with various numbers of graph transformer layers (part 1).

| #GT-layers | MMD-degree | MMD-clustering | MMD-spectral | KS-node behavior | KS-random walk |
|---|---|---|---|---|---|
| 2 | 0.3897 | 0.9233 | 0.3768 | 0.9140 | 0.6937 |
| 4 | 0.3764 | 1.0734 | 0.3721 | 0.9296 | 0.6843 |
| 6 | 0.3923 | 0.8955 | 0.3818 | 0.9218 | 0.6968 |
| 8 | 0.4166 | 1.1583 | 0.4117 | 0.9765 | 0.7281 |

Table 14: Sensitivity test with various numbers of graph transformer layers (part 2).

| #GT-layers | KS-page rank | KS-degree centrality | KS-betweenness centrality | KS-closeness centrality |
|---|---|---|---|---|
| 2 | 0.5859 | 0.8281 | 0.6562 | 0.9140 |
| 4 | 0.5742 | 0.8710 | 0.6523 | 0.9335 |
| 6 | 0.5820 | 0.8398 | 0.6523 | 0.9218 |
| 8 | 0.5937 | 0.9609 | 0.6640 | 0.9765 |

Nevertheless, since we generate a batch of graph sequences, processing each snapshot requires only a few seconds, which is within the acceptable range.

To examine the sensitivity of parameter $\gamma_0$ in MulDyDiff, we conduct tests on Wiki-vote, Twitter, and Superuser, and report the results in Table 16. In all three datasets, the KS metrics perform better as $\gamma_0$ increases, showing the importance of temporal transition dynamics in the diffusion process of MulDyDiff.

To examine the sensitivity of MulDyDiff to sequence length, we conduct tests on dynamic attributed multiplex networks of varying lengths. We compare the training and sampling time of attribute-aware dynamic transition-aware denoising, with and without cross-layer correlation-aware denoising, to assess whether modeling inter-layer dependencies incurs additional time costs as sequences grow longer. Table 17 presents the results on the Twitter dataset. The results manifest that jointly considering temporal dynamics and cross-layer correlations is more scalable than using solely dynamic transition-aware denoising when dealing with temporal sequences of multiplex graphs, thanks to the parallel processing of the multiplex structure with the entire size distributed to each of the layers. Table 18 reports the MMD metrics of MulDyDiff as the number of diffusion steps increases on the Wiki-vote dataset. Table 19 presents the execution times of MulDyDiff in each phase with various batch sizes on Wiki-vote and Twitter. Table 20 measures the memory usage of MulDyDiff with various batch sizes on Wiki-vote and Twitter.

### K.4 TRAINING SCALABILITY AND SAMPLING TIME

To demonstrate the scalability of MulDyDiff, the results of the scalability tests on the Wiki-vote dataset (with 4 layers) are presented in Table 21. From the results, we observe that MulDyDiff is able to process graphs with 300 nodes per layer.

Besides, Table 22 presents (1) the training time under various amounts of graph sequences with different lengths extracted from the Wiki-vote, Twitter, and Superuser datasets over 50 training epochs, and (2) the elapsed time for sampling approximately 1000 multiplex graph sequences of various lengths. The results manifest that MulDyDiff scales well on both datasets since it processes the intra-layer and inter-layer parts in parallel. Wiki-vote requires less time for training and sampling than Twitter and Superuser, as graphs sampled from Twitter and Superuser instances have greater density than those sampled from Wiki-vote.

## L LLM USAGE

We use ChatGPT to aid or polish writing, and debugging in our implementation.

Table 15: Number of diffusion steps vs. training/sampling time on Wiki-vote.

| #Diffusion steps | Training time (min.) | Sampling time (min./per batch) |
|---|---|---|
| 250 | 79.118 | 18.406 |
| 500 | 75.787 | 18.251 |
| 750 | 77.665 | 21.021 |
| 1000 | 76.051 | 23.318 |

Table 16: Sensitivity tests varying $\gamma_0$.

| Data/$\gamma_0$ | | MMD-degree ($\downarrow$) | MMD-clustering ($\downarrow$) | MMD-spectral ($\downarrow$) | KS-node behavior ($\downarrow$) | KS-RW ($\downarrow$) |
|---|---|---|---|---|---|---|
| Wiki-vote | 0.25 | 0.3644 | 1.2376 | 0.3407 | 0.7930 | 0.3375 |
| | 0.5 | 0.3954 | 1.2396 | 0.3312 | 0.7422 | 0.3344 |
| | 0.75 | 0.2209 | 0.2171 | 0.2860 | 0.8038 | 0.6000 |
| | 1.0 | 0.3689 | 1.1393 | 0.3321 | 0.6523 | 0.2469 |
| Twitter | 0.25 | 0.4073 | 0.9589 | 0.3361 | 0.7637 | 0.2734 |
| | 0.5 | 0.3867 | 1.1729 | 0.3104 | 0.5931 | 0.1979 |
| | 0.75 | 0.4222 | 0.7639 | 0.3233 | 0.6143 | 0.1493 |
| | 1.0 | 0.4062 | 0.9875 | 0.3402 | 0.5957 | 0.1172 |
| Superuser | 0.25 | 0.3630 | 1.1751 | 0.3266 | 0.6836 | 0.3016 |
| | 0.5 | 0.3577 | 1.3202 | 0.3155 | 0.6426 | 0.3125 |
| | 0.75 | 0.3323 | 1.1665 | 0.3067 | 0.5664 | 0.1750 |
| | 1.0 | 0.3389 | 1.0552 | 0.3023 | 0.5484 | 0.1684 |

Table 17: Seq. length vs training/sampling time on Twitter.

| Variation | Seq. length | Training (hr.) | Sampling (hr.) |
|---|---|---|---|
| with cross-layer | 3 | 1.0184 | 0.9803 |
| | 4 | 1.4317 | 1.2201 |
| | 5 | 1.8044 | 1.5471 |
| w/o cross-layer | 3 | 1.3799 | 1.1472 |
| | 4 | 1.3355 | 1.5701 |
| | 5 | 1.6341 | 1.3075 |

Table 18: MMD vs. number of diffusion steps on Wiki-vote.

| #steps | 250 | 500 | 750 | 1000 |
|---|---|---|---|---|
| Degree dist. | 0.1713 | 0.2788 | 0.2627 | 0.2412 |
| Clustering Coeff. | 0.6319 | 0.7967 | 0.6443 | 0.8450 |
| Spectral | 0.1199 | 0.2030 | 0.1687 | 0.2001 |

Table 19: Execution time (sec.) vs. batch size in different phases.

| Data/Phase | | 32 | 64 | 128 | 256 |
|---|---|---|---|---|---|
| Wiki-vote | Training (per epoch) | 88.7 | 44.4 | 25 | 14.2 |
| | Validation (per epoch) | 14.9 | 7.8 | 4.6 | 2.8 |
| | Inference (per epoch) | 26.6 | 19.6 | 13.1 | 13.2 |
| | Sampling (per batch) | 22.88 | 91.53 | 139.95 | 227.13 |
| Twitter | Training (per epoch) | 996.8 | 624.9 | 443.8 | 107.4 |
| | Validation (per epoch) | 113.6 | 58.9 | 34.5 | 19.6 |
| | Inference (per epoch) | 153 | 82.8 | 49.9 | 34 |
| | Sampling (per batch) | 132.88 | 231.55 | 410.96 | 761.73 |

Table 20: Memory usage (MiB) vs. batch size in different phases.

| Dataset | 32 | 64 | 128 | 256 |
|---|---|---|---|---|
| Wiki-vote | 476.19 | 720.49 | 1183 | 2116 |
| Twitter | 357.43 | 480.38 | 744 | 1209 |

Table 21: Scalability test on Wiki-vote.

| Snapshot size (per layer) | Training time (hr.) |
|---|---|
| 50 | 1.0114 |
| 70 | 1.5179 |
| 100 | 5.7083 |
| 300 | 5.9675 |

Table 22: Training scalability and sampling time.

| Data/length | | Training (min.) | Sampling (min.) |
|---|---|---|---|
| Wiki-vote | 2 | 23.5871 | 26.9479 |
| | 4 | 34.9766 | 64.2513 |
| | 6 | 39.7568 | 108.4987 |
| Twitter | 2 | 54.8505 | 41.0588 |
| | 4 | 68.7585 | 55.0928 |
| | 6 | 79.9298 | 75.3005 |
| Superuser | 2 | 90.2640 | 63.2339 |
| | 4 | 104.6353 | 74.6572 |
| | 6 | 128.2297 | 99.5931 |

