# OpenReview forum: "On Diffusion-based Multiplex Dynamic Attributed Network Generator"
_ICLR.cc/2026/Conference — ICLR 2026 Conference Withdrawn Submission_

### Official Review · Reviewer_YWSR · 2025-11-01

**Soundness:** 3
**Presentation:** 2
**Contribution:** 3
**Rating:** 6
**Confidence:** 2

**Summary:**

This paper proposes MulDyDiff, a diffusion-based framework for generating multiplex dynamic attributed networks that evolve across time and layers. It addresses well-known limitations of prior models by jointly modeling structure–attribute coupling, temporal evolution, and inter-layer dependencies. The method further introduces behavior-aware guidance to reproduce emergent phenomena such as explosive synchronization and hysteresis. Experiments on three real-world datasets show consistent improvements (6–9%) over strong baselines, with clear ablation evidence supporting each component.

**Strengths:**

1. Clear and meaningful motivation
The paper nicely motivates why existing dynamic graph generators fall short.
2. Novel diffusion-based architecture
The proposed MulDyDiff framework extends discrete diffusion models to handle multiplex dynamic attributed graphs, combining attribute-aware, cross-layer correlation-aware, and behavior-aware denoising processes. This integration is technically sound and original to me.
2. Strong experimental results
The model shows consistent improvements over competitive baselines on three real-world datasets. The ablation studies are also convincing and clearly show why each component matters.

**Weaknesses:**

1. Behavioral validation is limited
The “behavior-aware” part is conceptually interesting, but I’d like to see stronger evidence that the generated networks really show realistic emergent behaviors
2. Baselines could be more up-to-date
The comparisons mostly cover traditional dynamic graph models. It would be nice to include more recent diffusion-based graph generators for completeness.
3. Efficiency and scalability discussion is light
The paper reports training/sampling time but doesn’t really analyze scalability with graph size or diffusion steps. Some discussion on this would help assess practicality.

**Questions:**

1. Why did you choose KS metrics only? Have you tried other dynamic metrics like MMD or behavior-based measures?
2. Can you show results comparing with and without the behavior-aware loss L_behavior to clarify its real impact?
3. How does the computational cost grow with the number of layers and timestamps? Would this scale to very large networks?

---

> ### Author Response · Authors · 2025-11-24
> **Answer to Q1**
>
> Thanks for the valuable comments. As suggested, we 1) conduct a comparative study with an additional baseline MoDiff [KDD25] to evaluate its MMD and KS metrics, and 2) explain the importance and advantages of KS metrics in detecting significant graph property change following [WWW21] (in Appendix J of the revised version), as well as comparative studies with MMD metrics.
>
> The results of MoDiff and MulDyDiff on Wiki-vote, Twitter, and Superuser datasets are presented in the following tables. On the Wiki-vote and Twitter datasets, MulDyDiff outperforms MoDiff on almost all metrics listed in the table since it captures structural and attributive evolution simultaneously. MoDiff slightly outperforms MulDyDiff in only MMD of spectral and KS of betweenness on the Superuser dataset. Nevertheless, the overall performance of MulDyDiff is better than MoDiff since MoDiff performs well in some MMD or KS metrics only on the Superuser dataset. This is insufficient to show the effectiveness of MoDiff in multi-layer temporal graph generation, as the effectiveness needs to assessed by various metrics comprehensively.
>
> |(Wiki-vote)|MMD (degree)|MMD (spectre)|KS (node behavior)|KS (random walk)|KS (degree centrality)|KS (betweenness centrality)|
> |-|-|-|-|-|-|-|
> |MoDiff|0.9919|0.4127|1.0000|0.4000|0.9474|0.8500|
> |MulDyDiff|0.3676|0.3214|0.5430|0.2219|0.8281|0.6563|
>
> |(Twitter)|MMD (degree)|MMD (spectre)|KS (node behavior)|KS (random walk)|KS (degree centrality)|KS (betweenness centrality)|
> |-|-|-|-|-|-|-|
> |MoDiff|1.0128|0.6621|1.0000|0.3000|1.0000|0.7288|
> |MulDyDiff|0.4062|0.3402|0.5957|0.1172|0.6489|0.6895|
>
> |(Superuser)|MMD (degree)|MMD (spectre)|KS (node behavior)|KS (random walk)|KS (degree centrality)|KS (betweenness centrality)|
> |-|-|-|-|-|-|-|
> |MoDiff|0.9396|0.2255|1.0000|0.0500|1.0000|0.1069|
> |MulDyDiff|0.3389|0.3023|0.5484|0.1684|0.6411|0.6119|
>
> The part of KS in the above results is presented in Table 2 in Sec. 5 of the revised version; the part of MMD is presented in Table 11 in Appendix K of the revised version.
>
> We choose KS metrics since it better captures local and global property changes of graph sequences than MMD. According to [WWW21], the importance of KS metrics for significant graph property change detection lies in 1) assessing the discrepancy of the distributions between input and generated graph sequences, 2) capturing individual behavior of each node and joint behavior of all nodes in a graph snapshots, and 3) benefits in capturing variability or dispersion with KS tests on inter-quartile range, in which the latter two cannot be achieved by MMD.
>
> [KDD25] Yuwei Xu and Chenhao Ma. 2025. MoDiff - Graph Generation with Motif-aware Diffusion Model. In Proceedings of the 31st ACM SIGKDD Conference on Knowledge Discovery and Data Mining V.2 (KDD '25).
>
> [WWW21] Giselle Zeno, Timothy La Fond, and Jennifer Neville. Dymond: Dynamic motif-nodes network
> generative model. In Proceedings of the Web Conference 2021 (WWW ’21).

---

> ### Author Response · Authors · 2025-11-24
> **Answer to Q3**
>
> Thanks for the valuable comments. As suggested, we 1) conduct sensitivity tests with various diffusion steps and graph sizes, and 2) theoretically analyze the time complexity and explain the techniques to reduce the overhead when processing large-scale network datasets.
>
> 1. Sensitivity and scalability tests: We conduct sensitivity tests on Wiki-vote dataset with various diffusion steps and present the execution time of various diffusion steps in the following table (as shown in Table 15 in Appendix K of the revised version).
>
> |#Diffusion steps|Training time (min.)|Sampling time (min./per batch)|
> |-|-|-|
> |250|79.118|18.406|
> |500|75.787|18.251|
> |750|77.665|21.021|
> |1000|76.051|23.318|
>
> From the above results, we observe that the training time does not vary significantly since we adopt multi-step Markov transition matrices with a sampled number $s$ of diffusion steps in each iteration of the training phase. Since the denoising process iterates in all steps, the sampling time slightly increases as the number of diffusion steps increases. Nevertheless, since we generate a batch of graph sequences, processing each snapshot requires only a few seconds, which is within the acceptable range.
>
> Besides, the results of the scalability tests on the Wiki-vote dataset (with 4 layers) are presented in the following table (as shown in Table 21 in Appendix K of the revised version).
>
> |Snapshot size (per layer)|Training time (hr.)|
> |-|-|
> |50|1.0114|
> |70|1.5179|
> |100|5.7083|
> |300|5.9675|
>
> From the above results, we observe that MulDyDiff is able to process graphs with 300 nodes per layer.
>
> 2. Time complexity analysis and overhead reduction techniques: The complexity of processing a graph snapshot is $O(Sl(nd_{x}+n^{2}d_{e}))$, where $S$ denotes the number of diffusion steps; $l$ denotes the number of (graph) transformer layers; $n$ denotes the number of nodes; $d_{x}$ denotes the number of node attributes; $d_{e}$ denotes the number of edge attributes. To reduce the overhead for scaling to very large network datasets, we process the layers in parallel with cross-layer attention mechanism according to the weights determined by cross-layer predictor. In order to achieve efficiency and avoid error propagation, we aggregate the past snapshots with dynamic transition function to generate future snapshots according to clean previous snapshots instead of noisy ones with a graph transformer architecture in parallel.
> Furthermore, the cross-layer attention mechanism in MulDyDiff can adopt sparse attention in convolutional transformer layers and message-passing with a random attention mechanism in [TMLR25] to restrict attention to existing edges, reducing the time complexity of processing edges from $O(ln^{2}d_{e})$ to $O(lmd_{e})$, where $l$ denotes the number of (graph) transformer layers; $n$ denotes the number of nodes; $m$ denotes the number of edges; $d_{e}$ denotes the number of edge attributes.
>
> [TMLR25] Yiming Qin, Clement Vignac, and Pascal Frossard. "SparseDiff: Sparse Discrete Diffusion for Scalable Graph Generation." Transactions on Machine Learning Research (2025).

---

> ### Author Response · Authors · 2025-12-02
> **Answer to Q2**
>
> Thanks for the valuable suggestions. As suggested, we conduct an ablation study to observe the contribution of $\mathcal{L}_ {behavior}$ (i.e., compare the performance with and without $\mathcal{L}_ {behavior}$). Figs. 4a to 4c in Sec. 5.4 present the results of $R(t)$ curves on the Wiki-vote, Twitter, and Superuser datasets with $\mathcal{L}_ {behavior}$ (red) and without $\mathcal{L}_ {behavior}$ (green). On the Wiki-vote dataset, MulDyDiff is able to capture the abrupt increase in $R(t)$ at $t=13$ of the attribute "receive" in the layer "support" on Wiki-vote dataset. In contrast, the curve of $R(t)$ obtained without $\mathcal{L}_ {behavior}$ indicates unstable and over-reactive updates of $R(t)$. This manifests the contribution of $\mathcal{L}_ {behavior}$ to capturing emergent behaviors. On the Superuser and Twitter datasets, the red curves exhibit sharper peaks coinciding with the timestamps where emergent behaviors occur. In contrast, the green curves remain relatively bounded and fail to reflect these sudden changes.

---

### Official Review · Reviewer_Mm6d · 2025-11-02

**Soundness:** 3
**Presentation:** 3
**Contribution:** 2
**Rating:** 4
**Confidence:** 4

**Summary:**

This paper introduces MulDyDiff, a diffusion-based framework for generating multiplex dynamic attributed networks. The model integrates attribute-aware temporal denoising, cross-layer correlation modeling, and behavior-guided regularization using Kuramoto-based synchronization descriptors. The approach is technically interesting and addresses an important gap in dynamic network generation, extending diffusion-based methods to more complex, multi-layer temporal settings. However, the central claim that MulDyDiff can reproduce emergent, system-level behaviors (absent from prior models) is not substantiated by the empirical evaluation. The empirical results show limited and/or uneven gains across datasets/metrics, and it remains unclear whether these improvements stem from genuine modeling insights or simply increased parameter capacity. The paper would be significantly strengthened by targeted experiments that directly assess emergent behavior, a more systematic analysis of trade-offs across network measures, and clearer discussion of model parameterization. Overall, this is a promising direction, but the current paper falls short of demonstrating sufficient practical or conceptual impact.

**Strengths:**

The paper is technically well written and motivated by a clear limitation in current dynamic graph generators. The incorporation of cross-layer coupling and behavior-aware loss is novel, and the empirical setup includes multiple datasets and ablations.

**Weaknesses:**

- Unsubstantiated core claim. The paper’s main claim (that existing methods fail to reproduce emergent behaviors such as explosive synchronization or hysteresis) is not evaluated. The reported metrics (e.g., KS distance) capture temporal distributional differences but, because they (a) consider distributions over the full graph and (b) are averaged across time, they are unlikely to detect failures in system-level emergent dynamics. If the authors believe such behavior explains the gains in Table 1, a qualitative exploration of specific graph transitions that MulDyDiff captures (but baselines miss) would strengthen the case. Otherwise, evaluation using new metrics explicitly designed to quantify emergent behavior is needed.
- Limited and uneven empirical support. The model is compared to baselines on only two of the three datasets, and results are mixed: gains on some measures (node behavior, BC) come at the expense of degradation on others (RW). Without deeper analysis of these trade-offs, it is unclear whether the improvements reflect genuine modeling benefits or selective optimization of certain graph statistics.
- Model complexity. The cross-layer coupling and behavior-guided components substantially increase model parametrization, yet there is no comparison of parameter counts, asymptotic complexity, or parameter efficiency relative to baselines. The modest empirical improvements could simply stem from higher model capacity. This is not inherently a flaw, but given that the gains are not consistent across all network metrics, it remains unclear how much of the improvement derives from MulDyDiff’s conceptual innovations versus its expanded parameterization.

**Questions:**

See weaknesses.

---

> ### Author Response · Authors · 2025-11-26
> **Answer to Q3**
>
> Thanks for the valuable comments. As suggested, we 1) analyze the parameter count of MulDyDiff and compare with baselines, and 2) conduct sensitivity (and ablation) tests on the third dataset (Wiki-vote) with various numbers of model parameters to show how the number of model parameters affect the performance.
> 1. Analysis of parameter count.
> The parameter counts of MulDyDiff and the baselines are presented in the following table. MulDyDiff requires $O(Sl(nd_{x}+n^{2}d_{e}))$ to process graph snapshots (in parallel with the graph transformer architecture), where $S$ denotes the number of diffusion steps; $l$ denotes the number of (graph) transformer layers; $n$ denotes the number of nodes; $d_{x}$ denotes the number of node attributes; $d_{e}$ denotes the number of edge attributes. MoDiff [KDD2025] requires $O(n^{2})$ for Hermitian encoding, $O(kn^{2})$ to perform spectral decomposition with $k$ eigenvalues selected, and $O(Sln^{2}d)$ for a $l$-layer GCN model denoising a $d$-dimensional features of $n$ nodes with $S$ diffusion steps. DAMNETS [ICML22] requires $O(n)$ to process an adjacency row for each of $T$ snapshots, and each node has a $d$-dimensional embedding. DYMOND [WWW21] requires $O(n^{3}T)$ to scan 3-node motifs in each of $T$ snapshots. TagGen [KDD20] requires $O(n^{2}T^{2})$ to process random-walk sequences of length $T$ with bi-level self-attention. AGE requires $O(n^{2}dl)$ for  $l$ self-attention layers in the encoder. Although DAMNETS has lower complexity than MulDyDiff, DAMNETS is insufficient to capture cross-layer dependency and graph-level behaviors.
>
> |Model/Algorithm|Parameter count|
> |-|-|
> |MulDyDiff (Ours)|$O(Sl(nd_{x}+n^{2}d_{e}))$|
> |MoDiff [KDD2025]|$O(kn^{2}+Sln^{2}d)$|
> |DAMNETS [ICML22]|$O(Tnd)$|
> |DYMOND [WWW21]|$O(n^{3}T)$|
> |TagGen [KDD20]|$O(n^{2}T^{2})$|
> |AGE [PAKDD20]|$O(n^{2}dl)$|
>
> The above analysis is presented in Table 8 in Appendix I of the revised version.
>
> [KDD25] Yuwei Xu and Chenhao Ma. 2025. MoDiff - Graph Generation with Motif-aware Diffusion Model. In Proceedings of the 31st ACM SIGKDD Conference on Knowledge Discovery and Data Mining V.2 (KDD '25).
>
> [ICML22] J. Clarkson, M. Cucuringu, A. Elliott, and G. Reinert. (2022). DAMNETS: A Deep Autoregressive Model for Generating Markovian Network Time Series. Proceedings of the First Learning on Graphs Conference, in Proceedings of Machine Learning Research.
>
> [WWW21] Giselle Zeno, Timothy La Fond, and Jennifer Neville. Dymond: Dynamic motif-nodes network generative model. In Proceedings of the Web Conference 2021 (WWW ’21).
>
> [KDD20] Dawei Zhou, Lecheng Zheng, Jiawei Han, and Jingrui He. 2020. A Data-Driven Graph Generative Model for Temporal Interaction Networks. In Proceedings of the 26th ACM SIGKDD International Conference on Knowledge Discovery & Data Mining (KDD '20).
>
> [PAKDD20] Shuangfei Fan and Bert Huang. Attention-Based Graph Evolution. Advances in Knowledge Discovery and Data Mining, 12084:436, 2020.
>
> 2. Sensitivity test.
> We conduct the sensitivity tests on Wiki-vote to evaluate the performance under various numbers model parameters, which are determined by the number of graph transformer layers, in the following table (as shown in Tables 13 and 14 in Appendix K of the revised version).
>
> |#GT-layers|MMD (degree)|MMD (clustering coefficient)|MMD (spectre)|KS (node behavior)|KS (random walk)|KS (page rank)|KS (degree centrality)|KS (betweenness centrality)|KS (closeness centrality)|
> |-|-|-|-|-|-|-|-|-|-|
> |2|0.3897|0.9233|0.3768|0.9140|0.6937|0.5859|0.8281|0.6562|0.9140|
> |4|0.3764|1.0734|0.3721|0.9296|0.6843|0.5742|0.8710|0.6523|0.9335|
> |6|0.3923|0.8955|0.3818|0.9218|0.6968|0.5820|0.8398|0.6523|0.9218|
> |8|0.4166|1.1583|0.4117|0.9765|0.7281|0.5937|0.9609|0.6640|0.9765|
>
> From the above results, we observe that the performance obtained with 2~6 graph transformer layers is comparable, but an excessive number of graph transformer layers may deteriorate the performance due to overfitting [ICLR23]. The advantages of MulDyDiff lie on the capability of capturing cross-layer dependencies cross-layer coupling with graph-level behaviors with behavioral-aware guidance instead of massive parameters.
>
> [ICLR23] Haiteng Zhao, et al. "Are More Layers Beneficial to Graph Transformers?." The Eleventh International Conference on Learning Representations.

---

> ### Author Response · Authors · 2025-12-02
> **Answer to Q1**
>
> Thanks for the valuable comments. As suggested, we conduct the evaluations using the synchronization degree $R(t)$ to assess explosive synchronization in graph sequences generated by MulDyDiff and the baselines in Figs. 3a to 3c in Sec. 5.4. Across the three datasets, MulDyDiff (red) exhibits clear and sharp peaks corresponding to explosive increases of $R(t)$, indicating that MulDyDiff can generate node attributes that faithfully reflect emergent synchronization when trained with $\mathcal{L}_{behavior}$. In contrast, methods with node attributes that do not change over time, such as AGE (orange) and DAMNETS (purple), produce flat curves with constant values of $R(t)$, showing that static attributes cannot trigger emergent synchronization. Methods with time-varying node attributes, such as MoDiff (blue), yield strongly oscillatory $R(t)$ curves but without clear explosive peaks, suggesting that simply perturbing node attributes over time is insufficient to capture emergent behaviors. Finally, TagGen (green) and DYMOND (black), which do not model node attributes and instead assign them randomly in post-processing, only display random fluctuations in $R(t)$ without any pronounced bursts.

---

> ### Author Response · Authors · 2025-12-03
> **Answer to Q2**
>
> The results on the Wiki-vote dataset (with an additional baseline MoDiff [KDD25]) are listed in the following table. On the Wiki-vote dataset, MulDyDiff outperforms the baselines in almost all metrics listed in the table, as it captures structural and attributive evolution simultaneously. Some baselines perform slightly better in the KS of random walk on the Wiki-vote dataset. Nevertheless, MulDyDiff overall outperforms these baselines since they only perform well in one or two MMD or KS metrics. This is insufficient to demonstrate the effectiveness of the baselines in multi-layer temporal graph generation, as effectiveness needs to be assessed comprehensively by various metrics.
>
> |Models|MMD (degree)|MMD (spectre)|KS (node behavior)|KS (random walk)|KS (degree centrality)|KS (betweenness centrality)|
> |-|-|-|-|-|-|-|
> |AGE|0.5145|0.3395|0.8052|0.0766|0.9991|0.8592|
> |DAMNET|0.3904|0.6162|0.6853|0.1805|0.7152|0.7318|
> |TagGen|0.6365|0.4231|0.9500|0.1400|0.8750|0.8500|
> |DYMOND|0.5384|0.3069|0.8256|0.2111|0.8398|0.8941|
> |MoDiff|0.9919|0.4127|1.0000|0.4000|0.9474|0.8500|
> |MulDyDiff|0.3676|0.3214|0.5430|0.2219|0.8281|0.6563|
>
> The part of KS in the above results is presented in Table 2 in Sec. 5 of the revised version; the part of MMD is presented in Table 11 in Appendix K of the revised version.
>
> [KDD25] Yuwei Xu and Chenhao Ma. 2025. MoDiff - Graph Generation with Motif-aware Diffusion Model. In Proceedings of the 31st ACM SIGKDD Conference on Knowledge Discovery and Data Mining V.2 (KDD '25).

---

### Official Review · Reviewer_mWRo · 2025-11-08

**Soundness:** 1
**Presentation:** 2
**Contribution:** 2
**Rating:** 2
**Confidence:** 3

**Summary:**

The authors of the work propose a method to generate synthetic dynamic multiplex graphs, where nodes are assumed to have time-varying attributes that influence the link generation in multiple layers. The proposed diffusion-based model considers dynamic node attributes, models correlation of edges across layers as well as desired emergent behavior based on the order parameter of the Kuramoto model for self-organized synchronization. The denoising process progressively generates subsequent network snapshots that are conditioned on all previous snapshots of the dynamic graph, thus giving rise to a sequence of snapshots of a discrete-time temporal graph.

The method is evaluated in three data sets on multiplex graphs from Wikipedia, Twitter, and the Superuser forum. For two of these data sets, the authors compare the resulting distributions of structural metrics (e.g. centralities) to the ground truth and benchmark their approach in comparison to four baseline temporal graph generators. The results show moderate improvements in the chosen metrics. An ablation study investigates the impact of different model components in the third data sets, which suggests that both the temporal link generation and the cross-layer correlation mechanism contribute to the results.

**Strengths:**

- [S1] The paper considers a diffusion-based generative model for temporal graphs, which could have interesting applications in practice.

- [S2] The paper considers the generatiion of multiplex attributed graphs, which have many applications.

- [S3] The generation process consider the emergent behavior of a synchronization process in the generated graphs, which is a potentially interesting idea.

**Weaknesses:**

- [W1] The practical motivation of generating temporal multiplex networks that exhibit explosive synchronization under the Kuramoto model is weak, see Q1.

- [W2] The model decription is hard to follow and some aspects of the generation process are unclear and lack intution, see my detailed comments in Q2 - Q3 below.

- [W3] The dicussion of related works is weak and the contribution over prior works on generative temporal graph models is unclear, see my detailed question Q4.

- [W4] The experimental evaluation only uses two of the three mentioned data set, and the ablation study is conducted on a third data set, but not the first two. Evaluation metrics are not properly motivated and some of them are not explained and defined. It is unclear how hyperparameters for the main results have been chosen (and even what was their value). The experimental evaluation is generally weak and does not support the claims of the paper, see detailed comments in Q5.

**Questions:**

- [Q1] While I like the general idea, I did not get the practical motivation of the behavior guidance, i.e. why do we want to generate temporal graphs that specifically exhibit a certain synchronization behavior in the Kuramoto model in the first place? Could the authors explain this motivation of their work better?

Also, the motivation mentions hysteresis as another important emergent phenomenon, but as far as I see this is actually not considered in the paper. How are hystereses effects considered by your model? I see that there are comments on hysteresis in the Kuramoto model in appendix G2 but I think it should be more clearly stated how this relates to the hysteresis example in the motivation (I think that the relation is weak at best).

Finally, the motivation suggests that multi-layer structure is a precondition for explosive synchronization and hysteresis, which is not the case as there many examples for such effects in dynamical systems on non-multiplex graphs as well. I think this should be reformulated.

- [Q2] From the rather opaque description of the attributed evolution-aware forward process in section 4.1 I could not follow what kind of temporal patterns this part of the model is actually able to capture. Also, I find the notation hard to parse, which does not help to appreciate the underlying ideas. Explaining the meaning of variable s = 1, ..., S early on would also help the reader understand the process. The same holds for the description of the section 4.2, which simply gives the equations without much additional explanations.

Moreover, I have some questions about the cross-layer correlation aware denoising, which builds on a time-evolving cross-layer coupling graph which captures edges between nodes in different layers. What kind of cross-layer correlations does this capture? It seems that those are necessarily based on inter-layer links between specific nodes, which may miss more subtle patterns (e.g. nodes in different communities in different layers that are preferentially connected). Could the authors clarify this?

I would generally recommend that the authors include an intuitive description of what the diffusion process actually captures and then potentially move part of the specific mathematical formulation to the appendix?

- [Q3] Similar to my comments in Q2, from the description in section 4.2.3 I could not follow how the behavior-guidance is actually done. Since this is a major point that could point to a more fundamental misunderstanding of the authors' work, I have structured this in a separate question.

In particular, the mathematical formulation in eq. 11 and 12 seems to suggest that we actually need to simulate the Kuramoto model to obtain the order parameter, which is then included in the diffusion model. However, this also requires us to set timescales for the simulation, i.e. at what speed does the (continuous-time) model evolve compared to the speed of the evolution of the temporal graph. This is not discussed in the paper, but I believe it is an important point that must be clarified.

Also, instead of actually simulation the model wouldn't it be possible to use spectral properties to implement the behavior guidance, e.g. the eigenration of the Laplacian matrix which is known to determine the synchronization behavior of Kuramoto oscillators in graphs?

- [Q4] I think it would greatly help the motivation of this work, if the authors could clearly work out a research gap that this work addresses. To this end, the current very brief description of related works is insufficient and - in my view - partly misleading. As an example, in the remark at the end of section 4.1 the authors state that:

"Unlike prior temporal graph generators (Campbell et al., 2024; Liu & Sariyuce, 2023; Wang et al., 2022; Gupta et al., 2022; Zeno et al., 2021), our forward process explicitly encodes temporal dependencies between the current and previous timestamps."

It sounds curious that prior works on temporal graph generators did not consider dependencies between the current and past timesteps of a graph (which is the key point in modelling temporal graphs) and indeed a quick check reveals that some of the cited works include a conditioning on the previous timestamp much in the same way as the present work. I thus believe that the statement above is too general and kindly ask the authors to clarify the contribution over these prior works.

- [Q5] I was very confused by the experimental evaluation, which mentions three data sets and then only presents results on two of them (the third one only being used for the ablation study). Why did you not include results for the third data set.

Also, it is not clear to me how the evaluation metrics have been chosen (why do we care about the KS-statistic between centrality distributions). Finally, it is not even clear what some of the metrics mean (node behavior, Random Walk) and how they are defined. This must be clarified before this paper can be published.

Also, in the sensitivity analysis the authors check the impact of the hyperparameters \gamma_0 and sequence length, but the values used for the main results are not mentioned (and it is not made clear whether these hyperparameters were optimized). Also, the model formulation includes another parameter (noise strength) which is not mentioned in the experimental evaluation. This must be clarified.

---

> ### Author Response · Authors · 2025-11-24
> **Answer to Q2 (Part 1)**
>
> Thank you for the valuable comments. As suggested, we (1) first explain the notations, (2) clarify that our attributed evolution-aware forward process is designed to capture long-range temporal coupling and continuity, with detailed formula explanations, (3) clarify that our correlation-aware denoising captures not only explicit inter-layer links but also implicit co-evolution via an attention-based predictor, again with detailed formula explanations, allowing one layer to assist another even without observed inter-layer edges, and finally (4) provide a concise and intuitive description of our model.
>
> 1. Explanation of the notations.
> * We first define the notations in the following table (added in Table 5 in Appendix C of the revised version):
> |Notation|Meaning|
> |-|-|
> |$s=0, 1,\ldots,S$|diffusion steps|
> |$t=0, 1,\ldots,T$|timestamps|
> |$l=1,\ldots,L$|layers|
> |$\mathbf{X}_{l,t}^{(s)} \in \mathbb{R}^{a \times N}$|the diffused node representation at step $s$ with $a$ attributes of $N$ nodes in layer $l$ at timestamp $t$|
> |${\alpha}_{s}$|a parameter that controls noise strength, defining how fast information is washed out by noise along the diffusion axis $s$; $\overline{\alpha}_ {s}=\prod_{i=1}^{s}{\alpha}_{i}$|
> |${\gamma}_{s}$|a parameter that controls the dependence on previous snapshots, specifying how strong the temporal smoothing is along the time axis $t$; $\overline{\gamma}_ {s}=\prod_{i=1}^{s}{\gamma}_{i}$|
> |$\overline{\mathbf{Q}}_ {\mathbf{X}_{l,(0:t-1)}}^{(s)}$|multi-step Markov transition matrix that transits $\mathbf{X}_ {l,t}^{(0)}$ to $\mathbf{X}_{l,t}^{(s)}$|
> |$\mathbf{Q}_ {\mathbf{X}_{l,(0:t-1)}}^{(s)}$|single-step Markov transition matrix that transits $\mathbf{X}_ {l,t}^{(s-1)}$ to $\mathbf{X}_ {l,t}^{(s)}$|
>
> 2. Temporal patterns: Capturing long-range temporal coupling and continuity.
> * Intuition: Unlike standard temporal models that rely on one-step Markov dependencies (conditioning only on $G_ {t-1}$), our forward process explicitly incorporates the entire history $G_ {0:t-1}$ when defining the distribution at time $t$. This is achieved through the attribute-aware dynamic transition mixture $\text{D}$ in Eq. (1):
> $$\text{D}(\mathbf{X}^{(0)}_ {l,(0:t)}, \bar{\gamma}_ {s}) = \bar{\gamma}_ {s} \mathbf{X}^{(0)}_ {l,t} + (1-\bar{\gamma}_ {s})\text{D}(\mathbf{X}^{(0)}_ {l,(0:t-1)}, \bar{\gamma}_ {s}),$$which recursively accumulates clean snapshots from all previous timestamps. As a result, the forward prior for $\mathbf{X}^{(s)}_ {l,t}$ is not a local variation of $\mathbf{X}^{(0)}_ {l,t-1}$ but a history-mixed representation that embeds long-range temporal signals. This design has two key benefits: i) it enables the reverse denoising network to perform history-guided denoising, capturing persistent temporal structures and long-range dependencies that one-step models miss; and ii) since the mixture is formed from clean states $\mathbf{X}^{(0)}$, it mitigates the error propagation issue of autoregressive temporal generators that repeatedly condition on noisy predictions.
> * Derivation logic: We aim to capture long-range temporal coupling and continuity in temporal graph sequences, which is achieved by the derivation of the temporal-aware diffusion model with the following logic flow:
>     1) We first define a temporal aggregation function (Eq. (1) of the revised version) that summarizes all past clean snapshots using an exponentially weighted mixture. This establishes how temporal information from earlier timestamps is incorporated into the model.
>     2) We then inject noise in a temporally consistent way (Eqs. (2)-(3) of the revised version) by blending local diffusion at time $t$ with the diffused representation at time $t−1$. This step defines how attribute noise interacts with temporal smoothness.
>     3) We show that this forward process admits a closed-form expression (Eq. (4) of the revised version), which explicitly reveals the influence of the entire history.
>     4) We reinterpret the closed form of the forward process as a Markov transition (Eq. (5) of the revised version), clarifying how each diffusion step decomposes into self-preservation, history-driven drift, and uniform noise injection.
>     5) We decompose the multistep transition into single-step transitions (Eqs. (6)–(7) of the revised version) to make posterior inference tractable.
>     6) We derive the exact posterior for reverse diffusion (Eq. (9) of the revised version), enabling us to compute the probability of the previous noisy state conditioned on the current one.
>     7) Finally, we approximate the reverse process with a learned denoiser (Eq. (10) of the revised version), which maps noisy states back to clean states in a history-aware manner.
> Together, these steps establish a temporally coherent forward diffusion process whose reverse process can reconstruct each snapshot using the entire clean history, enabling the model to capture long-range temporal dependencies rather than purely local transitions.

---

> ### Author Response · Authors · 2025-11-24
> **Answer to Q2 (Part 2)**
>
> 2. Temporal patterns: Capturing long-range temporal coupling and continuity. (Continued)
> * Derivation details
>     * (1) To incorporate information from all historical snapshots up to time $t$, we define a recursive temporal aggregation function. The idea is to let the most recent snapshot contribute most strongly, while earlier snapshots contribute with exponentially decaying weights. Given historical snapshots $\mathbf{X}_ {l,(0:t-1)}^{(0)}$ and the current snapshot $\mathbf{X}_ {l,t}^{(0)}$, we define the dynamic transition in Eq. (1) of the revised version by combining the current snapshot at timestamp $t$ and the dynamic transition over past snapshots from timestamp $0$ to $t-1$ as follows:
> \begin{align}
> \mathrm{D}(\mathbf{X}_ {l,(0:t)}^{(0)},\overline{\gamma}_ {s}) &= \overline{\gamma}_ {s}\mathbf{X}_ {l,t}^{(0)}+(1-\overline{\gamma}_ {s})\mathrm{D}(\mathbf{X}_ {l,(0:t-1)}^{(0)},\overline{\gamma}_ {s}),\quad\text{(1)}
> \end{align}
> with the initial condition $\mathrm{D}(\mathbf{X}_ {l,0}^{(0)},\overline{\gamma}_ {s}) = \mathbf{X}_ {l,0}^{(0)}$.
> This recursion $\mathrm{D}$ means that each historical snapshot influences the aggregated representation, but with strength controlled by $\overline{\gamma}_ {s}$. A larger $\overline{\gamma}_ {s}$ prioritizes the current snapshot, while a smaller one increases the influence of earlier snapshots.
>     * (2) In the traditional diffusion process, the diffused snapshot $\mathbf{Y}_ {l,t}^{(s)}$ in the $s$-th step of the snapshot $\mathbf{X}_ {l,t}^{(0)}$ is $\mathbf{Y}_ {l,t}^{(s)}=\overline{\alpha}_ {s}\mathbf{X}_ {l,t}^{(0)}+(1-\overline{\alpha}_ {s})\tfrac{\overline{\mathbf{1}}_ {a}}{a}$ at timestamp $t$. To generate temporally consistent noisy representations $\mathbf{X}_ {l,t}^{(s)}$, we blend the locally diffused state $\mathbf{Y}_ {l,t}^{(s)}$ at time $t$ with the diffused state $\mathbf{X}_ {l,t-1}^{(s)}$ from the previous timestamp $t-1$, which ensures smoothness across time. Thus, we define the dynamic transition-aware forward process in Eq. (2) of the revised version:
> \begin{align}
> \mathbf{X}_ {l,t}^{(s)} &= \overline{\gamma}_ {s}\mathbf{Y}_ {l,t}^{(s)}+(1-\overline{\gamma}_ {s})\mathbf{X}_ {l,t-1}^{(s)}\\\\
> &= \overline{\gamma}_ {s}(\overline{\alpha}_ {s}\mathbf{X}_ {l,t}^{(0)}+(1-\overline{\alpha}_ {s})\tfrac{\overline{\mathbf{1}}_ {a}}{a})+(1-\overline{\gamma}_ {s})\mathbf{X}_ {l,t-1}^{(s)}, \forall s, t \geq 1,\quad\text{(2)}
> \end{align}
> where the first term in Eq. (2) adds noise to the current snapshot, while the second term propagates temporal influence forward from $t−1$. The parameter $\overline{\gamma}_ {s}$ adjusts the balance: larger values emphasize the current snapshot; smaller values enforce stronger temporal continuity.
> Since $t=0$ is the start point, we initialize its forward diffusion to be the same as the traditional diffusion process in Eq. (3) of the revised version.
> \begin{align}
> \mathbf{X}_ {l,0}^{(s)} =\mathbf{Y}_ {l,0}^{(s)} = \overline{\alpha}_ {s}\mathbf{X}_ {l,0}^{(0)}+(1-\overline{\alpha}_ {s})\tfrac{\overline{\mathbf{1}}_ {a}}{a},\quad\text{(3)}
> \end{align}
> where $\overline{\mathbf{1}}_ {a} \in \mathbb{R}^{N \times a}$ denotes all-one matrix. The weight $\overline{\alpha}_ {s}$ determines how much the original structure is retained.
>     * (3) By expanding the recursive temporal-aware forward equations (Eq. (2) with induction on $t$), we obtain a direct relationship between the $s$-step noisy snapshot and all historical clean snapshots, explicitly incorporating the entire history $\mathbf{X}_ {l,(0:t)}^{(0)}$ in Eq. (4) of the revised version as follows:
> \begin{align}
> \mathbf{X}_ {l,t}^{(s)} = \overline{\alpha}_ {s}\mathrm{D}(\mathbf{X}_ {l,(0:t)}^{(0)},\overline{\gamma}_ {s})+(1-\overline{\alpha}_ {s})\tfrac{\overline{\mathbf{1}}_ {a}}{a},\quad\text{(4)}
> \end{align}
> Thus, the noisy snapshot is a mixture of a history-aggregated signal and a uniform noise baseline. As $\overline{\alpha}_ {s}$ decreases, the influence of the uniform noise grows, gradually removing temporal structure.

---

> ### Author Response · Authors · 2025-11-24
> **Answer to Q2 (Part 3)**
>
> 2. Temporal patterns: Capturing long-range temporal coupling and continuity. (Continued)
> * Derivation details (Continued)
>     * (4) We express Eq. (4) as a Markov transition that changes the state of $\mathbf{X}_ {l,t}^{(s)}$, through Eq. (1)
>     \begin{align}
>     \mathbf{X}_ {l,t}^{(s)} &= \overline{\alpha}_ {s}\overline{\gamma}_ {s}\mathbf{X}_ {l,t}^{(0)}+\overline{\alpha}_ {s}(1-\overline{\gamma}_ {s})\mathrm{D}(\mathbf{X}_ {l,(0:t-1)}^{(0)},\overline{\gamma}_ {s})+(1-\overline{\alpha}_ {s})\tfrac{\overline{\mathbf{1}}_ {a}}{a}\quad\text{(By Eq. (1))},\\\\
>     &=(\overline{\alpha}_ {s}\overline{\gamma}_ {s}\mathbf{I}+\overline\alpha_ {s}(1-\overline{\gamma}_ {s})\text{D}(\mathbf{X}_ {l,(0:t-1)}^{(0)},\overline{\gamma}_ {s})\overline{\mathbf{1}}_ {a}^{\top}+(1-\overline\alpha_ {s})\overline{\mathbf{1}}_ {a}\tfrac{\overline{\mathbf{1}}^{\top}_ {a}}{a})\mathbf{X}_ {l,t}^{(0)}\\\\
>     &(\because \overline{\mathbf{1}}_ {a}\overline{\mathbf{1}}^{\top}_ {a}\mathbf{X}_ {l,t}^{(0)}=\overline{\mathbf{1}}_ {a},\text{D}(\mathbf{X}_ {l,(0:t-1)}^{(0)},\overline{\gamma}_ {s})\overline{\mathbf{1}}_ {a}^{\top}\mathbf{X}_ {l,t}^{(0)}=\text{D}(\mathbf{X}_ {l,(0:t-1)}^{(0)},\overline{\gamma}_ {s}))\\\\
>     &=(\overline{\alpha}_ {s}\overline{\gamma}_ {s}\mathbf{I}+\overline\alpha_ {s}(1-\overline{\gamma}_ {s})\overline{\mathbf{1}}_ {a}\text{D}(\mathbf{X}_ {l,(0:t-1)}^{(0)},\overline{\gamma}_ {s})^{\top}+(1-\overline\alpha_ {s})\overline{\mathbf{1}}_ {a}\tfrac{\overline{\mathbf{1}}^{\top}_ {a}}{a})^{\top}\mathbf{X}_ {l,t}^{(0)}\quad\text{(Transpose)}\\\\
>     &={{\overline{\mathbf{Q}}_ {\mathbf{X}_ {l,(0:t-1)}}^{(s)}}^{\top}}\mathbf{X}_ {l,t}^{(0)},\\\\
>     \end{align}
>     where $\overline{\mathbf{Q}}_ {\mathbf{X}_ {l,(0:t-1)}}^{(s)} = \overline\alpha_ {s}\overline{\gamma}_ {s}\mathbf{I}+\overline\alpha_ {s}(1-\overline{\gamma}_ {s})\mathbf{1}_ {a}\text{D}(\mathbf{X}_ {l,(0:t-1)}^{(0)},\overline{\gamma}_ {s})^{\top}+(1-\overline\alpha_ {s})\mathbf{1}_ {a}\tfrac{\overline{\mathbf{1}}^{\top}_ {a}}{a}$ is a Markov transition matrix decomposing the diffusion into three intuitive effects: (1) self-preservation (i.e., staying in the same state), (2) drifting toward the aggregated history $D(\cdot)$, and (3) injecting uniform noise.
>     Thus, we rewrite Eq. (4) using the Markov transition matrix $\overline{\mathbf{Q}}_ {\mathbf{X}_ {l,(0:t-1)}}^{(s)}$ as Eq. (5) of the revised version:
> \begin{align}
> q(\mathbf{X}_ {l,t}^{(s)} | \mathbf{X}_ {l,(0:t)}^{(0)}) = Cat(\mathbf{X}_ {l,t}^{(s)};{{\overline{\mathbf{Q}}_ {\mathbf{X}_ {l,(0:t-1)}}^{(s)}}^{\top}}\mathbf{X}_ {l,t}^{(0)})={\mathbf{X}_ {l,t}^{(s)}}^{\top}{{\overline{\mathbf{Q}}_ {\mathbf{X}_ {l,(0:t-1)}}^{(s)}}^{\top}}\mathbf{X}_ {l,t}^{(0)}.\quad\text{(5)}
> \end{align}

---

> ### Author Response · Authors · 2025-11-24
> **Answer to Q2 (Part 4)**
>
> 2. Temporal patterns: Capturing long-range temporal coupling and continuity. (Continued)
> * Derivation details (Continued)
>     * (5) To derive the posterior and reverse process, we rewrite the multi-step update as a single-step Markov transition. By subtracting $\alpha_ {s}\gamma_ {s}\mathbf{X}_ {l,t}^{(s-1)}$ from $\mathbf{X}_ {l,t}^{(s)}$ (using Eq. (4) to cancel out $\mathbf{X}_ {l,t}^{(0)}$), we obtain the single-step forward process in Eq. (6) of the revised version:
> \begin{align}
> &\mathbf{X}_ {l,t}^{(s)}=\alpha_ {s}\gamma_ {s}\mathbf{X}_ {l,t}^{(s-1)}+\overline{\alpha}_ {s}[(1-\overline{\gamma}_ {s})\mathrm{D}(\mathbf{X}_ {l,(0:t-1)}^{(0)},\overline{\gamma}_ {s})\\\\
> &-(\gamma_ {s}-\overline{\gamma}_ {s})\mathrm{D}(\mathbf{X}_ {l,(0:t-1)}^{(0)},\overline{\gamma}_ {s-1})]+[1-\alpha_ {s}\gamma_ {s}-\overline{\alpha}_ {s}(1-\gamma_ {s})]\tfrac{\overline{\mathbf{1}}_ {a}}{a},\quad\text{(6)}
> \end{align}
> in which the three parts correspond to: (1) keeping part of the previous noisy state via $\alpha_ {s}\gamma_ {s}\mathbf{X}_ {l,t}^{(s-1)}$, (2) adjusting toward the history-consistent direction implied by the multi-step dynamics (the two $\mathrm{D}(\cdot)$ terms ensure that the single-step behavior matches the $s$-step closed form), and (3) injecting uniform noise to maintain stochasticity and preserve a valid categorical distribution. This decomposition reveals how the model preserves previous noise, incorporates temporal structure, and adds randomness.
>         * Similar to Eq. (5), we have Eq. (7) of the revised version as follows:
>     \begin{align}
> q(\mathbf{X}_ {l,t}^{(s)}|\mathbf{X}_ {l,t}^{(s-1)},\mathbf{X}_ {l,(0:t)}^{(0)}) = Cat(\mathbf{X}_ {l,t}^{(s)};{{\mathbf{Q}_ {\mathbf{X}_ {l,(0:t-1)}}^{(s)}}^{\top}}\mathbf{X}_ {l,t}^{(s-1)})={\mathbf{X}_ {l,t}^{(s)}}^{\top}{{\mathbf{Q}_ {\mathbf{X}_ {l,(0:t-1)}}^{(s)}}^{\top}}\mathbf{X}_ {l,t}^{(s-1)},\quad\text{(7)}
> \end{align}
> where $\mathbf{Q}_ {\mathbf{X}_ {l,(0:t-1)}}^{(s)}= \overline\alpha_ {s}\overline{\gamma}_ {s}\mathbf{I}+\overline\alpha_ {s}(1-\overline{\gamma}_ {s})\mathbf{1}_ {a}\text{D}(\mathbf{X}_ {l,(0:t-1)}^{(0)},\overline{\gamma}_ {s})^{\top}+(1-\overline\alpha_ {s})\mathbf{1}_ {a}\tfrac{\overline{\mathbf{1}}^{\top}_ {a}}{a}$ is a Markov transition matrix encoding: (1) self-preservation, (2) a shift toward the aggregated history $D(\cdot)$, and (3) movement toward the uniform noise baseline.
>         * It is worthy to note that $\mathbf{Q}_ {\mathbf{X}_ {l,(0:t-1)}}^{(s)}$ is a transition matrix satisfying the property of Markov chain, i.e., $\overline{\mathbf{Q}}_ {\mathbf{X}_ {l,(0:t-1)}}^{(s-1)}\mathbf{Q}_ {\mathbf{X}_ {l,(0:t-1)}}^{(s)}=\overline{\mathbf{Q}}_ {\mathbf{X}_ {l,(0:t-1)}}^{(s)}$. Thus, the distribution $q(\mathbf{X}_ {l,t}^{(s)}|\mathbf{X}_ {l,(0:t)}^{(0)})$ can be marginalized by $q(\mathbf{X}_ {l,t}^{(s)}|\mathbf{X}_ {l,t}^{(s-1)}\mathbf{X}_ {l,(0:t)}^{(0)})$ and $q(\mathbf{X}_ {l,t}^{(s-1)}|\mathbf{X}_ {l,(0:t)}^{(0)})$ as follows:
>     \begin{align}
> &q(\mathbf{X}_ {l,t}^{(s)}|\mathbf{X}_ {l,(0:t)}^{(0)})\\\\
> &=\sum\nolimits_ {\mathbf{X}_ {l,t}^{(s-1)}}q(\mathbf{X}_ {l,t}^{(s)},\mathbf{X}_ {l,t}^{(s-1)}|\mathbf{X}_ {l,(0:t)}^{(0)})\quad\text{(Marginalization)}\\\\
> &=\sum\nolimits_ {\mathbf{X}_ {l,t}^{(s-1)}}q(\mathbf{X}_ {l,t}^{(s)}|\mathbf{X}_ {l,t}^{(s-1)},\mathbf{X}_ {l,(0:t)}^{(0)})q(\mathbf{X}_ {l,t}^{(s-1)}|\mathbf{X}_ {l,(0:t)}^{(0)})\quad\text{(By Bayesian formula)}\\\\
> &=\sum\nolimits_ {\mathbf{X}_ {l,t}^{(s-1)}}({\mathbf{X}_ {l,t}^{(s)}}^{\top}{{\mathbf{Q}_ {\mathbf{X}_ {l,(0:t-1)}}^{(s)}}^{\top}}\mathbf{X}_ {l,t}^{(s-1)})({\mathbf{X}_ {l,t}^{(s-1)}}^{\top}{{\overline{\mathbf{Q}}_ {\mathbf{X}_ {l,(0:t-1)}}^{(s-1)}}^{\top}}\mathbf{X}_ {l,t}^{(0)})\quad\text{(By Eqs. (5) and (7))}\\\\
> &={\mathbf{X}_ {l,t}^{(s)}}^{\top}{{\mathbf{Q}_ {\mathbf{X}_ {l,(0:t-1)}}^{(s)}}^{\top}}\sum\nolimits_ {\mathbf{X}_ {l,t}^{(s-1)}}(\mathbf{X}_ {l,t}^{(s-1)}{\mathbf{X}_ {l,t}^{(s-1)}}^{\top}){{\overline{\mathbf{Q}}_ {\mathbf{X}_ {l,(0:t-1)}}^{(s-1)}}^{\top}}\mathbf{X}_ {l,t}^{(0)}\\\\
> &={\mathbf{X}_ {l,t}^{(s)}}^{\top}{{\mathbf{Q}_ {\mathbf{X}_ {l,(0:t-1)}}^{(s)}}^{\top}}{{\overline{\mathbf{Q}}_ {\mathbf{X}_ {l,(0:t-1)}}^{(s-1)}}^{\top}}\mathbf{X}_ {l,t}^{(0)}\quad(\because \sum\nolimits_ {\mathbf{X}_ {l,t}^{(s-1)}}(\mathbf{X}_ {l,t}^{(s-1)}{\mathbf{X}_ {l,t}^{(s-1)}}^{\top})=\mathbf{I})\\\\
> &={\mathbf{X}_ {l,t}^{(s)}}^{\top}{{\overline{\mathbf{Q}}_ {\mathbf{X}_ {l,(0:t-1)}}^{(s)}}^{\top}}\mathbf{X}_ {l,t}^{(0)}\quad(\because \overline{\mathbf{Q}}_ {\mathbf{X}_ {l,(0:t-1)}}^{(s-1)}\mathbf{Q}_ {\mathbf{X}_ {l,(0:t-1)}}^{(s)}=\overline{\mathbf{Q}}_ {\mathbf{X}_ {l,(0:t-1)}}^{(s)}),
> \end{align}
> which proves Theorem 4.1.

---

> ### Author Response · Authors · 2025-11-24
> **Answer to Q2 (Part 5)**
>
> 2. Temporal patterns: Capturing long-range temporal coupling and continuity. (Continued)
> * Derivation details (Continued)
>     * (6) To reverse the diffusion, we use Bayesian formula to compute the posterior distribution over the previous noisy state given the current noisy state and the clean history. From Eqs. (5) and (7), we derive the posterior of the forward process $q$ in Eq. (9) of the revised version as follows:
>     \begin{align}
> q(\mathbf{X}_ {l,t}^{(s-1)}|\mathbf{X}_ {l,t}^{(s)},\mathbf{X}_ {l,(0:t)}^{(0)}) &= \tfrac{q(\mathbf{X}_ {l,t}^{(s)}|\mathbf{X}_ {l,t}^{(s-1)},\mathbf{X}_ {l,(0:t)}^{(0)})q(\mathbf{X}_ {l,t}^{(s-1)}|\mathbf{X}_ {l,(0:t)}^{(0)})}{q(\mathbf{X}_ {l,t}^{(s)}|\mathbf{X}_ {l,(0:t)}^{(0)})}\quad\text{(By Bayesian formula)}\\\\
> &=\tfrac{({\mathbf{X}_ {l,t}^{(s)}}^{\top}{\mathbf{Q}_ {\mathbf{X}_ {l,(0:t-1)}}^{(s)}}^{\top}\mathbf{X}_ {l,t}^{(s-1)})({\mathbf{X}_ {l,t}^{(s-1)}}^{\top}{{\overline{\mathbf{Q}}_ {\mathbf{X}_ {l,(0:t-1)}}^{(s-1)}}^{\top}}\mathbf{X}_ {l,t}^{(0)})}{({\mathbf{X}_ {l,t}^{(s)}}^{\top}{{\overline{\mathbf{Q}}_ {\mathbf{X}_ {l,(0:t-1)}}^{(s)}}^{\top}}\mathbf{X}_ {l,t}^{(0)})}\quad\text{(By Eqs. (5) and (7))}\\\\
> &=\tfrac{({\mathbf{X}_ {l,t}^{(s-1)}}^{\top}\mathbf{Q}_ {\mathbf{X}_ {l,(0:t-1)}}^{(s)}\mathbf{X}_ {l,t}^{(s)})({\mathbf{X}_ {l,t}^{(s-1)}}^{\top}{{\overline{\mathbf{Q}}_ {\mathbf{X}_ {l,(0:t-1)}}^{(s-1)}}^{\top}}\mathbf{X}_ {l,t}^{(0)})}{({\mathbf{X}_ {l,t}^{(s)}}^{\top}{{\overline{\mathbf{Q}}_ {\mathbf{X}_ {l,(0:t-1)}}^{(s)}}^{\top}}\mathbf{X}_ {l,t}^{(0)})}\quad\text{(Transpose the first term in the numerator)}\\\\
> &={\mathbf{X}_ {l,t}^{(s-1)}}^{\top}\tfrac{\mathbf{Q}_ {\mathbf{X}_ {l,(0:t-1)}}^{(s)}\mathbf{X}_ {l,t}^{(s)} \odot {\overline{\mathbf{Q}}_ {\mathbf{X}_ {l,(0:t-1)}}^{(s-1)}}^{\top}\mathbf{X}_ {l,t}^{(0)}}{{\mathbf{X}_ {l,t}^{(s)}}^{\top}{\overline{\mathbf{Q}}_ {\mathbf{X}_ {l,(0:t-1)}}^{(s)}}^{\top}\mathbf{X}_ {l,t}^{(0)}},\quad\text{(9)}
> \end{align}
> which proves Theorem 4.2 (the closed form of the posterior). The numerator multiplies: (a) how likely each prior state leads to $X^{(s)}$, and (b) how likely it is under the multi-step history-informed prior. The denominator normalizes these weights into a valid categorical distribution.
>     * (7) Since the true clean snapshot is unknown, we approximate the reverse transition by combining the exact posterior with a learned clean-state predictor. The reverse denoising process is approximated by the posterior (Eq. (9)) in Eq. (10) of the revised version by marginalization as follows:
>     \begin{align}
> &p_ {\theta}(\mathbf{X}_ {l,t}^{(s-1)}|\mathbf{X}_ {l,t}^{(s)},\mathbf{X}_ {l,(0:t-1)}^{(0)}) \\\\
> &= \sum\nolimits_ {\mathbf{X}_ {l,t}^{(0)}}p_ {\theta}(\mathbf{X}_ {l,t}^{(s-1)},\mathbf{X}_ {l,t}^{(0)}|\mathbf{X}_ {l,t}^{(s)},\mathbf{X}_ {l,(0:t-1)}^{(0)})\quad\text{(Marginalization)}\\\\
> &= \sum\nolimits_ {\mathbf{X}_ {l,t}^{(0)}}p_ {\theta}(\mathbf{X}_ {l,t}^{(s-1)}|\mathbf{X}_ {l,t}^{(s)},\mathbf{X}_ {l,(0:t-1)}^{(0)},\mathbf{X}_ {l,t}^{(0)})p_ {\theta}(\mathbf{X}_ {l,t}^{(0)}|\mathbf{X}_ {l,t}^{(s)},\mathbf{X}_ {l,(0:t-1)}^{(0)})\quad\text{(By Bayesian formula)}\\\\
> &\approx \sum\nolimits_ {\mathbf{X}_ {l,t}^{(0)}}q(\mathbf{X}_ {l,t}^{(s-1)}|\mathbf{X}_ {l,t}^{(s)},\mathbf{X}_ {l,(0:t-1)}^{(0)},\mathbf{X}_ {l,t}^{(0)})\hat{p}_ {l,t}^{(X)}(\mathbf{X}_ {l,t}^{(0)}|\mathbf{X}_ {l,t}^{(s)},\mathbf{X}_ {l,(0:t-1)}^{(0)}),\quad\text{(10)}
> \end{align}
> where $\hat{p}_ {l,t}^{(X)}$ is learned by a denoising network (parametrized with $\theta$) denoising $\mathbf{X}_ {l,t}^{(s)}$ to the clean representation $\mathbf{X}_ {l,t}^{(0)}$ conditioned on the given historical information $\mathbf{X}_ {l,(0:t-1)}^{(0)}$, which captures long-range temporal coupling and coherence. The denoiser predicts plausible clean states, and the model averages the exact backward transitions over those predictions, yielding an effective reverse diffusion step.

---

> ### Author Response · Authors · 2025-11-24
> **Answer to Q2 (Part 6)**
>
> 3. Cross-layer patterns: Capturing implicit co-evolution via attention.
> * Intuition: Our model captures both explicit structural coupling and implicit co-evolution. We do not rely solely on static, observed inter-layer edges; instead, we leverage attention to capture latent correlations where layers evolve in synchrony without direct connections. This is achieved in the cross-layer correlation-aware denoising module (Eqs. (15) and (16) of the revised version), specifically through the predicted coupling graph $\hat{G}_ {t}^{(C)}$, which dynamically estimates cross-layer correlation strengths at each timestamp. The denoising network then uses these correlations through a cross-attention mechanism, assigning weights to other layers based on their state similarity and temporal co-evolution—even when no explicit inter-layer edge $B_ {(l,m)}$ exists (detailed in Appendix F). For example, if nodes in layer $A$ and nodes in layer B repeatedly undergo similar attribute transitions or community-level changes at the same times (a “shared temporal shock”), the attention mechanism can learn this pattern and allow layer $A$ to guide the reconstruction of layer $B$, and vice versa, despite the absence of direct inter-layer links between specific node pairs.
> * Derivation logic: We aim to capture explicit structural coupling and implicit co-evolution in multi-layer temporal graph sequences, which is achieved by multiplying the temporal-aware forward and reverse processes of all layers with the following logic flow:
>     1) The multi-step (single-step) forward process of a multi-layer temporal graph sequence is extended to Eq. (11) of the revised version from Eq. (5) (Eq. (7)) by multiplying multi-step (single-step) forward processes of all layers, which are assumed to be independent to each other.
>     2) The posterior of the forward process of a multi-layer temporal graph sequence is extended to Eq. (12) of the revised version from Eq. (9) by multiplying posteriors of all layers with Bayesian formula using Eq. (11).
>     3) The denoising distribution is derived in Eq. (13) of the revised version. It is the product of the denoising distributions of all layers, as shown in Eq. (14) of the revised version. These layer-wise distributions are learned separately and are conditionally independent given the clean snapshots from timestamps $0$ to $t-1$. Each layer's distribution is learned through a cross-layer attention mechanism. The attention weights are learned from the cross-layer coupling graph predicted by a cross-layer predictor.
>     4) The reverse process of a multi-layer temporal graph sequence is extended to Eq. (15) of the revised version by multiplying reverse processes of all layers (Eq.(16) of the revised version) that are conditionally independent given the clean snapshots $\mathbf{X}_ {(1:L),(0:t-1)}^{(0)}$ during preceding timestamps $0$ to $t-1$.
> * Derivation details:
>     * (1) We assume that the noise-injection process at diffusion step $s$ is independent across layers. Intuitively, each layer is corrupted by noise separately, while the reverse denoising process later leverages cross-layer attention to reintroduce interdependencies. The multi-step and single-step forward processes of a multi-layer temporal graph sequence is extended from Eqs. (5) and (7) as follows:
>     \begin{align}
> q(\mathbf{X}_ {(1:L),t}^{(s)}|\mathbf{X}_ {(1:L),(0:t)}^{(0)})&=\prod_ {l=1}^{L}q(\mathbf{X}_ {l,t}^{(s)}|\mathbf{X}_ {l,(0:t)}^{(0)});\\\\
> q(\mathbf{X}_ {(1:L),t}^{(s-1)}|\mathbf{X}_ {(1:L),t}^{(s)},\mathbf{X}_ {(1:L),(0:t)}^{(0)})&=\prod_ {l=1}^{L}q(\mathbf{X}_ {l,t}^{(s)}|\mathbf{X}_ {l,t}^{(s-1)},\mathbf{X}_ {l,(0:t)}^{(0)}).\quad\text{(11)}
> \end{align}
>     * (2) Due to the independence of the forward processes of all layers, the posterior of the forward process of a multi-layer temporal graph sequence is extended from Eq. (9) as follows:
> \begin{align}
> q(\mathbf{X}_ {(1:L),t}^{(s-1)}|\mathbf{X}_ {(1:L),t}^{(s)},\mathbf{X}_ {(1:L),(0:t)}^{(0)})&=\tfrac{q(\mathbf{X}_ {(1:L),t}^{(s)}|\mathbf{X}_ {(1:L),t}^{(s-1)},\mathbf{X}_ {(1:L),(0:t)}^{(0)})q(\mathbf{X}_ {(1:L),t}^{(s-1)}|\mathbf{X}_ {(1:L),(0:t)}^{(0)})}{q(\mathbf{X}_ {(1:L),t}^{(s)}|\mathbf{X}_ {(1:L),(0:t)}^{(0)})}\quad\text{(By Bayesian formula)}\\\\
> &=\tfrac{\prod_ {l=1}^{L}q(\mathbf{X}_ {l,t}^{(s)}|\mathbf{X}_ {l,t}^{(s-1)},\mathbf{X}_ {l,(0:t)}^{(0)})\prod_ {l=1}^{L}q(\mathbf{X}_ {l,t}^{(s-1)}|\mathbf{X}_ {l,(0:t)}^{(0)})}{\prod_ {l=1}^{L}q(\mathbf{X}_ {l,t}^{(s)}|\mathbf{X}_ {l,(0:t)}^{(0)})}\quad\text{(By Eq. (11))}\\\\
> &=\prod_ {l=1}^{L}\tfrac{q(\mathbf{X}_ {l,t}^{(s)}|\mathbf{X}_ {l,t}^{(s-1)},\mathbf{X}_ {l,(0:t)}^{(0)})q(\mathbf{X}_ {l,t}^{(s-1)}|\mathbf{X}_ {l,(0:t)}^{(0)})}{q(\mathbf{X}_ {l,t}^{(s)}|\mathbf{X}_ {l,(0:t)}^{(0)})}\\\\
> &=\prod_ {l=1}^{L}q(\mathbf{X}_ {l,t}^{(s-1)}|\mathbf{X}_ {l,t}^{(s)},\mathbf{X}_ {l,(0:t)}^{(0)}).\quad\text{(12)}
> \end{align}

---

> ### Author Response · Authors · 2025-11-24
> **Answer to Q2 (Part 7)**
>
> 3. Cross-layer patterns: Capturing implicit co-evolution via attention. (Continued)
> * Derivation details (Continued)
>     * (3) Formally, layers exhibit interdependencies and implicit co-evolution in a multi-layer temporal graph sequence. To model such co-evolution, since layers are conditionally independent given the clean snapshots $\mathbf{X}_ {(1:L),(0:t-1)}^{(0)}$, the denoising distribution is the product of distributions of all layers conditioned on $\mathbf{X}_ {(1:L),(0:t-1)}^{(0)}$ as in Eq. (13) of the revised version:
>     \begin{align}
> p_ {\theta}(\mathbf{X}_ {(1:L),t}^{(0)}|\mathbf{X}_ {(1:L),t}^{(s)},\mathbf{X}_ {(1:L),(0:t-1)}^{(0)})=\prod_ {l=1}^{L}p_ {\theta}(\mathbf{X}_ {l,t}^{(0)}|\mathbf{X}_ {l,t}^{(s)},\mathbf{X}_ {(1:L),(0:t-1)}^{(0)}), \text{(13)}
> \end{align}
> where $p_ {\theta}(\mathbf{X}_ {l,t}^{(0)}|\mathbf{X}_ {l,t}^{(s)},\mathbf{X}_ {(1:L),(0:t-1)}^{(0)})$ is learned by a denoising network with a cross-layer attention mechanism with weights determined by the predicted cross-layer coupling graph ${G}_ {t}^{(C)}$, enabling to capture implicit co-evolution via cross-layer attention as in Eq. (14) of the revised version.
> \begin{align}
> p_ {\theta}(\mathbf{X}_ {l,t}^{(0)}|\mathbf{X}_ {l,t}^{(s)},\mathbf{X}_ {(1:L),(0:t-1)}^{(0)})&=\sum_ {G_ {t}^{(C)}}p_ {\theta}(\mathbf{X}_ {l,t}^{(0)}|\mathbf{X}_ {l,t}^{(s)},\mathbf{X}_ {(1:L),(0:t-1)}^{(0)},G_ {t}^{(C)})p_ {\theta}(G_ {t}^{(C)}|\mathbf{X}_ {(1:L),(0:t-1)}^{(0)}), \text{(14)}
> \end{align}
> where $p_ {\theta}(G_ {t}^{(C)}|\mathbf{X}_ {(1:L),(0:t-1)}^{(0)})$ serves as a learnable prior over cross-layer dependencies. It maps the clean historical snapshots into a latent adjacency-like structure that reflects the current degree of coupling across layers at timestamp $t$. This latent graph then parameterizes the cross-layer attention mechanism in $p_ {\theta}(\mathbf{X}_ {l,t}^{(0)}|\mathbf{X}_ {l,t}^{(s)},\mathbf{X}_ {(1:L),(0:t-1)}^{(0)},G_ {t}^{(C)})$, meaning that the denoiser’s parameters are modulated by the inferred strength of cross-layer correlations.
> As a result, cross-layer dependencies are injected before factorization, ensuring that the final product form in Eq. (13) still embeds inter-layer influence. This mechanism allows the model to capture both edge-level cross-layer correlations (when explicit inter-layer edges exist) and higher-order co-evolution patterns (when groups of nodes across layers move together in time, even without explicit inter-layer links).
>     * (4) The reverse process of a multi-layer graph sequence conditioning on the clean snapshots $\mathbf{X}_ {(1:L),(0:t-1)}^{(0)}$ is the product of reverse process of all layers as in Eq. (15) of the revised version:
> \begin{align}
> p_ {\theta}(\mathbf{X}_ {(1:L),t}^{(s-1)}|\mathbf{X}_ {(1:L),t}^{(s)},\mathbf{X}_ {(1:L),(0:t-1)}^{(0)})=\prod_ {l=1}^{L}p_ {\theta}(\mathbf{X}_ {l,t}^{(s-1)}|\mathbf{X}_ {l,t}^{(s)},\mathbf{X}_ {(1:L),(0:t-1)}^{(0)}), \text{(15)}
> \end{align}
> where the reverse process of each layer $l$ can be derived by approximation with the posterior (Eq. (12)) and denoiser (Eq. (14)) of each layer $l$ as in Eq. (16) of the revised version,
> \begin{align}
> p_ {\theta}(\mathbf{X}_ {l,t}^{(s-1)}|\mathbf{X}_ {l,t}^{(s)},\mathbf{X}_ {(1:L),(0:t-1)}^{(0)})&=\sum_ {\mathbf{X}_ {l,1}^{(0)}}q(\mathbf{X}_ {l,t}^{(s-1)}|\mathbf{X}_ {l,t}^{(s)},\mathbf{X}_ {l,(0:t)}^{(0)})p_ {\theta}(\mathbf{X}_ {l,t}^{(0)}|\mathbf{X}_ {l,t}^{(s)},\mathbf{X}_ {(1:L),(0:t-1)}^{(0)}), \text{(16)}
> \end{align}
> The reverse process factorizes across layers because each layer produces its own categorical transition from step $s$ to $s−1$. However, the transition probabilities themselves are not independent: they depend on the shared cross-layer attention weights computed from the predicted coupling graph $G_{t}^{(C)}$. Thus, each layer’s reverse update incorporates information from all other layers before generating its own transition. This design cleanly separates (1) how information flows across layers (attention), and (2) how categorical diffusion is applied per layer (reverse transition), making the derivation tractable while still capturing rich cross-layer co-evolution.

---

> ### Author Response · Authors · 2025-11-24
> **Answer to Q2 (Part 8)**
>
> 4. Intuitive description.
> Figure 2 (p. 4 in our paper) provides an intuitive view of what the diffusion process captures. The forward process derived in Eqs. (4) to (7) of the revised version (red arrows) gradually adds noise while continually mixing in clean historical snapshots through the dynamic transition-aware mixture. As a result, each timestamp reflects the accumulated temporal context rather than depending only on $t-1$, enabling the reverse process derived in Eq. (10) of the revised version (blue arrows) to denoise with a history-aware prior and recover long-range temporal patterns that simple one-step models miss. In addition, the cross-layer correlation predictor ($p_ {\theta}(G_ {t}^{(C)}|\mathbf{X}_ {(1:L),(0:t-1)}^{(0)})$ in Eq. (14) of the revised version) provides time-varying weights indicating how strongly different layers should influence one another during denoising, allowing the model to exploit implicit cross-layer co-evolution even when no explicit inter-layer edges exist. Moreover, the behavior guidance (Eq. (20) in Sec. 4.2.3 of the revised version) examines the graph-level behavior of the generated graph with Kuramoto model-based synchronization degree to regularize graph-level behavior of generated graphs to be similar to input graphs.

---

> ### Author Response · Authors · 2025-11-24
> **Answer to Q3**
>
> Thank you for the valuable comments. As suggested, 1) we clarify (in Appendix H of the revised version) that we use the Kuramoto order parameter purely as a phase-coherence descriptor without simulating continuous-time Kuramoto dynamics; therefore, no alignment between oscillator time and graph timestamps is required. 2) We also add a discussion on using spectral properties for behavior guidance in Appendix H of the revised version, following the multiplex-spectral framework [TCyb24,SIAM21].
> 1. Clarification that no alignment is required.
> To make behavior guidance computationally feasible inside the diffusion steps, we compute the Kuramoto order parameter snapshot-wise by projecting node attributes to phases, rather than simulating continuous-time Kuramoto dynamics. This avoids the substantial computational and implementation complexity that full Kuramoto ODE simulation would require, and keeps the guidance practical for long temporal sequences. In addition, our formulation avoids any assumptions about timescales. Since the order parameter is computed independently for each snapshot, there is no continuous oscillator time variable that must be aligned with the discrete timestamps of the temporal graph.
> 2. Spectral properties for behavior guidance.
> We thank the reviewer for the insightful suggestion regarding the use of spectral properties for behavior guidance. The idea is fully compatible with our formulation, and the spectral quantities such as  grounded supra-Laplacian eigenvalues [TCyb24] and spectral heterogeneity in [SIAM21] can naturally serve as behavior descriptors within our behavior-aware guidance mechanism.
>     * Pinning-control spectral theory in [TCyb24] provides grounded supra-Laplacian eigenvalues such as $\lambda_{1}(\tilde{L})$ which quantify the structural tendency of a multiplex network to sustain or resist coherent states. A derived index such as $1/\lambda_ {1}(\tilde{L})$ can be directly inserted into the behavior-aware loss as a structural guidance signal.
>     * Multiplex decomposition and generalized master stability analysis offer mode-wise stability parameters $\psi_ {i}$ derived from the supra-Laplacian spectrum. Their distribution (e.g., $\text{Var}_ {i}(\psi_ {i})$) characterizes spectral heterogeneity across modes and can likewise be integrated as a structural behavior descriptor.
>
> These spectral quantities serve as structural indicators that enrich the behavioral information available to the diffusion process. Our formulation of behavior guidance does not require any modifications to accommodate these spectral terms, and their inclusion naturally broadens the range of behaviors that can be captured.
>
> [TCyb24] H. Liu, J. Li, J. Zhao, X. Wu, Z. Zeng and J. Lü, "Pinning Control of Multiplex Dynamical Networks Using Spectral Graph Theory," in IEEE Transactions on Cybernetics, vol. 54, no. 9, pp. 5309-5322, Sept. 2024.
>
> [SIAM21] Rico Berner, et al. "The multiplex decomposition: An analytic framework for multilayer dynamical networks." SIAM Journal on Applied Dynamical Systems 20.4 (2021): 1752-1772.

---

> ### Author Response · Authors · 2025-11-24
> **Answer to Q4**
>
> Thank you for the valuable comments. As suggested, we clarify the gap between our work and prior temporal graph generators from the aspects of (1) temporal dependency, (2) multi-layer structure, (3) inductivity, (4) graph-level behavior guidance, (5) global and local evolution. The differences are summarized in the table below.
> 1. Temporal dependency: While several existing models indeed condition on only a single snapshot (the immediately preceding snapshot $G_ {t-1}$ [MIDL24, KDD22] or $G_ {t-\Delta t}$ [AAAI22]) or merely local motif transition statistics [WWW21,KDD23], our contribution is not the mere use of temporal conditioning but the design of an explicitly history-mixing forward diffusion that recursively aggregates all past snapshots $G_{0:t-1}$ when defining the distribution at time $t$. This formulation provides long-range temporal coupling and continuity that extend beyond the typical one-step conditioning scheme.
> Specifically, Section 4.1 introduces an attribute-aware dynamic transition mixture $\text{D}$ (Eq. (1) of the revised version). For $t \geq 1$, the forward process combines the current snapshot with the recursively accumulated context from all previous timestamps, so that the marginal $q\left(\mathbf{X}^{(s)}_ {l,t} \mid \mathbf{X}^{(0)}_ {l,0:t}\right)$ (Eq. (4) of the revised version) depends on the entire trajectory up to $t$. This enables the generation of future snapshots to exploit long-range temporal signals. In contrast, many existing temporal graph generators specify their forward or conditional distributions using only $G_ {t-1}$, and therefore do not couple multiple past snapshots within the forward process.
> 2. Multi-layer structure: Among the compared methods, most of the compared studies only consider single-layer structure; only DBGDGM in [MIDL24] handles multiple aspects. In contrast, MulDyDiff models cross-layer correlation during generation, supported by a learned cross-layer coupling graph. MulDyDiff not only captures temporal evolution in a single layer but also co-evolution of each layer influenced by other layers.
> 3. Inductivity: Most previous works are transductive since they do not consider unseen nodes. MulDyDiff (ours) and TIGGER-I in [AAAI22] are inductive since the former adopts a permutation-invariant temporal graph transformer architecture, which does not rely on node ID information; the latter builds a multi-mode decoder to learn distributions of node embeddings; and the others are not inductive since they tend to use fixed nodes and cannot generalize to unseen nodes.
> 4. Graph-level behavior guidance: None of the prior temporal or multiplex generators model system-level phenomena such as explosive synchronization or hysteresis. In contrast, MulDyDiff is the first to introduce behavior-aware guidance to reproduce these global dynamics, enabling to regularize graph-level behavior of generated graphs to be similar to input graphs.
> 5. Global and local evolution:
> In contrast to previous works focusing on generating the current snapshot conditioning on only a single snapshot (or merely local motif transition statistics) with only local evolution taken into account, our model enables future snapshot generation considering both local and global evolution from given historical snapshots.
>
> |Models|Temporal dependency|Multi-layer structure|Inductive|Graph-level behavior guidance|Global evolution|Local evolution|
> |-|-|-|-|-|-|-|
> |MulDyDiff (ours)|$p(G_{t} \mid G_{0:t-1})$|O (with cross-layer dependency)|O|O|O|O|
> |[MIDL24]|$p(G_{t} \mid G_{t-1})$|O (without cross-layer dependency)|X|X|X|O|
> |[KDD23]|local motif transition statistics|X|X|X|X|O|
> |[KDD22]|$p(G_{t} \mid G_{t-1})$|X|X|X|X|O|
> |[AAAI22]|$p(G_{t} \mid G_{t-\Delta t})$|X|O|X|X|O|
> |[WWW21]|local motif transition statistics|X|X|X|X|O|
>
> The above comparison table is presented in Table 7 in Appendix B.3 of the revised paper.
>
> [MIDL24] Alexander Campbell, et al. "DBGDGM: Dynamic Brain Graph Deep Generative Model." Medical Imaging with Deep Learning. PMLR, 2024.
>
> [KDD23] Penghang Liu and Ahmet Erdem Sariyüce. 2023. Using Motif Transitions for Temporal Graph Generation. In Proceedings of the 29th ACM SIGKDD Conference on Knowledge Discovery and Data Mining (KDD '23).
>
> [KDD22] Junshan Wang, Wenhao Zhu, Guojie Song, and Liang Wang. 2022. Streaming Graph Neural Networks with Generative Replay. In Proceedings of the 28th ACM SIGKDD Conference on Knowledge Discovery and Data Mining (KDD '22).
>
> [AAAI22] Shubham Gupta, Sahil Manchanda, Sayan Ranu, and Srikanta Bedathur. Tigger: Scalable generative modelling for temporal interaction graphs. In Proc. of the 36th AAAI Conference on Artificial Intelligence (AAAI), 2022.
>
> [WWW21] Giselle Zeno, Timothy La Fond, and Jennifer Neville. Dymond: Dynamic motif-nodes network generative model. In Proceedings of the Web Conference 2021, WWW ’21, pp. 718–729, New York, NY, USA, 2021. Association for Computing Machinery.

---

> ### Author Response · Authors · 2025-11-24
> **Answer to Q5**
>
> Thank you for the valuable comments. As suggested, 1) we have added full results on the third dataset to the main evaluation (the results will be updated as long as the experiment is finished). 2) We use the Kolmogorov-Smirnov (KS) statistic between the per-layer degree/betweenness centrality distributions, the cross-layer node-behavior distribution, and the cross-layer random-walk reachability distribution to examine whether the generator preserves the role heterogeneity, cross-layer engagement, and accessibility that shape dynamics on temporal multiplex graphs. 3) We have also added a table listing all hyperparameters used in the main results.
> * The results on the Wiki-vote dataset (with an additional baseline MoDiff [KDD25]) are listed in the following table. On the Wiki-vote dataset, MulDyDiff outperforms the baselines in almost all metrics listed in the table, as it captures structural and attributive evolution simultaneously. Some baselines perform slightly better in the KS of random walk on the Wiki-vote dataset. Nevertheless, MulDyDiff overall outperforms these baselines since they only perform well in one or two MMD or KS metrics. This is insufficient to demonstrate the effectiveness of the baselines in multi-layer temporal graph generation, as effectiveness needs to be assessed comprehensively by various metrics.
>
> |Models|MMD (degree)|MMD (spectre)|KS (node behavior)|KS (random walk)|KS (degree centrality)|KS (betweenness centrality)|
> |-|-|-|-|-|-|-|
> |AGE|0.5145|0.3395|0.8052|0.0766|0.9991|0.8592|
> |DAMNET|0.3904|0.6162|0.6853|0.1805|0.7152|0.7318|
> |TagGen|0.6365|0.4231|0.9500|0.1400|0.8750|0.8500|
> |DYMOND|0.5384|0.3069|0.8256|0.2111|0.8398|0.8941|
> |MoDiff|0.9919|0.4127|1.0000|0.4000|0.9474|0.8500|
> |MulDyDiff|0.3676|0.3214|0.5430|0.2219|0.8281|0.6563|
>
> The part of KS in the above results is presented in Table 2 in Sec. 5 of the revised version; the part of MMD is presented in Table 11 in Appendix K of the revised version.
>
> [KDD25] Yuwei Xu and Chenhao Ma. 2025. MoDiff - Graph Generation with Motif-aware Diffusion Model. In Proceedings of the 31st ACM SIGKDD Conference on Knowledge Discovery and Data Mining V.2 (KDD '25).
>
> * Explanation of metrics
>    * KS metrics: We choose KS metrics since it better captures local and global property changes of graph sequences than MMD. According to [WWW21], the importance of KS metrics for significant graph property change detection lies in 1) assessing the discrepancy of the distributions between input and generated graph sequences, 2) capturing individual behavior of each node and joint behavior of all nodes in a graph snapshots, and 3) benefits in capturing variability or dispersion with KS tests on inter-quartile range, in which the latter two cannot be achieved by MMD.
>    * Per-layer degree/betweenness centrality distributions: This metric measures the distributional fidelity of degree and betweenness centrality via the KS distance. Matching these distributions verifies whether the generator preserves the heterogeneity of roles within each layer (i.e., the relative proportions of hubs, bridges, and peripheral nodes), which strongly influences temporal dynamics.
>    * Cross-layer node-behavior distribution: It is defined as the number of unique neighbors a node has across all layers. This metric assesses whether the model accurately reproduces the heterogeneity of cross-layer engagement, thereby complementing the per-layer centrality metrics.
>    * Cross-layer random-walk reachability distribution: This metric evaluates cross-layer, multi-hop accessibility by the number of distinct nodes reachable within a fixed-length random walk. It tests whether the generator preserves higher-order structural connectivity beyond direct neighbors.
> * Hyperparameter setting
>    * The hyperparameter settings used to derive the main results are listed in the following table:
>
> |Hyperparameter|Value|
> |-|-|
> |Graph transformer layers|$6$|
> |Batch size|$64$|
> |Epochs|$50$|
> |Learning rate|$0.2$|
> |Weight decay|$10^{-12}$|
> |Diffusion steps $S$|$1000$|
> |$\overline{\alpha}_ {s}$|$\cos^{2}{[0.5\pi(\tfrac{s/S+p}{1+p})]}$, $p=0.008$|
> |$\overline{\gamma}_ {s}$|$\eta\overline{\alpha}_ {s}+(1-\eta)$, $\eta=0.5$|
>
> The above hyperparameter settings are listed in Table 10 in Appendix J of the revised version.
>
> [WWW21] Giselle Zeno, Timothy La Fond, and Jennifer Neville. Dymond: Dynamic motif-nodes network generative model. In Proceedings of the Web Conference 2021, WWW ’21, pp. 718–729, New York, NY, USA, 2021. Association for Computing Machinery.

---

> ### Author Response · Authors · 2025-11-24
> **Answer to Q1 (Part 1)**
>
> Thank you for the valuable comments. As suggested, 1) we clarify the motivation of preservation of graph-level synchronization behavior inherent to the input multiplex graph, which cannot be guaranteed by structural generators, with a practical example. 2) We explain how behavior guidance (explosive synchronization and hysteresis) in our model is performed with the practical example. 3) We agree that multiplex structure is not a strict precondition for explosive synchronization or hysteresis. Nevertheless, following [Nature19, Nature23], many real systems are naturally cross-context, and multiplex representations provide clearer access to interdependence, competition, and asymmetry across layers, which in turn makes behavior-preserving generation more interpretable and controllable.
>
> 1. Practical motivation for behavior guidance.
> To clarify the motivation for behavior-aware guidance, we provide an illustrative example in Figure 8 (in Appendix H of the revised version) inspired by the dynamics of real social platforms such as Instagram or X. These platforms support multiple interaction types—most notably repost/share and reply/comment—and each interaction has a directional nature. When user $A$ reposts user $B$’s post, $A$’s active repost count increases and $B$’s passive repost count increases. The same active/passive semantics apply to replies. Such data naturally form a temporal multiplex network, where
>     * each layer corresponds to an interaction type (e.g., repost vs. reply), and
>     * each attribute corresponds to the direction of participation (active vs. passive activity).
>
> This representation reflects the fact that user behavior is usually not uniform across interactions: frequent reposters may rarely reply, and vice versa.
>
> Consider a common scenario on Instagram. A brand plans a giveaway and instructs its followers:
> * Before the giveaway opens ($t=1,2$):
> Followers are encouraged to repost the promotional poster to tell their friends.
> This produces moderately elevated repost activity across the network—users amplify the message, but not explosively.
> * At the giveaway time ($t=3$): The brand announces “Leave a comment on our post now for a chance to win!” Suddenly, a large number of users reply at exactly the same moment, producing a sharp, collective surge—an explosive synchronization event—in the reply layer. Importantly, the reply activity at earlier times does not show strong pre-burst signs, even though the repost layer has already become active due to brand promotion.
> * This pattern—one interaction channel warming up, while another exhibits a sudden synchronized burst—is extremely common across real social platforms.
>
> To make this concrete, we consider three nodes $i=1,2,3$ connecting with one another, two layers: repost ($l=1$) and reply ($l=2$), and two directional activity attributes: active ($m=1$) and passive ($m=2$).
> We construct a minimal temporal sequence of interaction records $\mathbf{x}_ {l,t}[i,m]$ (e.g., $\mathbf{x}_ {1,2}[3,1]=5$ is the frequency of active ($m=1$) repost ($l=1$) at $t=2$) consistent with the above scenario:
> \begin{align}
> \mathbf{x}_ {1,1}[i,1]=[1,2,3]; \mathbf{x}_ {1,2}[i,1]=[2,3,4]; \mathbf{x}_ {1,3}[i,1]=[1,2,1],
> \end{align}
> representing moderately elevated active reposts during brand promotion.
> Passive reposts are also moderately elevated:
> \begin{align}
> \mathbf{x}_ {1,1}[i,2]=[0,1,0]; \mathbf{x}_ {1,2}[i,2]=[1,2,1]; \mathbf{x}_ {1,3}[i,2]=[2,2,2].
> \end{align}
> For the reply layer, the pre-event active reply activity is low:
> \begin{align}
> \mathbf{x}_ {2,1}[i,1]=[0,0,0]; \mathbf{x}_ {2,2}[i,1]=[1,2,1],
> \end{align}
> but when the giveaway opens at $t=3$, users synchronously comment: $\mathbf{x}_ {2,3}[i,1]=[5,5,5].$
>
> Passive replies are similarly stable:
> \begin{align}
> \mathbf{x}_ {2,1}[i,2]=[0,1,0]; \mathbf{x}_ {2,2}[i,2]=[1,1,1]; \mathbf{x}_ {2,3}[i,2]=[1,1,1].
> \end{align}
>
> Thus, the repost layer shows early promotional activity at $t=2$, while the reply layer exhibits a sudden burst at $t=3$, mimicking the real behavior seen on social platforms during time-sensitive events.
> This example highlights that collective temporal behavior, such as event-driven synchronous reply bursts, cannot be captured by structure-only generators. Traditional models replicate edges, degrees, and multiplex topology but fail to reproduce: 1) burst intensity and 2) cross-interaction causal influence (repost → reply burst).
>
> Thus, behavior-aware guidance is essential for generating temporal multiplex networks that faithfully preserve how users behave over time—not just the connection of each other.

---

> ### Author Response · Authors · 2025-11-24
> **Answer to Q1 (Part 2)**
>
> 2. Motivated by the above observation, we follow the Kuramoto model to define the phase angle of the $m$-th attribute of each node $i$ in layer $l$ at timestamp $t$ by using the attribute value $\mathbf{x}_ {l,t}^{(0)}[i, m]$ of the $m$-th attribute (e.g., the frequency of delivering reposts) of each node $i$ in layer $l$ at timestamp $t$ in Eq. (25) in Appendix H of the revised version (with a typo fixed):
> \begin{align}
> \theta_ {i,m}^{(l)}(t) = \pi\cdot\tfrac{(\mathbf{x}_ {l,t}^{(0)}[i, m] - \min_ {j}\mathbf{x}_ {l,t}^{(0)}[j, m])}{(\max_ {j}\mathbf{x}_ {l,t}^{(0)}[j, m] - \min_ {j}\mathbf{x}_ {l,t}^{(0)}[j, m])},
> \end{align}
> representing the position of node $i$ associated with the $m$-th attribute on layer $l$.
> We map these activity levels of active ($m=1$) reply ($l=2$) to Kuramoto phases as follows:
> \begin{align}
> &\theta_ {1,1}^{(2)}(1)=\theta_ {2,1}^{(2)}(1)=\theta_ {3,1}^{(2)}(1)=0; \\\\
> &\theta_ {1,1}^{(2)}(2)=\pi\cdot\tfrac{1-1}{2-1}=0;\theta_ {2,1}^{(2)}(2)=\pi\cdot\tfrac{2-1}{2-1}=\pi;\theta_ {3,1}^{(2)}(2)=\pi\cdot\tfrac{1-1}{2-1}=0; \\\\
> &\theta_ {1,1}^{(2)}(3)=\theta_ {2,1}^{(2)}(3)=\theta_ {3,1}^{(2)}(3)=0,
> \end{align}
> which yields dispersed phases at $t=1$, perfectly aligned phases at the burst time $t=2$, and dispersed phases again at $t=3$.
> Then we denote the synchronization degree $R_ {m}^{(l)}(t)$ of the $m$-th attribute in interaction layer $l$ at timestamp $t$ by following Kuramoto model $R_ {m}^{(l)}(t) = \left| \tfrac{1}{N_ {t}^{(l)}} \sum\nolimits_ {i=1}^{N_ {t}^{(l)}} e^{j \cdot \theta_ {i,m}^{(l)}(t)} \right|$ (Eq.(24) in Appendix H), which quantifies collective phase coherence: values near 1 indicate that many nodes occupy nearly the same behavioral stage, whereas values near 0 reflect dispersed or uncoordinated behavior.
> To observe the explosive synchronization, we define the first-order difference of Kuramoto order parameter $\Delta R^{(l)}_ {m}(t)=R^{(l)}_ {m}(t)-R^{(l)}_ {m}(t-1), \forall t=1,\ldots,T-1$ and calculate its variance as in Eq. (19) in Sec. 4.2.3 of the revised version:
> \begin{align}
> \text{Var}^{(l)}_ {m} &= \tfrac{1}{T - 1} \sum\nolimits_ {t=1}^{T-1} \left( \Delta R^{(l)}_ {m}(t)-\overline{\Delta R}^{(l)}_ {m} \right)^2,
> \end{align}
> where $\overline{\Delta R}^{(l)}_ {m} = \tfrac{1}{T-1} \sum\nolimits_ {t=1}^{T-1} \Delta R^{(l)}_ {m}(t)$ is the average of the first-order differences of Kuramoto order parameters.
> We calculate the corresponding order parameters accordingly to obtain
> \begin{align}
> R_ {1}^{(2)}(1)=|\tfrac{1}{3}(1+1+1)|=1, R_ {1}^{(2)}(2)=|\tfrac{1}{3}(1-1+1)|=1/3, R_ {1}^{(2)}(3)=|\tfrac{1}{3}(1+1+1)|=1,
> \end{align}
> whose first-order differences $\Delta R_ {1}^{(2)}(1)=1/3-1=-2/3, \Delta R_ {1}^{(2)}(2)=1-1/3=2/3$, and $\overline{\Delta R}_ {1}^{(2)}=0$ produce a large variance $\text{Var}_ {1}^{(2)}=\tfrac{1}{3-1}[(-2/3)^{2}+(2/3)^{2}]=4/9$ by Eq. (19) in Sec. 4.2.3, capturing a strong explosive-synchronization phenomenon of active reply. The calculation results of the variances in all layers and attributes are listed in the following Table 6 in Appendix H.
> |Layer $l$|Attribute $m$|$R_ {m}^{(l)}(t)$ ($t=1,2,3$)|$\Delta R_ {m}^{(l)}(t)$ ($t=1,2$)|$Var_ {m}^{(l)}$|
> |-|-|-|-|-|
> |1|1|$\\{1/3,1/3,1/3\\}$|$\\{0,0\\}$|$0$|
> |1|2|$\\{1/3,1/3,1\\}$|$\\{0,+2/3\\}$|$1/9$|
> |2|1|$\\{1,1/3,1\\}$|$\\{−2/3,+2/3\\}$|$4/9$|
> |2|2|$\\{1/3,1,1\\}$|$\\{+2/3,0\\}$|$1/9$|
>
> From the above results, since the variance $Var_ {1}^{(2)}$ in active reply behaviors is the largest among all behavior types, we observe that the explosive synchronization phenomenon of active reply behaviors is strong.
> To observe the hysteresis phenomenon, we follow the above example and denote $R_ {m,f}$ and $R_ {m,b}$ as the sum of increasing and decreasing values of $R_{m}^{(l)}(t)$, respectively, to separate the synchronization trajectory into forward and backward segments:
> \begin{align*}
> R_ {m,f} = \sum\nolimits_ {t:\Delta R_ {m}^{(l)}(t)>0}R_ {m}^{(l)}(t); R_ {m,b} = \sum\nolimits_{t:\Delta R_ {m}^{(l)}(t)<0}R_ {m}^{(l)}(t).
> \end{align*}
> Moreover, the rising and falling trajectories of active reply behaviors differ ($R_ {1,f}=R_ {1}^{(2)}(2)=1/3, R_ {1,b}=R_ {1}^{(2)}(3)=1 \Rightarrow |R_ {m,f}-R_ {b,f}|=1-1/3=2/3$), indicating a clear hysteresis gap. The values of $|R_ {m,f}-R_ {b,f}|$ in all layers and attributes are calculated in the following Table 7 in Appendix H.
> |Layer $l$|Attribute $m$|$R_ {m}^{(l)}(t)$ ($t=1,2,3$)|$\Delta R_ {m}^{(l)}(t)$ ($t=1,2$)|$\|R_ {m,f}-R_ {b,f}\|$|
> |-|-|-|-|-|
> |1|1|$\\{1/3,1/3,1/3\\}$|$\\{0,0\\}$|$0$|
> |1|2|$\\{1/3,1/3,1\\}$|$\\{0,+2/3\\}$|$1$|
> |2|1|$\\{1,1/3,1\\}$|$\\{−2/3,+2/3\\}$|$2/3$|
> |2|2|$\\{1/3,1,1\\}$|$\\{+2/3,0\\}$|$1$|
>
> Then we use $\mathcal{L}_ {\text{hyst}}=-|R_ {m,f}-R_ {m,b}|$ (Eq.(26) in Appendix H) to guide the generation process to preserve the hysteresis phenomenon. This encourages the generator to preserve the asymmetric rise-and-decay pattern (as in the repost-deliver burst) rather than collapsing to a reversible trajectory.

---

> ### Author Response · Authors · 2025-11-24
> **Answer to Q1 (Part 3)**
>
> 3. Rationale for using a temporal multiplex representation.
> We clarify (in Appendix H of the revised version) that multiplexity is not required, but is a natural and effective modeling choice when the underlying system involves interacting contexts. While explosive synchronization and hysteresis can arise on non-multiplex graphs, our motivation for adopting temporal multiplex graphs follows [Nature19, Nature23], which emphasize that many real systems exhibiting such behaviors unfold across multiple interacting contexts (e.g., different social platforms). In such settings, single-layer representations often collapse cross-context dependencies that critically shape the resulting dynamics, making it difficult for a generative model to preserve behavior. Temporal multiplex graphs, by contrast, offer an explicit structure for representing interdependence, competition, and asymmetry across layers, which better preserves the mechanisms driving explosive synchronization and hysteresis.
>
> [Nature19] Danziger, M.M., Bonamassa, I., Boccaletti, S. et al. Dynamic interdependence and competition in multilayer networks. Nature Phys 15, 178–185 (2019).
>
> [Nature23] De Domenico, M. More is different in real-world multilayer networks. Nat. Phys. 19, 1247–1262 (2023).

---

### Note · Authors · 2026-01-12

I have read and agree with the venue's withdrawal policy on behalf of myself and my co-authors.